# Gradient Descent Maximizes the Margin of Homogeneous Neural Networks

**Kaifeng Lyu & Jian Li**
Institute for Interdisciplinary Information Sciences
Tsinghua University
Beijing, China
`vfleaking@gmail.com,lijian83@mail.tsinghua.edu.cn`

## Abstract

In this paper, we study the implicit regularization of the gradient descent algorithm in homogeneous neural networks, including fully-connected and convolutional neural networks with ReLU or LeakyReLU activations. In particular, we study the gradient descent or gradient flow (i.e., gradient descent with infinitesimal step size) optimizing the logistic loss or cross-entropy loss of any homogeneous model (possibly non-smooth), and show that if the training loss decreases below a certain threshold, then we can define a smoothed version of the normalized margin which increases over time. We also formulate a natural constrained optimization problem related to margin maximization, and prove that both the normalized margin and its smoothed version converge to the objective value at a KKT point of the optimization problem. Our results generalize the previous results for logistic regression with one-layer or multi-layer linear networks, and provide more quantitative convergence results with weaker assumptions than previous results for homogeneous smooth neural networks. We conduct several experiments to justify our theoretical finding on MNIST and CIFAR-10 datasets. Finally, as margin is closely related to robustness, we discuss potential benefits of training longer for improving the robustness of the model.

## 1 Introduction

A major open question in deep learning is why gradient descent or its variants, are biased towards solutions with good generalization performance on the test set. To achieve a better understanding, previous works have studied the implicit bias of gradient descent in different settings. One simple but insightful setting is linear logistic regression on linearly separable data. In this setting, the model is parameterized by a weight vector $\boldsymbol{w}$, and the class prediction for any data point $\boldsymbol{x}$ is determined by the sign of $\boldsymbol{w}^\top \boldsymbol{x}$. Therefore, only the direction $\boldsymbol{w}/\|\boldsymbol{w}\|_2$ is important for making prediction. Soudry et al. (2018a;b); Ji & Telgarsky (2018; 2019c); Nacson et al. (2019c) investigated this problem and proved that the direction of $\boldsymbol{w}$ converges to the direction that maximizes the $L^2$-margin while the norm of $\boldsymbol{w}$ diverges to $+\infty$, if we train $\boldsymbol{w}$ with (stochastic) gradient descent on logistic loss. Interestingly, this convergent direction is the same as that of any *regularization path*: any sequence of weight vectors $\{\boldsymbol{w}_t\}$ such that every $\boldsymbol{w}_t$ is a global minimum of the $L^2$-regularized loss $\mathcal{L}(\boldsymbol{w}) + \frac{\lambda_t}{2}\|\boldsymbol{w}\|_2^2$ with $\lambda_t \to 0$ (Rosset et al., 2004). Indeed, the trajectory of gradient descent is also pointwise close to a regularization path (Suggala et al., 2018).

The aforementioned linear logistic regression can be viewed as a single-layer neural network. A natural and important question is to what extent gradient descent has similiar implicit bias for modern deep neural networks. For theoretical analysis, a natural candidate is to consider *homogeneous neural networks*. Here a neural network $\Phi$ is said to be (positively) homogeneous if there is a number $L > 0$ (called the *order*) such that the network output $\Phi(\boldsymbol{\theta}; \boldsymbol{x})$, where $\boldsymbol{\theta}$ stands for the parameter and $\boldsymbol{x}$ stands for the input, satisfies the following:

$$\forall c > 0 : \Phi(c\boldsymbol{\theta}; \boldsymbol{x}) = c^L \Phi(\boldsymbol{\theta}; \boldsymbol{x}) \text{ for all } \boldsymbol{\theta} \text{ and } \boldsymbol{x}. \tag{1}$$

It is important to note that many neural networks are homogeneous (Neyshabur et al., 2015a; Du et al., 2018). For example, deep fully-connected neural networks or deep CNNs with ReLU or

LeakyReLU activations can be made homogeneous if we remove all the bias terms, and the order $L$ is exactly equal to the number of layers.

In (Wei et al., 2019), it is shown that the regularization path does converge to the max-margin direction for homogeneous neural networks with cross-entropy or logistic loss. This result suggests that gradient descent or gradient flow may also converges to the max-margin direction by assuming homogeneity, and this is indeed true for some sub-classes of homogeneous neural networks. For gradient flow, this convergent direction is proven for linear fully-connected networks (Ji & Telgarsky, 2019a). For gradient descent on linear fully-connected and convolutional networks, (Gunasekar et al., 2018b) formulate a constrained optimization problem related to margin maximization and prove that gradient descent converges to the direction of a KKT point or even the max-margin direction, under various assumptions including the convergence of loss and gradient directions. In an independent work, (Nacson et al., 2019a) generalize the result in (Gunasekar et al., 2018b) to smooth homogeneous models (we will discuss this work in more details in Section 2).

## 1.1 MAIN RESULTS

In this paper, we identify a minimal set of assumptions for proving our theoretical results for homogeneous neural networks on classification tasks. Besides homogeneity, we make two additional assumptions:

1. **Exponential-type Loss Function.** We require the loss function to have certain exponential tail (see Appendix A for the details). This assumption is not restrictive as it includes the most popular classfication losses: exponential loss, logistic loss and cross-entropy loss.

2. **Separability.** The neural network can separate the training data during training (i.e., the neural network can achieve $100\%$ training accuracy)[1].

While the first assumption is natural, the second requires some explanation. In fact, we assume that at some time $t_0$, the training loss is smaller than a threshold, and the threshold here is chosen to be so small that the training accuracy is guaranteed to be $100\%$ (e.g., for the logistic loss and cross-entropy loss, the threshold can be set to $\ln 2$). Empirically, state-of-the-art CNNs for image classification can even fit randomly labeled data easily (Zhang et al., 2017). Recent theoretical work on over-parameterized neural networks (Allen-Zhu et al., 2019; Zou et al., 2018) show that gradient descent can fit the training data if the width is large enough. Furthermore, in order to study the margin, ensuring the training data can be separated is inevitable; otherwise, there is no positive margin between the data and decision boundary.

**Our Contribution.** Similar to linear models, for homogeneous models, only the direction of parameter $\boldsymbol{\theta}$ is important for making predictions, and one can see that the margin $\gamma(\boldsymbol{\theta})$ scales linearly with $\|\boldsymbol{\theta}\|_2^L$, when fixing the direction of $\boldsymbol{\theta}$. To compare margins among $\boldsymbol{\theta}$ in different directions, it makes sense to study the *normalized margin*, $\bar{\gamma}(\boldsymbol{\theta}) := \gamma(\boldsymbol{\theta})/\|\boldsymbol{\theta}\|_2^L$.

In this paper, we focus on the training dynamics of the network after $t_0$ (recall that $t_0$ is a time that the training loss is less than the threshold). Our theoretical results can answer the following questions regarding the normalized margin.

First, *how does the normalized margin change during training?* The answer may seem complicated since one can easily come up with examples in which $\bar{\gamma}$ increases or decreases in a short time interval. However, we can show that the overall trend of the normalized margin is to increase in the following sense: there exists a *smoothed* version of the normalized margin, denoted as $\tilde{\gamma}$, such that (1) $|\tilde{\gamma} - \bar{\gamma}| \to 0$ as $t \to \infty$; and (2) $\tilde{\gamma}$ is non-decreasing for $t > t_0$.

Second, *how large is the normalized margin at convergence?* To answer this question, we formulate a natural constrained optimization problem which aims to directly maximize the margin. We show that every limit point of $\{\boldsymbol{\theta}(t)/\|\boldsymbol{\theta}(t)\|_2 : t > 0\}$ is along the direction of a KKT point of the max-margin problem. This indicates that gradient descent/gradient flow performs margin maximization implicitly in deep homogeneous networks. This result can be seen as a significant generalization of previous works (Soudry et al., 2018a;b; Ji & Telgarsky, 2019a; Gunasekar et al., 2018b) from linear classifiers to homogeneous classifiers.

---

[1]Note that this does NOT mean the training loss is 0.

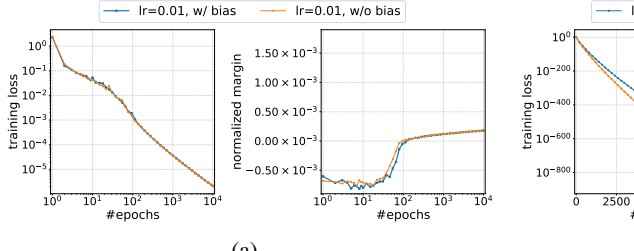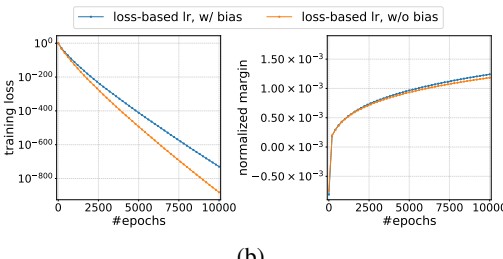

(a)                                      (b)

Figure 1: (a) Training CNNs with and without bias on MNIST, using SGD with learning rate $0.01$. The training loss (left) decreases over time, and the normalized margin (right) keeps increasing after the model is fitted, but the growth rate is slow ($\approx 1.8 \times 10^{-4}$ after 10000 epochs). (b) Training CNNs with and without bias on MNIST, using SGD with the loss-based learning rate scheduler. The training loss (left) decreases exponentially over time ($< 10^{-800}$ after 9000 epochs), and the normalized margin (right) increases rapidly after the model is fitted ($\approx 1.2 \times 10^{-3}$ after 10000 epochs, $10\times$ larger than that of SGD with learning rate $0.01$). Experimental details are in Appendix K.

As by-products of the above results, we derive tight asymptotic convergence/growth rates of the loss and weights. It is shown in (Soudry et al., 2018a;b; Ji & Telgarsky, 2018; 2019c) that the loss decreases at the rate of $O(1/t)$, the weight norm grows as $O(\log t)$ for linear logistic regression. In this work, we generalize the result by showing that the loss decreases at the rate of $O(1/(t(\log t)^{2-2/L}))$ and the weight norm grows as $O((\log t)^{1/L})$ for homogeneous neural networks with exponential loss, logistic loss, or cross-entropy loss.

**Experiments.**[2]    The main practical implication of our theoretical result is that training longer can enlarge the normalized margin. To justify this claim empiricaly, we train CNNs on MNIST and CIFAR-10 with SGD (see Section K.1). Results on MNIST are presented in Figure 1. For constant step size, we can see that the normalized margin keeps increasing, but the growth rate is rather slow (because the gradient gets smaller and smaller). Inspired by our convergence results for gradient descent, we use a learning rate scheduling method which enlarges the learning rate according to the current training loss, then the training loss decreases exponentially faster and the normalized margin increases significantly faster as well.

For feedforward neural networks with ReLU activation, the normalized margin on a training sample is closely related to the $L^2$-*robustness* (the $L^2$-distance from the training sample to the decision boundary). Indeed, the former divided by a Lipschitz constant is a lower bound for the latter. For example, the normalized margin is a lower bound for the $L^2$-robustness on fully-connected networks with ReLU activation (see, e.g., Theorem 4 in (Sokolic et al., 2017)). This fact suggests that training longer may have potential benefits on improving the robustness of the model. In our experiments, we observe noticeable improvements of $L^2$-robustness on both training and test sets (see Section K.2).

## 2    RELATED WORK

**Implicit Bias in Training Linear Classifiers.**    For linear logistic regression on linearly separable data, Soudry et al. (2018a;b) showed that full-batch gradient descent converges in the direction of the max $L^2$-margin solution of the corresponding hard-margin Support Vector Machine (SVM). Subsequent works extended this result in several ways: Nacson et al. (2019c) extended the results to the case of stochastic gradient descent; Gunasekar et al. (2018a) considered other optimization methods; Nacson et al. (2019b) considered other loss functions including those with poly-exponential tails; Ji & Telgarsky (2018; 2019c) characterized the convergence of weight direction without assuming separability; Ji & Telgarsky (2019b) proved a tighter convergence rate for the weight direction.

Those results on linear logistic regression have been generalized to deep linear networks. Ji & Telgarsky (2019a) showed that the product of weights in a deep linear network with strictly decreasing loss converges in the direction of the max $L^2$-margin solution. Gunasekar et al. (2018b) showed

---

[2]Code available: `https://github.com/vfleaking/max-margin`

more general results for gradient descent on linear fully-connected and convolutional networks with exponential loss, under various assumptions on the convergence of the loss and gradient direction.

Margin maximization phenomenon is also studied for boosting methods (Schapire et al., 1998; Schapire & Freund, 2012; Shalev-Shwartz & Singer, 2010; Telgarsky, 2013) and Normalized Perceptron (Ramdas & Pena, 2016).

**Implicit Bias in Training Nonlinear Classifiers.** Soudry et al. (2018a) analyzed the case where there is only one trainable layer of a ReLU network. Xu et al. (2018) characterized the implicit bias for the model consisting of one single ReLU unit. Our work is closely related to a recent independent work by (Nacson et al., 2019a) which we discuss in details below.

**Comparison with (Nacson et al., 2019a).** Very recently, (Nacson et al., 2019a) analyzed gradient descent for smooth homogeneous models and proved the convergence of parameter direction to a KKT point of the aforementioned max-margin problem. Compared with their work, our work adopt much weaker assumptions: (1) They assume the training loss converges to 0, but in our work we only require that the training loss is lower than a small threshold value at some time $t_0$ (and we prove the exact convergence rate of the loss after $t_0$); (2) They assume the convergence of parameter direction[3], while we prove that KKT conditions hold for all limit points of $\{\boldsymbol{\theta}(t)/\|\boldsymbol{\theta}(t)\|_2 : t > 0\}$, without requiring any convergence assumption; (3) They assume the convergence of the direction of losses (the direction of the vector whose entries are loss values on every data point) and Linear Independence Constraint Qualification (LICQ) for the max-margin problem, while we do not need such assumptions. Besides the above differences in assumptions, we also prove the monotonicity of the normalized margin and provide tight convergence rate for training loss. We believe both results are interesting in their own right.

Another technical difference is that their work analyzes discrete gradient descent on smooth homogeneous models (which fails to capture ReLU networks). In our work, we analyze both gradient descent on smooth homogeneous models and also gradient flow on homogeneous models which could be non-smooth.

**Other Works on Implicit Bias.** Banburski et al. (2019) also studied the dynamics of gradient flow and among other things, provided mathematical insights to the implicit bias towards max margin solution for homogeneous networks. We note that their analysis of gradient flow decomposes the dynamics to the tangent component and radial component, which is similar to our proof of Theorem 4.1 in spirit. Wilson et al. (2017); Ali et al. (2019); Gunasekar et al. (2018a) showed that for the linear least-square problem gradient-based methods converge to the unique global minimum that is closest to the initialization in $L^2$ distance. Du et al. (2019); Jacot et al. (2018); Lee et al. (2019); Arora et al. (2019b) showed that over-parameterized neural networks of sufficient width (or infinite width) behave as linear models with Neural Tangent Kernel (NTK) with proper initialization and gradient descent converges linearly to a global minimum near the initial point. Other related works include (Ma et al., 2019; Gidel et al., 2019; Arora et al., 2019a; Suggala et al., 2018; Blanc et al., 2019; Neyshabur et al., 2015b;a).

## 3 PRELIMINARIES

**Basic Notations.** For any $N \in \mathbb{N}$, let $[N] = \{1, \ldots, N\}$. $\|\boldsymbol{v}\|_2$ denotes the $L^2$-norm of a vector $\boldsymbol{v}$. The default base of $\log$ is $e$. For a function $f : \mathbb{R}^d \to \mathbb{R}$, $\nabla f(\boldsymbol{x})$ stands for the gradient at $\boldsymbol{x}$ if it exists. A function $f : X \to \mathbb{R}^d$ is $\mathcal{C}^k$-smooth if $f$ is $k$ times continuously differentiable. A function $f : X \to \mathbb{R}$ is locally Lipschitz if for every $\boldsymbol{x} \in X$ there exists a neighborhood $U$ of $\boldsymbol{x}$ such that the restriction of $f$ on $U$ is Lipschitz continuous.

**Non-smooth Analysis.** For a locally Lipschitz function $f : X \to \mathbb{R}$, the Clarke's subdifferential (Clarke, 1975; Clarke et al., 2008; Davis et al., 2020) at $\boldsymbol{x} \in X$ is the convex set

$$\partial^\circ f(\boldsymbol{x}) := \text{conv} \left\{ \lim_{k \to \infty} \nabla f(\boldsymbol{x}_k) : \boldsymbol{x}_k \to \boldsymbol{x}, f \text{ is differentiable at } \boldsymbol{x}_k \right\}.$$

---

[3] Assuming the convergence of the parameter direction may seem quite reasonable, however, the problem here can be quite subtle in theory. In Appendix J, we present a smooth homogeneuous function $f$, based on the Mexican hat function (Absil et al. (2005)), such that even the direction of the parameter does not converge along gradient flow (it moves around a cirle when $t$ increases).

For brevity, we say that a function $\boldsymbol{z} : I \to \mathbb{R}^d$ on the interval $I$ is an *arc* if $\boldsymbol{z}$ is absolutely continuous for any compact sub-interval of $I$. For an arc $\boldsymbol{z}$, $\boldsymbol{z}'(t)$ (or $\frac{d\boldsymbol{z}}{dt}(t)$) stands for the derivative at $t$ if it exists. Following the terminology in (Davis et al., 2020), we say that a locally Lipschitz function $f : \mathbb{R}^d \to \mathbb{R}$ *admits a chain rule* if for any arc $\boldsymbol{z} : [0, +\infty) \to \mathbb{R}^d$, $\forall \boldsymbol{h} \in \partial^\circ f(z(t)) : (f \circ \boldsymbol{z})'(t) = \langle \boldsymbol{h}, \boldsymbol{z}'(t) \rangle$ holds for a.e. $t > 0$ (see also Appendix I).

**Binary Classification.** Let $\Phi$ be a neural network, assumed to be parameterized by $\boldsymbol{\theta}$. The output of $\Phi$ on an input $\boldsymbol{x} \in \mathbb{R}^{d_\times}$ is a real number $\Phi(\boldsymbol{\theta}; \boldsymbol{x})$, and the sign of $\Phi(\boldsymbol{\theta}; \boldsymbol{x})$ stands for the classification result. A dataset is denoted by $\mathcal{D} = \{(\boldsymbol{x}_n, y_n) : n \in [N]\}$, where $\boldsymbol{x}_n \in \mathbb{R}^{d_\times}$ stands for a data input and $y_n \in \{\pm 1\}$ stands for the corresponding label. For a loss function $\ell : \mathbb{R} \to \mathbb{R}$, we define the training loss of $\Phi$ on the dataset $\mathcal{D}$ to be $\mathcal{L}(\boldsymbol{\theta}) := \sum_{n=1}^{N} \ell(y_n \Phi(\boldsymbol{\theta}; \boldsymbol{x}_n))$.

**Gradient Descent.** We consider the process of training this neural network $\Phi$ with either gradient descent or gradient flow. For gradient descent, we assume the training loss $\mathcal{L}(\boldsymbol{\theta})$ is $\mathcal{C}^2$-smooth and describe the gradient descnet process as $\boldsymbol{\theta}(t+1) = \boldsymbol{\theta}(t) - \eta(t) \nabla \mathcal{L}(\boldsymbol{\theta}(t))$, where $\eta(t)$ is the learning rate at time $t$ and $\nabla \mathcal{L}(\boldsymbol{\theta}(t))$ is the gradient of $\mathcal{L}$ at $\boldsymbol{\theta}(t)$.

**Gradient Flow.** For gradient flow, we do not assume the differentibility but only some regularity assumptions including locally Lipschitz. Gradient flow can be seen as gradient descent with infinitesimal step size. In this model, $\boldsymbol{\theta}$ changes continuously with time, and the trajectory of parameter $\boldsymbol{\theta}$ during training is an arc $\boldsymbol{\theta} : [0, +\infty) \to \mathbb{R}^d, t \mapsto \boldsymbol{\theta}(t)$ that satisfies the differential inclusion $\frac{d\boldsymbol{\theta}(t)}{dt} \in -\partial^\circ \mathcal{L}(\boldsymbol{\theta}(t))$ for a.e. $t \geq 0$. The Clarke's subdifferential $\partial^\circ \mathcal{L}$ is a natural generalization of the usual differential to non-differentiable functions. If $\mathcal{L}(\boldsymbol{\theta})$ is actually a $\mathcal{C}^1$-smooth function, the above differential inclusion reduces to $\frac{d\boldsymbol{\theta}(t)}{dt} = -\nabla \mathcal{L}(\boldsymbol{\theta}(t))$ for all $t \geq 0$, which corresponds to the gradient flow with differential in the usual sense.

## 4 GRADIENT DESCENT / GRADIENT FLOW ON HOMOGENEOUS MODEL

In this section, we first state our results for gradient flow and gradient descent on homogeneous models with exponential loss $\ell(q) := e^{-q}$ for simplicity of presentation. Due to space limit, we defer the more general results which hold for a large family of loss functions (including logistic loss and cross-entropy loss) to Appendix A, F and G.

### 4.1 ASSUMPTIONS

**Gradient Flow.** For gradient flow, we assume the following:

**(A1).** (Regularity). For any fixed $\boldsymbol{x}$, $\Phi(\,\cdot\,; \boldsymbol{x})$ is locally Lipschitz and admits a chain rule;

**(A2).** (Homogeneity). There exists $L > 0$ such that $\forall \alpha > 0 : \Phi(\alpha\boldsymbol{\theta}; \boldsymbol{x}) = \alpha^L \Phi(\boldsymbol{\theta}; \boldsymbol{x})$;

**(A3).** (Exponential Loss). $\ell(q) = e^{-q}$;

**(A4).** (Separability). There exists a time $t_0$ such that $\mathcal{L}(\boldsymbol{\theta}(t_0)) < 1$.

(A1) is a technical assumption about the regularity of the network output. As shown in (Davis et al., 2020), the output of almost every neural network admits a chain rule (as long as the neural network is composed by definable pieces in an o-minimal structure, e.g., ReLU, sigmoid, LeakyReLU).

(A2) assumes the homogeneity, the main property we assume in this work. (A3), (A4) correspond to the two conditions introduced in Section 1. The exponential loss in (A3) is main focus of this section. (A4) is a separability assumption: the condition $\mathcal{L}(\boldsymbol{\theta}(t_0)) < 1$ ensures that $\ell(y_n \Phi(\boldsymbol{\theta}(t_0); \boldsymbol{x}_n)) < 1$ for all $n \in [N]$, and thus $y_n \Phi(\boldsymbol{\theta}(t_0); \boldsymbol{x}_n) > 0$, meaning that $\Phi$ classifies every $\boldsymbol{x}_n$ correctly.

**Gradient Descent.** For gradient descent, we assume (A2), (A3), (A4) similarly as for gradient flow, and the following two assumptions (S1) and (S5).

**(S1).** (Smoothness). For any fixed $\boldsymbol{x}$, $\Phi(\,\cdot\,; \boldsymbol{x})$ is $\mathcal{C}^2$-smooth on $\mathbb{R}^d \setminus \{\boldsymbol{0}\}$.

**(S5).** (Learning rate condition, Informal). $\eta(t) = \eta_0$ for a sufficiently small constant $\eta_0$. In fact, $\eta(t)$ is even allowed to be as large as $O(\mathcal{L}(t)^{-1} \text{polylog} \frac{1}{\mathcal{L}(t)})$. See Appendix E.1 for the details.

(S5) is natural since deep neural networks are usually trained with constant learning rates. (S1) ensures the smoothness of $\Phi$, which is often assumed in the optimization literature in order to analyze gradient descent. While (S1) does not hold for neural networks with ReLU, it does hold for neural networks with smooth homogeneous activation such as the quadratic activation $\phi(x) := x^2$ (Li et al., 2018b; Du & Lee, 2018) or powers of ReLU $\phi(x) := \mathrm{ReLU}(x)^\alpha$ for $\alpha > 2$ (Zhong et al., 2017; Klusowski & Barron, 2018; Li et al., 2019).

## 4.2 MAIN THEOREM: MONOTONICITY OF NORMALIZED MARGINS

The *margin* for a single data point $(\boldsymbol{x}_n, y_n)$ is defined to be $q_n(\boldsymbol{\theta}) := y_n \Phi(\boldsymbol{\theta}; \boldsymbol{x}_n)$, and the margin for the entire dataset is defined to be $q_{\min}(\boldsymbol{\theta}) := \min_{n \in [N]} q_n(\boldsymbol{\theta})$. By homogenity, the margin $q_{\min}(\boldsymbol{\theta})$ scales linearly with $\|\boldsymbol{\theta}\|_2^L$ for any fixed direction since $q_{\min}(c\boldsymbol{\theta}) = c^L q_{\min}(\boldsymbol{\theta})$. So we consider the *normalized margin* defined as below:

$$\bar{\gamma}(\boldsymbol{\theta}) := q_{\min}\left(\frac{\boldsymbol{\theta}}{\|\boldsymbol{\theta}\|_2}\right) = \frac{q_{\min}(\boldsymbol{\theta})}{\|\boldsymbol{\theta}\|_2^L}. \tag{2}$$

We say $f$ is an $\epsilon$-*additive approximation* for the normalized margin if $\bar{\gamma} - \epsilon \leq f \leq \bar{\gamma}$, and $c$-*multiplicative approximation* if $c\bar{\gamma} \leq f \leq \bar{\gamma}$.

**Gradient Flow.** Our first result is on the overall trend of the normalized margin $\bar{\gamma}(\boldsymbol{\theta}(t))$. For both gradient flow and gradient descent, we identify a smoothed version of the normalized margin, and show that it is non-decreasing during training. More specifically, we have the following theorem for gradient flow.

**Theorem 4.1** (Corollary of Theorem A.7). *Under assumptions (A1) - (A4), there exists an $O(\|\boldsymbol{\theta}\|_2^{-L})$-additive approximation function $\tilde{\gamma}(\boldsymbol{\theta})$ for the normalized margin such that the following statements are true for gradient flow:*

1. *For a.e. $t > t_0$, $\frac{d}{dt}\tilde{\gamma}(\boldsymbol{\theta}(t)) \geq 0$;*

2. *For a.e. $t > t_0$, either $\frac{d}{dt}\tilde{\gamma}(\boldsymbol{\theta}(t)) > 0$ or $\frac{d}{dt}\frac{\boldsymbol{\theta}(t)}{\|\boldsymbol{\theta}(t)\|_2} = 0$;*

3. *$\mathcal{L}(\boldsymbol{\theta}(t)) \to 0$ and $\|\boldsymbol{\theta}(t)\|_2 \to \infty$ as $t \to +\infty$; therefore, $|\bar{\gamma}(\boldsymbol{\theta}(t)) - \tilde{\gamma}(\boldsymbol{\theta}(t))| \to 0$.*

More concretely, the function $\tilde{\gamma}(\boldsymbol{\theta})$ in Theorem 4.1 is defined as

$$\tilde{\gamma}(\boldsymbol{\theta}) := \frac{\log\frac{1}{\mathcal{L}(\boldsymbol{\theta})}}{\|\boldsymbol{\theta}\|_2^L} = \frac{-\log\left(\sum_{n=1}^N e^{-q_n(\boldsymbol{\theta})}\right)}{\|\boldsymbol{\theta}\|_2^L}. \tag{3}$$

Note that the only difference between $\bar{\gamma}(\boldsymbol{\theta})$ and $\tilde{\gamma}(\boldsymbol{\theta})$ is that the margin $q_{\min}(\boldsymbol{\theta})$ in $\bar{\gamma}(\boldsymbol{\theta})$ is replaced by $\log\frac{1}{\mathcal{L}(\boldsymbol{\theta})} = -\mathrm{LSE}(-q_1(\boldsymbol{\theta}), \ldots, -q_N(\boldsymbol{\theta}))$, where $\mathrm{LSE}(a_1, \ldots, a_N) = \log(\exp(a_1) + \cdots + \exp(a_N))$ is the LogSumExp function. This is indeed a very natural idea, and previous works on linear models (e.g., (Telgarsky, 2013; Nacson et al., 2019b)) also approximate $q_{\min}$ with LogSumExp in the analysis of margin. It is easy to see why $\tilde{\gamma}(\boldsymbol{\theta})$ is an $O(\|\boldsymbol{\theta}\|_2^{-L})$-additive approximation for $\bar{\gamma}(\boldsymbol{\theta})$: $e^{a_{\max}} \leq \sum_{n=1}^N e^{a_n} \leq Ne^{a_{\max}}$ holds for $a_{\max} = \max\{a_1, \ldots, a_N\}$, so $a_{\max} \leq \mathrm{LSE}(a_1, \ldots, a_N) \leq a_{\max} + \log N$; combining this with the definition of $\tilde{\gamma}(\boldsymbol{\theta})$ gives $\bar{\gamma}(\boldsymbol{\theta}) - \|\boldsymbol{\theta}\|_2^{-L}\log N \leq \tilde{\gamma}(\boldsymbol{\theta}) \leq \bar{\gamma}(\boldsymbol{\theta})$.

**Gradient Descent.** For gradient descent, Theorem 4.1 holds similarly with a slightly different function $\hat{\gamma}(\boldsymbol{\theta})$ that approximates $\bar{\gamma}(\boldsymbol{\theta})$ multiplicatively rather than additively.

**Theorem 4.2** (Corollary of Theorem E.2). *Under assumptions (S1), (A2) - (A4), (S5), there exists an $(1 - O(1/(\log\frac{1}{\mathcal{L}})))$-multiplicative approximation function $\hat{\gamma}(\boldsymbol{\theta})$ for the normalized margin such that the following statements are true for gradient descent:*

1. *For all $t > t_0$, $\hat{\gamma}(\boldsymbol{\theta}(t+1)) \geq \hat{\gamma}(\boldsymbol{\theta}(t))$;*

2. *For all $t > t_0$, either $\hat{\gamma}(\boldsymbol{\theta}(t+1)) > \hat{\gamma}(\boldsymbol{\theta}(t))$ or $\frac{\boldsymbol{\theta}(t+1)}{\|\boldsymbol{\theta}(t+1)\|_2} = \frac{\boldsymbol{\theta}(t)}{\|\boldsymbol{\theta}(t)\|_2}$;*

3. *$\mathcal{L}(\boldsymbol{\theta}(t)) \to 0$ and $\|\boldsymbol{\theta}(t)\|_2 \to \infty$ as $t \to +\infty$; therefore, $|\bar{\gamma}(\boldsymbol{\theta}(t)) - \hat{\gamma}(\boldsymbol{\theta}(t))| \to 0$.*

Due to the discreteness of gradient descent, the explicit formula for $\hat{\gamma}(\boldsymbol{\theta})$ is somewhat technical, and we refer the readers to Appendix E for full details.

**Convergence Rates.** It is shown in Theorem 4.1, 4.2 that $\mathcal{L}(\boldsymbol{\theta}(t)) \to 0$ and $\|\boldsymbol{\theta}(t)\|_2 \to \infty$. In fact, with a more refined analysis, we can prove tight loss convergence and weight growth rates using the monotonicity of normalized margins.

**Theorem 4.3** (Corollary of Theorem A.10 and E.5). *For gradient flow under assumptions (A1) - (A4) or gradient descent under assumptions (S1), (A2) - (A4), (S5), we have the following tight bounds for training loss and weight norm:*

$$\mathcal{L}(\boldsymbol{\theta}(t)) = \Theta\left(\frac{1}{T(\log T)^{2-2/L}}\right) \qquad and \qquad \|\boldsymbol{\theta}(t)\|_2 = \Theta\left((\log T)^{1/L}\right),$$

*where $T = t$ for gradient flow and $T = \sum_{\tau=t_0}^{t-1} \eta(\tau)$ for gradient descent.*

### 4.3 MAIN THEOREM: CONVERGENCE TO KKT POINTS

For gradient flow, $\tilde{\gamma}$ is upper-bounded by $\tilde{\gamma} \leq \bar{\gamma} \leq \sup\{q_n(\boldsymbol{\theta}) : \|\boldsymbol{\theta}\|_2 = 1\}$. Combining this with Theorem 4.1 and the monotone convergence theorem, it is not hard to see that $\lim_{t \to +\infty} \bar{\gamma}(\boldsymbol{\theta}(t))$ and $\lim_{t \to +\infty} \tilde{\gamma}(\boldsymbol{\theta}(t))$ exist and equal to the same value. Using a similar argument, we can draw the same conclusion for gradient descent.

To understand the implicit regularization effect, a natural question arises: what optimality property does the limit of normalized margin have? To this end, we identify a natural constrained optimization problem related to margin maximization, and prove that $\boldsymbol{\theta}(t)$ directionally converges to its KKT points, as shown below. We note that we can extend this result to the finite time case, and show that gradient flow or gradient descent passes through an approximate KKT point after a certain amount of time. See Theorem A.9 in Appendix A and Theorem E.4 in Appendix E for the details. We will briefly review the definition of KKT points and approximate KKT points for a constraint optimization problem in Appendix C.1.

**Theorem 4.4** (Corollary of Theorem A.8 and E.3). *For gradient flow under assumptions (A1) - (A4) or gradient descent under assumptions (S1), (A2) - (A4), (S5), any limit point $\bar{\boldsymbol{\theta}}$ of $\left\{\frac{\boldsymbol{\theta}(t)}{\|\boldsymbol{\theta}(t)\|_2} : t \geq 0\right\}$ is along the direction of a KKT point of the following constrained optimization problem (P):*

$$\min \quad \frac{1}{2}\|\boldsymbol{\theta}\|_2^2 \qquad s.t. \quad q_n(\boldsymbol{\theta}) \geq 1 \qquad \forall n \in [N]$$

*That is, for any limit point $\bar{\boldsymbol{\theta}}$, there exists a scaling factor $\alpha > 0$ such that $\alpha\bar{\boldsymbol{\theta}}$ satisfies Karush-Kuhn-Tucker (KKT) conditions of (P).*

Minimizing (P) over its feasible region is equivalent to maximizing the normalized margin over all possible directions. The proof is as follows. Note that we only need to consider all feasible points $\boldsymbol{\theta}$ with $q_{\min}(\boldsymbol{\theta}) > 0$. For a fixed $\boldsymbol{\theta}$, $\alpha\boldsymbol{\theta}$ is a feasible point of (P) iff $\alpha \geq q_{\min}(\boldsymbol{\theta})^{-1/L}$. Thus, the minimum objective value over all feasible points of (P) in the direction of $\boldsymbol{\theta}$ is $\frac{1}{2}\|\boldsymbol{\theta}/q_{\min}(\boldsymbol{\theta})^{1/L}\|_2^2 = \frac{1}{2}\bar{\gamma}(\boldsymbol{\theta})^{-2/L}$. Taking minimum over all possible directions, we can conclude that if the maximum normalized margin is $\bar{\gamma}_*$, then the minimum objective of (P) is $\frac{1}{2}\bar{\gamma}_*^{-2/L}$.

It can be proved that (P) satisfies the Mangasarian-Fromovitz Constraint Qualification (MFCQ) (See Lemma C.7). Thus, KKT conditions are first-order necessary conditions for global optimality. For linear models, KKT conditions are also sufficient for ensuring global optimality; however, for deep homogeneous networks, $q_n(\boldsymbol{\theta})$ can be highly non-convex. Indeed, as gradient descent is a first-order optimization method, if we do not make further assumptions on $q_n(\boldsymbol{\theta})$, then it is easy to construct examples that gradient descent does not lead to a normalized margin that is globally optimal. Thus, proving the convergence to KKT points is perhaps the best we can hope for in our setting, and it is an interesting future work to prove stronger convergence results with further natural assumptions.

Moreover, we can prove the following corollary, which characterizes the optimality of the normalized margin using SVM with Neural Tangent Kernel (NTK, introduced in (Jacot et al., 2018)) defined at limit points. The proof is deferred to Appendix C.6.

**Corollary 4.5** (Corollary of Theorem 4.4). *Assume (S1). Then for gradient flow under assumptions (A2) - (A4) or gradient descent under assumptions (A2) - (A4), (S5), any limit point $\bar{\boldsymbol{\theta}}$ of $\{\boldsymbol{\theta}(t)/\|\boldsymbol{\theta}(t)\|_2 : t \geq 0\}$ is along the max-margin direction for the hard-margin SVM with kernel*

$K_{\bar{\boldsymbol{\theta}}}(\boldsymbol{x}, \boldsymbol{x}') := \langle \nabla \Phi_{\boldsymbol{x}}(\bar{\boldsymbol{\theta}}), \nabla \Phi_{\boldsymbol{x}'}(\bar{\boldsymbol{\theta}}) \rangle$, *where* $\Phi_{\boldsymbol{x}}(\boldsymbol{\theta}) := \Phi(\boldsymbol{\theta}; \boldsymbol{x})$. *That is, for some* $\alpha > 0$, $\alpha\bar{\boldsymbol{\theta}}$ *is the optimal solution for the following constrained optimization problem:*

$$\min \quad \frac{1}{2}\|\boldsymbol{\theta}\|_2^2 \qquad s.t. \quad y_n \langle \boldsymbol{\theta}, \nabla\Phi_{\boldsymbol{x}_n}(\bar{\boldsymbol{\theta}}) \rangle \geq 1 \qquad \forall n \in [N]$$

*If we assume (A1) instead of (S1) for gradient flow, then there exists a mapping* $\boldsymbol{h}(\boldsymbol{x}) \in \partial^{\circ}\Phi_{\boldsymbol{x}}(\bar{\boldsymbol{\theta}})$ *such that the same conclusion holds for* $K_{\bar{\boldsymbol{\theta}}}(\boldsymbol{x}, \boldsymbol{x}') = \langle \boldsymbol{h}(\boldsymbol{x}), \boldsymbol{h}(\boldsymbol{x}') \rangle$.

### 4.4 OTHER MAIN RESULTS

The above results can be extended to other settings as shown below.

**Other Binary Classification Loss.** The results on exponential loss can be generalized to a much broader class of binary classification loss. The class includes the logistic loss which is one of the most popular loss functions, $\ell(q) = \log(1 + e^{-q})$. The function class also includes other losses with exponential tail, e.g., $\ell(q) = e^{-q^3}, \ell(q) = \log(1 + e^{-q^3})$. For all those loss functions, we can use its inverse function $\ell^{-1}$ to define the smoothed normalized margin as follows

$$\tilde{\gamma}(\boldsymbol{\theta}) = \frac{\ell^{-1}(\mathcal{L}(\boldsymbol{\theta}))}{\|\boldsymbol{\theta}\|_2^L}.$$

Then all our results for gradient flow continue to hold (Appendix A). Using a similar modification, we can also extend it to gradient descent (Appendix F).

**Cross-entropy Loss.** In multi-class classification, we can define $q_n$ to be the difference between the classification score for the true label and the maximum score for the other labels, then the margin $q_{\min} := \min_{n \in [N]} q_n$ and the normalized margin $\bar{\gamma}(\boldsymbol{\theta}) := \frac{q_{\min}(\boldsymbol{\theta})}{\|\boldsymbol{\theta}\|_2^L}$ can be similarly defined as before. In Appendix G, we define the smoothed normalized margin for cross-entropy loss to be the same as that for logistic loss (See Remark A.4). Then we show that Theorem 4.1 and Theorem 4.4 still hold (but with a slightly different definition of (P)) for gradient flow, and we also extend the results to gradient descent.

**Multi-homogeneous Models.** Some neural networks indeed possess a stronger property than homogeneity, which we call *multi-homogeneity*. For example, the output of a CNN (without bias terms) is 1-homogeneous with respect to the weights of each layer. In general, we say that a neural network $\Phi(\boldsymbol{\theta}; \boldsymbol{x})$ with $\boldsymbol{\theta} = (\boldsymbol{w}_1, \ldots, \boldsymbol{w}_m)$ is $(k_1, \ldots, k_m)$-homogeneous if for any $\boldsymbol{x}$ and any $c_1, \ldots, c_m > 0$, we have $\Phi(c_1\boldsymbol{w}_1, \ldots, c_m\boldsymbol{w}_m; \boldsymbol{x}) = \prod_{i=1}^m c_i^{k_i} \cdot \Phi(\boldsymbol{w}_1, \ldots, \boldsymbol{w}_m; \boldsymbol{x})$. In the previous example, an $L$-layer CNN with layer weights $\boldsymbol{\theta} = (\boldsymbol{w}_1, \ldots, \boldsymbol{w}_L)$ is $(1, \ldots, 1)$-homogeneous.

One can easily see that that $(k_1, \ldots, k_m)$-homogeneity implies $L$-homogeneity, where $L = \sum_{i=1}^m k_i$, so our previous analysis for homogeneous models still applies to multi-homogeneous models. But it would be better to define the normalized margin for multi-homogeneous model as

$$\bar{\gamma}(\boldsymbol{w}_1, \ldots, \boldsymbol{w}_m) := q_{\min}\left(\frac{\boldsymbol{w}_1}{\|\boldsymbol{w}_1\|_2}, \ldots, \frac{\boldsymbol{w}_m}{\|\boldsymbol{w}_m\|_2}\right) = \frac{q_{\min}}{\prod_{i=1}^m \|\boldsymbol{w}_i\|_2^{k_i}}. \tag{4}$$

In this case, the smoothed approximation of $\bar{\gamma}$ for general binary classification loss (under some conditions) can be similarly defined for gradient flow:

$$\tilde{\gamma}(\boldsymbol{w}_1, \ldots, \boldsymbol{w}_m) := \frac{\ell^{-1}(\mathcal{L})}{\prod_{i=1}^m \|\boldsymbol{w}_i\|_2^{k_i}}, \tag{5}$$

It can be shown that $\tilde{\gamma}$ is also non-decreasing during training when the loss is small enough (Appendix H). In the case of cross-entropy loss, we can still define $\tilde{\gamma}$ by (5) while $\ell(\cdot)$ is set to the logistic loss in the formula.

## 5 PROOF SKETCH: GRADIENT FLOW ON HOMOGENEOUS MODEL WITH EXPONENTIAL LOSS

In this section, we present a proof sketch in the case of gradient flow on homogeneous model with exponential loss to illustrate our proof ideas. Due to space limit, the proof for the main theorems on gradient flow and gradient descent in Section 4 are deferred to Appendix A and E respectively.

For convenience, we introduce a few more notations for a $L$-homogeneous neural network $\Phi(\boldsymbol{\theta}; \boldsymbol{x})$. Let $\mathcal{S}^{d-1} = \{\boldsymbol{\theta} \in \mathbb{R}^d : \|\boldsymbol{\theta}\|_2 = 1\}$ be the set of $L^2$-normalized parameters. Define $\rho := \|\boldsymbol{\theta}\|_2$ and $\hat{\boldsymbol{\theta}} := \frac{\boldsymbol{\theta}}{\|\boldsymbol{\theta}\|_2} \in \mathcal{S}^{d-1}$ to be the length and direction of $\boldsymbol{\theta}$. For both gradient descent and gradient flow, $\boldsymbol{\theta}$ is a function of time $t$. For convenience, we also view the functions of $\boldsymbol{\theta}$, including $\mathcal{L}(\boldsymbol{\theta}), q_n(\boldsymbol{\theta}), q_{\min}(\boldsymbol{\theta})$, as functions of $t$. So we can write $\mathcal{L}(t) := \mathcal{L}(\boldsymbol{\theta}(t)), q_n(t) := q_n(\boldsymbol{\theta}(t)), q_{\min}(t) := q_{\min}(\boldsymbol{\theta}(t))$.

Lemma 5.1 below is the key lemma in our proof. It decomposes the growth of the smoothed normalized margin into the ratio of two quantities related to the radial and tangential velocity components of $\boldsymbol{\theta}$ respectively. We will give a proof sketch for this later in this section. We believe that this lemma is of independent interest.

**Lemma 5.1** (Corollary of Lemma B.1). *For a.e. $t > t_0$,*

$$\frac{d}{dt} \log \rho > 0 \quad \text{and} \quad \frac{d}{dt} \log \tilde{\gamma} \geq L \left( \frac{d}{dt} \log \rho \right)^{-1} \left\| \frac{d\hat{\boldsymbol{\theta}}}{dt} \right\|_2^2.$$

Using Lemma 5.1, the first two claims in Theorem 4.1 can be directly proved. For the third claim, we make use of the monotonicity of the margin to lower bound the gradient, and then show $\mathcal{L} \to 0$ and $\rho \to +\infty$. Recall that $\tilde{\gamma}$ is an $O(\rho^{-L})$-additive approximation for $\bar{\gamma}$. So this proves the third claim. We defer the detailed proof to Appendix B.

To show Theorem 4.4, we first change the time measure to $\log \rho$, i.e., now we see $t$ as a function of $\log \rho$. So the second inequality in Lemma 5.1 can be rewritten as $\frac{d \log \tilde{\gamma}}{d \log \rho} \geq L \|\frac{d\hat{\boldsymbol{\theta}}}{d \log \rho}\|_2^2$. Integrating on both sides and noting that $\tilde{\gamma}$ is upper-bounded, we know that there must be many instant $\log \rho$ such that $\|\frac{d\hat{\boldsymbol{\theta}}}{d \log \rho}\|_2$ is small. By analyzing the landscape of training loss, we show that these points are "approximate" KKT points. Then we show that every convergent sub-sequence of $\{\hat{\boldsymbol{\theta}}(t) : t \geq 0\}$ can be modified to be a sequence of "approximate" KKT points which converges to the same limit. Then we conclude the proof by applying a theorem from (Dutta et al., 2013) to show that the limit of this convergent sequence of "approximate" KKT points is a KKT point. We defer the detailed proof to Appendix C.

Now we give a proof sketch for Lemma 5.1, in which we derive the formula of $\tilde{\gamma}$ step by step. In the proof, we obtain several clean close form formulas for several relevant quantities, by using the chain rule and Euler's theorem for homogeneous functions extensively.

*Proof Sketch of Lemma 5.1.* For ease of presentation, we ignore the regularity issues of taking derivatives in this proof sketch. We start from the equation $\frac{d\mathcal{L}}{dt} = -\langle \partial^\circ \mathcal{L}(\boldsymbol{\theta}(t)), \frac{d\boldsymbol{\theta}}{dt} \rangle = -\|\frac{d\boldsymbol{\theta}}{dt}\|_2^2$ which follows from the chain rule (see also Lemma I.3). Then we note that $\frac{d\boldsymbol{\theta}}{dt}$ can be decomposed into two parts: the radial component $\boldsymbol{v} := \hat{\boldsymbol{\theta}} \hat{\boldsymbol{\theta}}^\top \frac{d\boldsymbol{\theta}}{dt}$ and the tangent component $\boldsymbol{u} := (\boldsymbol{I} - \hat{\boldsymbol{\theta}} \hat{\boldsymbol{\theta}}^\top) \frac{d\boldsymbol{\theta}}{dt}$.

The radial component is easier to analyze. By the chain rule, $\|\boldsymbol{v}\|_2 = \hat{\boldsymbol{\theta}}^\top \frac{d\boldsymbol{\theta}}{dt} = \frac{1}{\rho} \langle \boldsymbol{\theta}, \frac{d\boldsymbol{\theta}}{dt} \rangle = \frac{1}{\rho} \cdot \frac{1}{2} \frac{d\rho^2}{dt}$. For $\frac{1}{2} \frac{d\rho^2}{dt}$, we have an exact formula:

$$\frac{1}{2} \frac{d\rho^2}{dt} = \left\langle \boldsymbol{\theta}, \frac{d\boldsymbol{\theta}}{dt} \right\rangle = \left\langle \sum_{n=1}^N e^{-q_n} \partial^\circ q_n, \boldsymbol{\theta} \right\rangle = L \sum_{n=1}^N e^{-q_n} q_n. \tag{6}$$

The last equality is due to $\langle \partial^\circ q_n, \boldsymbol{\theta} \rangle = L q_n$ by homogeneity of $q_n$. This is sometimes called Euler's theorem for homogeneous functions (see Theorem B.2). For differentiable $q_n$, it can be easily proved by taking the derivative over $c$ on both sides of $q_n(c\boldsymbol{\theta}) = c^L q_n(\boldsymbol{\theta})$ and letting $c = 1$.

With (6), we can lower bound $\frac{1}{2} \frac{d\rho^2}{dt}$ by

$$\frac{1}{2} \frac{d\rho^2}{dt} = L \sum_{n=1}^N e^{-q_n} q_n \geq L \sum_{n=1}^N e^{-q_n} q_{\min} \geq L \cdot \mathcal{L} \log \frac{1}{\mathcal{L}}, \tag{7}$$

where the last inequality uses the fact that $e^{-q_{\min}} \leq \mathcal{L}$. (7) also implies that $\frac{1}{2} \frac{d\rho^2}{dt} > 0$ for $t > t_0$ since $\mathcal{L}(t_0) < 1$ and $\mathcal{L}$ is non-increasing. As $\frac{d}{dt} \log \rho = \frac{1}{2\rho^2} \frac{d\rho^2}{dt}$, this also proves the first inequality of Lemma 5.1.

Now, we have $\|\boldsymbol{v}\|_2^2 = \frac{1}{\rho^2}\left(\frac{1}{2}\frac{d\rho^2}{dt}\right)^2 = \frac{1}{2}\frac{d\rho^2}{dt}\cdot\frac{d}{dt}\log\rho$ on the one hand; on the other hand, by the chain rule we have $\frac{d\hat{\boldsymbol{\theta}}}{dt} = \frac{1}{\rho^2}(\rho\frac{d\boldsymbol{\theta}}{dt} - \frac{d\rho}{dt}\boldsymbol{\theta}) = \frac{1}{\rho^2}(\rho\frac{d\boldsymbol{\theta}}{dt} - (\hat{\boldsymbol{\theta}}^\top\frac{d\boldsymbol{\theta}}{dt})\boldsymbol{\theta}) = \frac{\boldsymbol{u}}{\rho}$. So we have

$$-\frac{d\mathcal{L}}{dt} = \left\|\frac{d\boldsymbol{\theta}}{dt}\right\|_2^2 = \|\boldsymbol{v}\|_2^2 + \|\boldsymbol{u}\|_2^2 = \frac{1}{2}\frac{d\rho^2}{dt}\cdot\frac{d}{dt}\log\rho + \rho^2\left\|\frac{d\hat{\boldsymbol{\theta}}}{dt}\right\|_2^2$$

Dividing $\frac{1}{2}\frac{d\rho^2}{dt}$ on the leftmost and rightmost sides, we have

$$-\frac{d\mathcal{L}}{dt}\cdot\left(\frac{1}{2}\frac{d\rho^2}{dt}\right)^{-1} = \frac{d}{dt}\log\rho + \left(\frac{d}{dt}\log\rho\right)^{-1}\left\|\frac{d\hat{\boldsymbol{\theta}}}{dt}\right\|_2^2.$$

By $-\frac{d\mathcal{L}}{dt} \geq 0$ and (7), the LHS is no greater than $-\frac{d\mathcal{L}}{dt}\cdot\left(L\cdot\mathcal{L}\log\frac{1}{\mathcal{L}}\right)^{-1} = \frac{1}{L}\log\log\frac{1}{\mathcal{L}}$. Thus we have $\frac{d}{dt}\log\log\frac{1}{\mathcal{L}} - L\frac{d}{dt}\log\rho \geq L\left(\frac{d}{dt}\log\rho\right)^{-1}\left\|\frac{d\hat{\boldsymbol{\theta}}}{dt}\right\|_2^2$, where the LHS is exactly $\frac{d}{dt}\log\tilde{\gamma}$. $\qquad\square$

## 6 DISCUSSION AND FUTURE DIRECTIONS

In this paper, we analyze the dynamics of gradient flow/descent of homogeneous neural networks under a minimal set of assumptions. The main technical contribution of our work is to prove rigorously that for gradient flow/descent, the normalized margin is increasing and converges to a KKT point of a natural max-margin problem. Our results leads to some natural further questions:

- Can we generalize our results for gradient descent on smooth neural networks to non-smooth ones? In the smooth case, we can lower bound the decrement of training loss by the gradient norm squared, multiplied by a factor related to learning rate. However, in the non-smooth case, no such inequality is known in the optimization literature.

- Can we make more structural assumptions on the neural network to prove stronger results? In this work, we use a minimal set of assumptions to show that the convergent direction of parameters is a KKT point. A potential research direction is to identify more key properties of modern neural networks and show that the normalized margin at convergence is locally or globally optimal (in terms of optimizing (P)).

- Can we extend our results to neural networks with bias terms? In our experiments, the normalized margin of the CNN with bias also increases during training despite that its output is non-homogeneous. It is very interesting (and technically challenging) to provide a rigorous proof for this fact.

## ACKNOWLEDGMENTS

The research is supported in part by the National Natural Science Foundation of China Grant 61822203, 61772297, 61632016, 61761146003and the Zhongguancun Haihua Institute for Frontier Information Technology and Turing AI Institute of Nanjing. We thank Liwei Wang for helpful suggestions on the connection between margin and robustness. We thank Sanjeev Arora, Tianle Cai, Simon Du, Jason D. Lee, Zhiyuan Li, Tengyu Ma, Ruosong Wang for helpful discussions.

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

# A    RESULTS FOR GENERAL LOSS

In this section, we state our results for a broad class of binary classification loss. A major consequence of this generalization is that the logistic loss, one of the most popular loss functions, $\ell(q) = \log(1 + e^{-q})$ is included. The function class also includes other losses with exponential tail, e.g., $\ell(q) = e^{-q^3}, \ell(q) = \log(1 + e^{-q^3})$.

## A.1    ASSUMPTIONS

We first focus on gradient flow. We assume (A1), (A2) as we do for exponential loss. For (A3), (A4), we replace them with two weaker assumptions (B3), (B4). All the assumptions are listed below:

**(A1).** (Regularity). For any fixed $\boldsymbol{x}$, $\Phi(\,\cdot\,; \boldsymbol{x})$ is locally Lipschitz and admits a chain rule;

**(A2).** (Homogeneity). There exists $L > 0$ such that $\forall \alpha > 0 : \Phi(\alpha\boldsymbol{\theta}; \boldsymbol{x}) = \alpha^L \Phi(\boldsymbol{\theta}; \boldsymbol{x})$;

**(B3).** The loss function $\ell(q)$ can be expressed as $\ell(q) = e^{-f(q)}$ such that

    **(B3.1).** $f : \mathbb{R} \to \mathbb{R}$ is $\mathcal{C}^1$-smooth.

    **(B3.2).** $f'(q) > 0$ for all $q \in \mathbb{R}$.

    **(B3.3).** There exists $b_f \geq 0$ such that $f'(q)q$ is non-decreasing for $q \in (b_f, +\infty)$, and $f'(q)q \to +\infty$ as $q \to +\infty$.

    **(B3.4).** Let $g : [f(b_f), +\infty) \to [b_f, +\infty)$ be the inverse function of $f$ on the domain $[b_f, +\infty)$. There exists $b_g \geq \max\{2f(b_f), f(2b_f)\}, K \geq 1$ such that $g'(x) \leq Kg'(\theta x)$ and $f'(y) \leq Kf'(\theta y)$ for all $x \in (b_g, +\infty), y \in (g(b_g), +\infty)$ and $\theta \in [1/2, 1)$.

**(B4).** (Separability). There exists a time $t_0$ such that $\mathcal{L}(t_0) < e^{-f(b_f)} = \ell(b_f)$.

(A1) and (A2) remain unchanged. (B3) is satisfied by exponential loss $\ell(q) = e^{-q}$ (with $f(q) = q$) and logistic loss $\ell(q) = \log(1 + e^{-q})$ (with $f(q) = -\log\log(1 + e^{-q})$). (B4) are essentially the same as (A4) but (B4) uses a threshold value that depends on the loss function. Assuming (B3), it is easy to see that (B4) ensures the separability of data since $\ell(q_n) < e^{-f(b_f)}$ implies $q_n > b_f \geq 0$. For logistic loss, we can set $b_f = 0$ (see Remark A.2). So the corresponding threshold value in (B4) is $\ell(0) = \log 2$.

Now we discuss each of the assumptions in (B3). (B3.1) is a natural assumption on smoothness. (B3.2) requires $\ell(\cdot)$ to be monotone decreasing, which is also natural since $\ell(\cdot)$ is used for binary classification. The rest of two assumptions in (B3) characterize the properties of $\ell'(q)$ when $q$ is large enough. (B3.3) is an assumption that appears naturally from the proof. For exponential loss, $f'(q)q = q$ is always non-decreasing, so we can set $b_f = 0$. In (B3.4), the inverse function $g$ is defined. It is guaranteed by (B3.1) and (B3.2) that $g$ always exists and $g$ is also $\mathcal{C}^1$-smooth. Though (B3.4) looks very complicated, it essentially says that $f'(\Theta(q)) = \Theta(f'(q)), g'(\Theta(q)) = \Theta(g'(q))$ as $q \to \infty$. (B3.4) is indeed a technical assumption that enables us to asymptotically compare the loss or the length of gradient at different data points. It is possible to base our results on weaker assumptions than (B3.4), but we use (B3.4) for simplicity since it has already been satisfied by many loss functions such as the aforementioned examples.

We summarize the corresponding $f, g$ and $b_f$ for exponential loss and logistic loss below:

**Remark A.1.** *Exponential loss $\ell(q) = e^{-q}$ satisfies (B3) with*

$$f(q) = q \qquad f'(q) = 1 \qquad g(q) = q \qquad g'(q) = 1 \qquad b_f = 0 \qquad \ell(b_f) = 1.$$

**Remark A.2.** *Logistic loss $\ell(q) = \log(1 + e^{-q})$ satisfies (B3) with*

$$f(q) = -\log\log(1 + e^{-q}) = \Theta(q) \qquad\qquad f'(q) = \frac{e^{-q}}{(1 + e^{-q})\log(1 + e^{-q})} = \Theta(1)$$

$$g(q) = -\log(e^{e^{-q}} - 1) = \Theta(q) \qquad\qquad g'(q) = \frac{e^{-q}}{e^{e^{-q}} - 1} = \Theta(1)$$

$$b_f = 0 \qquad\qquad\qquad\qquad \ell(b_f) = \log 2.$$

The proof for Remark A.1 is trivial. For Remark A.2, we give a proof below.

*Proof for Remark A.2.* By simple calculations, the formulas for $f(q), f'(q), g(q), g'(q)$ are correct. (B3.1) is trivial. $f'(q) = \frac{e^{-q}}{(1+e^{-q})\log(1+e^{-q})} > 0$, so (B3.2) is satisfied. For (B3.3), note that $f'(q)q = \frac{q}{(1+e^q)\log(1+e^{-q})}$. The denominator is a decreasing function since

$$\frac{d}{dq}\left((1+e^q)\log(1+e^{-q})\right) = e^q\log(1+e^{-q}) - 1 < e^q \cdot e^{-q} - 1 = 0.$$

Thus, $f'(q)q$ is a strictly increasing function on $\mathbb{R}$. As $b_f$ is required to be non-negative, we set $b_f = 0$. For proving that $f'(q)q \to +\infty$ and (B4), we only need to notice that $f'(q) \sim \frac{e^{-q}}{1 \cdot e^{-q}} = 1$ and $g'(x) = 1/f'(g(x)) \sim 1$. $\qquad\square$

## A.2 SMOOTHED NORMALIZED MARGIN

For a loss function $\ell(\cdot)$ satisfying (B3), it is easy to see from (B3.2) that its inverse function $\ell^{-1}(\cdot)$ must exist. For this kind of loss functions, we define the smoothed normalized margin as follows:

**Definition A.3.** For a loss function $\ell(\cdot)$ satisfying (B3), the smoothed normalized margin $\tilde{\gamma}(\boldsymbol{\theta})$ of $\boldsymbol{\theta}$ is defined as

$$\tilde{\gamma}(\boldsymbol{\theta}) := \frac{\ell^{-1}(\mathcal{L})}{\rho^L} = \frac{g(\log\frac{1}{\mathcal{L}})}{\rho^L} = \frac{g\left(-\log\left(\sum_{n=1}^N e^{-f(q_n(\boldsymbol{\theta}))}\right)\right)}{\rho^L},$$

where $\ell^{-1}(\cdot)$ is the inverse function of $\ell(\cdot)$ and $\rho := \|\boldsymbol{\theta}\|_2$.

**Remark A.4.** *For logistic loss* $\ell(q) = \log(1+e^{-q})$, $\tilde{\gamma}(\boldsymbol{\theta}) = \rho^{-L}\log\frac{1}{\exp(\mathcal{L})-1}$; *for exponential loss* $\ell(q) = e^{-q}$, $\tilde{\gamma}(\boldsymbol{\theta}) = \rho^{-L}\log\frac{1}{\mathcal{L}}$, *which is the same as* (3).

Now we give some insights on how well $\tilde{\gamma}(\boldsymbol{\theta})$ approximates $\bar{\gamma}(\boldsymbol{\theta})$ using a similar argument as in Section 4.2. Using the LogSumExp function, the smoothed normalized margin $\tilde{\gamma}(\boldsymbol{\theta})$ can also be written as

$$\tilde{\gamma}(\boldsymbol{\theta}) = \frac{g(-\mathrm{LSE}(-f(q_1),\ldots,-f(q_N)))}{\rho^L}.$$

LSE is a $(\log N)$-additive approximation for $\max$. So we can roughly approximate $\tilde{\gamma}(\boldsymbol{\theta})$ by

$$\tilde{\gamma}(\boldsymbol{\theta}) \approx \frac{g(-\max\{-f(q_1),\ldots,-f(q_N)\})}{\rho^L} = \frac{g(f(q_{\min}))}{\rho^L} = \bar{\gamma}(\boldsymbol{\theta}).$$

Note that (B3.3) is crucial to make the above approximation reasonable. Similar to exponential loss, we can show the following lemma asserting that $\tilde{\gamma}$ is a good approximation of $\bar{\gamma}$.

**Lemma A.5.** *Assuming (B3)[4], we have the following properties about the margin:*

*(a)* $f(q_{\min}) - \log N \le \log\frac{1}{\mathcal{L}} \le f(q_{\min})$.

*(b) If* $\log\frac{1}{\mathcal{L}} > f(b_f)$, *then there exists* $\xi \in (f(q_{\min}) - \log N, f(q_{\min})) \cap (b_f, +\infty)$ *such that*

$$\bar{\gamma} - \frac{g'(\xi)\log N}{\rho^L} \le \tilde{\gamma} \le \bar{\gamma}.$$

*(c) For a seuqnce of parameters* $\{\boldsymbol{\theta}_m \in \mathbb{R}^d : m \in \mathbb{N}\}$, *if* $\mathcal{L}(\boldsymbol{\theta}_m) \to 0$, *then* $|\tilde{\gamma}(\boldsymbol{\theta}_m) - \bar{\gamma}(\boldsymbol{\theta}_m)| \to 0$.

*Proof.* (a) can be easily deduced from $e^{-f(q_{\min})} \le \mathcal{L} \le Ne^{-f(q_{\min})}$. Combining (a) and the monotonicity of $g(\cdot)$, we further have $g(s) \le g(\log\frac{1}{\mathcal{L}}) \le q_{\min}$ for $s := \max\{f(b_f), f(q_{\min}) - \log N\}$. By the mean value Theorem, there exists $\xi \in (s, f(q_{\min}))$ such that $g(s) = g(f(q_{\min})) - g'(\xi)(f(q_{\min}) - s) \ge q_{\min} - g'(\xi)\log N$. Dividing $\rho^L$ on each side of $q_{\min} - g'(\xi)\log N \le g(\log\frac{1}{\mathcal{L}}) \le q_{\min}$ proves (b).

Now we prove (c). Without loss of generality, we assume $\log\frac{1}{\mathcal{L}(\boldsymbol{\theta}_m)} > f(b_f)$ for all $\boldsymbol{\theta}_m$. It follows from (b) that for every $\boldsymbol{\theta}_m$ there exists $\xi_m \in (f(q_{\min}(\boldsymbol{\theta}_m)) - \log N, f(q_{\min}(\boldsymbol{\theta}_m))) \cap (b_f, +\infty)$ such that

$$\bar{\gamma}(\boldsymbol{\theta}_m) - (g'(\xi_m)\log N)/\rho(\boldsymbol{\theta}_m)^L \le \tilde{\gamma}(\boldsymbol{\theta}_m) \le \bar{\gamma}(\boldsymbol{\theta}_m). \tag{8}$$

---

[4]Indeed, (B3.4) is not needed for showing Lemma A.5 and Theorem A.7.

Note that $\xi_m \geq f(q_{\min}(\boldsymbol{\theta}_m)) - \log N \geq \log \frac{1}{\mathcal{L}(\boldsymbol{\theta}_m)} - \log N \to +\infty$. So $\frac{g(\xi_m)}{g'(\xi_m)} = f'(g(\xi_m))g(\xi_m) \to +\infty$ by (B3.3). Also note that there exists a constant $B_0$ such that $\bar{\gamma}(\boldsymbol{\theta}_m) \leq B_0$ for all $m$ since $\bar{\gamma}$ is continuous on the unit sphere $\mathcal{S}^{d-1}$. So we have

$$\frac{g'(\xi_m)}{\rho(\boldsymbol{\theta}_m)^L} = \frac{g'(\xi_m)}{g(\xi_m)} \frac{g(\xi_m)}{\rho(\boldsymbol{\theta}_m)^L} \leq \frac{g'(\xi_m)}{g(\xi_m)} \frac{q_{\min}(\boldsymbol{\theta}_m)}{\rho(\boldsymbol{\theta}_m)^L} = \frac{g'(\xi_m)}{g(\xi_m)} \cdot \bar{\gamma}(\boldsymbol{\theta}_m) \leq \frac{g'(\xi_m)}{g(\xi_m)} \cdot B \to 0,$$

where the first inequality follows since $\xi_m \leq f(q_{\min}(\boldsymbol{\theta}_m))$. Together with (8), we have $|\tilde{\gamma}(\boldsymbol{\theta}_m) - \bar{\gamma}(\boldsymbol{\theta}_m)| \to 0$. $\qquad\square$

**Remark A.6.** *For exponential loss, we have already shown in Section 4.2 that $\tilde{\gamma}(\boldsymbol{\theta})$ is an $O(\rho^{-L})$-additive approximation for $\bar{\gamma}(\boldsymbol{\theta})$. For logistic loss, it follows easily from $g'(q) = \Theta(1)$ and (b) of Lemma A.5 that $\tilde{\gamma}(\boldsymbol{\theta})$ is an $O(\rho^{-L})$-additive approximation for $\bar{\gamma}(\boldsymbol{\theta})$ if $\mathcal{L}$ is sufficiently small.*

## A.3 THEOREMS

Now we state our main theorems. For the monotonicity of the normalized margin, we have the following theorem. The proof is provided in Appendix B.

**Theorem A.7.** *Under assumptions (A1), (A2), (B3)[4], (B4), the following statements are true for gradient flow:*

1. *For a.e. $t > t_0$, $\frac{d}{dt}\tilde{\gamma}(\boldsymbol{\theta}(t)) \geq 0$;*

2. *For a.e. $t > t_0$, either $\frac{d}{dt}\tilde{\gamma}(\boldsymbol{\theta}(t)) > 0$ or $\frac{d}{dt}\frac{\boldsymbol{\theta}(t)}{\|\boldsymbol{\theta}(t)\|_2} = 0$;*

3. *$\mathcal{L}(\boldsymbol{\theta}(t)) \to 0$ and $\|\boldsymbol{\theta}(t)\|_2 \to \infty$ as $t \to +\infty$; therefore, $|\bar{\gamma}(\boldsymbol{\theta}(t)) - \tilde{\gamma}(\boldsymbol{\theta}(t))| \to 0$.*

For the normalized margin at convergence, we have two theorems, one for infinite-time limiting case, and the other being a finite-time quantitative result. Their proofs can be found in Appendix C. As in the exponential loss case, we define the constrained optimization problem (P) as follows:

$$\min \quad \frac{1}{2}\|\boldsymbol{\theta}\|_2^2$$
$$\text{s.t.} \quad q_n(\boldsymbol{\theta}) \geq 1 \qquad \forall n \in [N]$$

First, we show the directional convergence of $\boldsymbol{\theta}(t)$ to a KKT point of (P).

**Theorem A.8.** *Consider gradient flow under assumptions (A1), (A2), (B3), (B4). For every limit point $\bar{\boldsymbol{\theta}}$ of $\left\{ \hat{\boldsymbol{\theta}}(t) : t \geq 0 \right\}$, $\bar{\boldsymbol{\theta}}/q_{\min}(\bar{\boldsymbol{\theta}})^{1/L}$ is a KKT point of (P).*

Second, we show that after finite time, gradient flow can pass through an approximate KKT point.

**Theorem A.9.** *Consider gradient flow under assumptions (A1), (A2), (B3), (B4). For any $\epsilon, \delta > 0$, there exists $r := \Theta(\log \delta^{-1})$ and $\Delta := \Theta(\epsilon^{-2})$ such that $\boldsymbol{\theta}/q_{\min}(\boldsymbol{\theta})^{1/L}$ is an $(\epsilon, \delta)$-KKT point at some time $t_*$ satisfying $\log \|\boldsymbol{\theta}(t_*)\|_2 \in (r, r + \Delta)$.*

For the definitions for KKT points and approximate KKT points, we refer the readers to Appendix C.1 for more details.

With a refined analysis, we can also provide tight rates for loss convergence and weight growth. The proof is given in Appendix D.

**Theorem A.10.** *Under assumptions (A1), (A2), (B3), (B4), we have the following tight rates for loss convergence and weight growth:*

$$\mathcal{L}(\boldsymbol{\theta}(t)) = \Theta\left(\frac{g(\log t)^{2/L}}{t(\log t)^2}\right) \quad \text{and} \quad \|\boldsymbol{\theta}(t)\|_2 = \Theta(g(\log t)^{1/L}).$$

Applying Theorem A.10 to exponential loss and logistic loss, in which $g(x) = \Theta(x)$, we have the following corollary:

**Corollary A.11.** *If $\ell(\cdot)$ is the exponential or logistic loss, then,*

$$\mathcal{L}(\boldsymbol{\theta}(t)) = \Theta\left(\frac{1}{t(\log t)^{2-2/L}}\right) \quad \text{and} \quad \|\boldsymbol{\theta}(t)\|_2 = \Theta((\log t)^{1/L}).$$

## B    MARGIN MONOTONICITY FOR GENERAL LOSS

In this section, we consider gradient flow and prove Theorem A.7. We assume (A1), (A2), (B3), (B4) as mentioned in Appendix A.

We follow the notations in Section 5 to define $\rho := \|\boldsymbol{\theta}\|_2$ and $\hat{\boldsymbol{\theta}} := \frac{\boldsymbol{\theta}}{\|\boldsymbol{\theta}\|_2} \in \mathcal{S}^{d-1}$, and sometimes we view the functions of $\boldsymbol{\theta}$ as functions of $t$.

### B.1    PROOF FOR PROPOSITION 1 AND 2

To prove the first two propositions, we generalize our key lemma (Lemma 5.1) to general loss.

**Lemma B.1.** *For $\tilde{\gamma}$ defined in Definition A.3, the following holds for all $t > t_0$,*

$$\frac{d}{dt} \log \rho > 0 \quad and \quad \frac{d}{dt} \log \tilde{\gamma} \geq L \left( \frac{d}{dt} \log \rho \right)^{-1} \left\| \frac{d\hat{\boldsymbol{\theta}}}{dt} \right\|_2^2. \tag{9}$$

Before proving Lemma B.1, we review two important properties of homogeneous functions. Note that these two properties are usually shown for smooth functions. By considering Clarke's subdifferential, we can generalize it to locally Lipschitz functions that admit chain rules:

**Theorem B.2.** *Let $F : \mathbb{R}^d \to \mathbb{R}$ be a locally Lipschitz function that admits a chain rule. If $F$ is $k$-homogeneous, then*

*(a) For all $\boldsymbol{x} \in \mathbb{R}^d$ and $\alpha > 0$,*
$$\partial^\circ F(\alpha x) = \alpha^{k-1} \partial^\circ F(x)$$
*That is, $\partial^\circ F(\alpha x) = \{\alpha^{k-1} \boldsymbol{h} : \boldsymbol{h} \in \partial^\circ F(x)\}$.*

*(b) (Euler's Theorem for Homogeneous Functions). For all $\boldsymbol{x} \in \mathbb{R}^d$,*
$$\langle \boldsymbol{x}, \partial^\circ F(\boldsymbol{x}) \rangle = k \cdot F(\boldsymbol{x})$$
*That is, $\langle \boldsymbol{x}, \boldsymbol{h} \rangle = k \cdot F(\boldsymbol{x})$ for all $\boldsymbol{h} \in \partial^\circ F(\boldsymbol{x})$.*

*Proof.* Let $D$ be the set of points $\boldsymbol{x}$ such that $F$ is differentiable at $\boldsymbol{x}$. According to the definition of Clarke's subdifferential, for proving *(a)*, it is sufficient to show that

$$\{ \lim_{k \to \infty} \nabla F(\alpha \boldsymbol{x}_k) : \boldsymbol{x}_k \to \boldsymbol{x}, \alpha \boldsymbol{x}_k \in D \} = \{\alpha^{k-1} \lim_{k \to \infty} \nabla F(\boldsymbol{x}_k) : \boldsymbol{x}_k \to \boldsymbol{x}, \boldsymbol{x}_k \in D\} \tag{10}$$

Fix $\boldsymbol{x}_k \in D$. Let $U$ be a neighborhood of $\boldsymbol{x}_k$. By definition of homogeneity, for any $\boldsymbol{h} \in \mathbb{R}^d$ and any $\boldsymbol{y} \in U \setminus \{\boldsymbol{x}_k\}$,

$$\frac{F(\alpha \boldsymbol{y}) - F(\alpha \boldsymbol{x}_k) - \langle \alpha \boldsymbol{y} - \alpha \boldsymbol{x}_k, \alpha^{k-1} \boldsymbol{h} \rangle}{\|\alpha \boldsymbol{y} - \alpha \boldsymbol{x}_k\|_2} = \alpha^{k-1} \cdot \frac{F(\boldsymbol{y}) - F(\boldsymbol{x}_k) - \langle \boldsymbol{y} - \boldsymbol{x}_k, \boldsymbol{h} \rangle}{\|\boldsymbol{y} - \boldsymbol{x}_k\|_2}.$$

Taking limits $\boldsymbol{y} \to \boldsymbol{x}_k$ on both sides, we know that the LHS converges to $0$ iff the RHS converges to $0$. Then by definition of differetiability and gradient, $F$ is differentiable at $\alpha \boldsymbol{x}_k$ iff it is differentiable at $\boldsymbol{x}_k$, and $\nabla F(\alpha \boldsymbol{x}_k) = \alpha^{k-1} \boldsymbol{h}$ iff $\nabla F(\boldsymbol{x}_k) = \boldsymbol{h}$. This proves (10).

To prove *(b)*, we fix $\boldsymbol{x} \in \mathbb{R}^d$. Let $\boldsymbol{z} : [0, +\infty) \to \mathbb{R}^d, \alpha \mapsto \alpha \boldsymbol{x}$ be an arc. By definition of homogeneity, $(F \circ \boldsymbol{z})(\alpha) = \alpha^k F(\boldsymbol{x})$ for $\alpha > 0$. Taking derivative with respect to $\alpha$ on both sides (for differentiable points), we have

$$\forall \boldsymbol{h} \in \partial^\circ F(\alpha \boldsymbol{x}) : \langle \boldsymbol{x}, \boldsymbol{h} \rangle = k\alpha^{k-1} F(\boldsymbol{x}) \tag{11}$$

holds for a.e. $\alpha > 0$. Pick an arbitrary $\alpha > 0$ making (11) hold. Then by *(a)*, (11) is equivalent to $\forall \boldsymbol{h} \in \partial^\circ F(\boldsymbol{x}) : \langle \boldsymbol{x}, \alpha^{k-1} \boldsymbol{h} \rangle = k\alpha^{k-1} F(\boldsymbol{x})$, which proves *(b)*.    $\square$

Applying Theorem B.2 to homogeneous neural networks, we have the following corollary:

**Corollary B.3.** *Under the assumptions (A1) and (A2), for any $\boldsymbol{\theta} \in \mathbb{R}^d$ and $\boldsymbol{x} \in \mathbb{R}^{d_\times}$,*

$$\langle \boldsymbol{\theta}, \partial^\circ \Phi_{\boldsymbol{x}}(\boldsymbol{\theta}) \rangle = L \cdot \Phi_{\boldsymbol{x}}(\boldsymbol{\theta}),$$

*where $\Phi_{\boldsymbol{x}}(\boldsymbol{\theta}) = \Phi(\boldsymbol{\theta}; \boldsymbol{x})$ is the network output for a fixed input $\boldsymbol{x}$.*

Corollary B.3 can be used to derive an exact formula for the weight growth during training.

**Theorem B.4.** *For a.e. $t \geq 0$,*

$$\frac{1}{2}\frac{d\rho^2}{dt} = L\sum_{n=1}^{N} e^{-f(q_n)}f'(q_n)q_n.$$

*Proof.* The proof idea is to use Corollary B.3 and chain rules (See Appendix I for chain rules in Clarke's sense). Applying the chain rule on $t \mapsto \rho^2 = \|\boldsymbol{\theta}\|_2^2$ yields $\frac{1}{2}\frac{d\rho^2}{dt} = -\langle\boldsymbol{\theta}, \boldsymbol{h}\rangle$ for all $\boldsymbol{h} \in \partial^\circ\mathcal{L}$ and a.e. $t > 0$. Then applying the chain rule on $\boldsymbol{\theta} \mapsto \mathcal{L}$, we have

$$-\partial^\circ\mathcal{L} \subseteq \sum_{n=1}^{N} e^{-f(q_n)}f'(q_n)\partial^\circ q_n = \left\{\sum_{n=1}^{N} e^{-f(q_n)}f'(q_n)\boldsymbol{h}_n : \boldsymbol{h}_n \in \partial^\circ q_n\right\}.$$

By Corollary B.3, $\langle\boldsymbol{\theta}, \boldsymbol{h}_n\rangle = Lq_n$, and thus $\frac{1}{2}\frac{d\rho^2}{dt} = L\sum_{n=1}^{N} e^{-f(q_n)}f'(q_n)q_n$. $\square$

For convenience, we define $\nu(t) := \sum_{n=1}^{N} e^{-f(q_n)}f'(q_n)q_n$ for all $t \geq 0$. Then Theorem B.4 can be rephrased as $\frac{1}{2}\frac{d\rho^2}{dt} = L\nu(t)$ for a.e. $t \geq 0$.

**Lemma B.5.** *For all $t > t_0$,*

$$\nu(t) \geq \frac{g(\log\frac{1}{\mathcal{L}})}{g'(\log\frac{1}{\mathcal{L}})}\mathcal{L}.$$

*Proof.* By Lemma A.5, $q_n \geq g(\log\frac{1}{\mathcal{L}})$ for all $n \in [N]$. Then by Assumption (B3),

$$f'(q_n)q_n \geq f'(g(\log\frac{1}{\mathcal{L}})) \cdot g(\log\frac{1}{\mathcal{L}}) = \frac{g(\log\frac{1}{\mathcal{L}})}{g'(\log\frac{1}{\mathcal{L}})}.$$

Combining this with the definitions of $\nu(t)$ and $\mathcal{L}$ gives

$$\nu(t) = \sum_{n=1}^{N} e^{-f(q_n)}f'(q_n)q_n \geq \sum_{n=1}^{N} e^{-f(q_n)}\frac{g(\log\frac{1}{\mathcal{L}})}{g'(\log\frac{1}{\mathcal{L}})} = \frac{g(\log\frac{1}{\mathcal{L}})}{g'(\log\frac{1}{\mathcal{L}})}\mathcal{L}.$$

$\square$

*Proof for Lemma B.1.* Note that $\frac{d}{dt}\log\rho = \frac{1}{2\rho^2}\frac{d\rho^2}{dt} = L\frac{\nu(t)}{\rho^2}$ by Theorem B.4. Then it simply follows from Lemma B.5 that $\frac{d}{dt}\log\rho > 0$ for a.e. $t > t_0$. For the second inequality, we first prove that $\log\tilde{\gamma} = \log(\ell^{-1}(\mathcal{L})/\rho^L) = \log(g(\log\frac{1}{\mathcal{L}})/\rho^L)$ exists for all $t \geq t_0$. $\mathcal{L}(t)$ is non-increasing with $t$. So $\mathcal{L}(t) < e^{-f(b_f)}$ for all $t \geq t_0$. This implies that (1) $\log\frac{1}{\mathcal{L}}$ is always in the domain of $g$; (2) $\rho > 0$ (otherwise $\mathcal{L} \geq Ne^{-f(0)} > e^{-f(b_f)}$, contradicting (B4)). Therefore, $\tilde{\gamma} := g(\log\frac{1}{\mathcal{L}})/\rho^L$ exists and is always positive for all $t \geq t_0$, which proves the existence of $\log\tilde{\gamma}$.

By the chain rule and Lemma B.5, we have

$$\frac{d}{dt}\log\tilde{\gamma} = \frac{d}{dt}\left(\log\left(g(\log\frac{1}{\mathcal{L}})\right) - L\log\rho\right) = \frac{g'(\log\frac{1}{\mathcal{L}})}{g(\log\frac{1}{\mathcal{L}})} \cdot \frac{1}{\mathcal{L}} \cdot \left(-\frac{d\mathcal{L}}{dt}\right) - L^2 \cdot \frac{\nu(t)}{\rho^2}$$

$$\geq \frac{1}{\nu(t)} \cdot \left(-\frac{d\mathcal{L}}{dt}\right) - L^2 \cdot \frac{\nu(t)}{\rho^2}$$

$$\geq \frac{1}{\nu(t)} \cdot \left(-\frac{d\mathcal{L}}{dt} - \frac{L^2\nu(t)^2}{\rho^2}\right).$$

On the one hand, $-\frac{d\mathcal{L}}{dt} = \left\|\frac{d\boldsymbol{\theta}}{dt}\right\|_2^2$ for a.e. $t > 0$ by Lemma I.3; on the other hand, $L\nu(t) = \langle\boldsymbol{\theta}, \frac{d\boldsymbol{\theta}}{dt}\rangle$ by Theorem B.4. Combining these together yields

$$\frac{d}{dt}\log\tilde{\gamma} \geq \frac{1}{\nu(t)}\left(\left\|\frac{d\boldsymbol{\theta}}{dt}\right\|_2^2 - \left\langle\hat{\boldsymbol{\theta}}, \frac{d\boldsymbol{\theta}}{dt}\right\rangle^2\right) = \frac{1}{\nu(t)}\left\|(\boldsymbol{I} - \hat{\boldsymbol{\theta}}\hat{\boldsymbol{\theta}}^\top)\frac{d\boldsymbol{\theta}}{dt}\right\|_2^2.$$

By the chain rule, $\frac{d\hat{\boldsymbol{\theta}}}{dt} = \frac{1}{\rho}(\boldsymbol{I} - \hat{\boldsymbol{\theta}}\hat{\boldsymbol{\theta}}^\top)\frac{d\boldsymbol{\theta}}{dt}$ for a.e. $t > 0$. So we have

$$\frac{d}{dt}\log\tilde{\gamma} \geq \frac{\rho^2}{\nu(t)}\left\|\frac{d\hat{\boldsymbol{\theta}}}{dt}\right\|_2^2 = L\left(\frac{d}{dt}\log\rho\right)^{-1}\left\|\frac{d\hat{\boldsymbol{\theta}}}{dt}\right\|_2^2.$$

□

### B.2 PROOF FOR PROPOSITION 3

To prove the third proposition, we prove the following lemma to show that $\mathcal{L} \to 0$ by giving an upper bound for $\mathcal{L}$. Since $\mathcal{L}$ can never be 0 for bounded $\rho$, $\mathcal{L} \to 0$ directly implies $\rho \to +\infty$. For showing $|\bar{\gamma} - \tilde{\gamma}| \to 0$, we only need to apply *(c)* in Lemma A.5, which shows this when $\mathcal{L} \to 0$.

**Lemma B.6.** *For all $t > t_0$,*

$$G(1/\mathcal{L}(t)) \geq L^2\tilde{\gamma}(t_0)^{2/L}(t - t_0) \qquad \text{for } G(x) := \int_{1/\mathcal{L}(t_0)}^x \frac{g'(\log u)^2}{g(\log u)^{2-2/L}}du.$$

*Therefore, $\mathcal{L}(t) \to 0$ and $\rho(t) \to +\infty$ as $t \to \infty$.*

*Proof for Lemma B.6.* By Lemma I.3 and Theorem B.4,

$$-\frac{d\mathcal{L}}{dt} = \left\|\frac{d\boldsymbol{\theta}}{dt}\right\|_2^2 \geq \left\langle\hat{\boldsymbol{\theta}}, \frac{d\boldsymbol{\theta}}{dt}\right\rangle^2 = L^2 \cdot \frac{\nu(t)^2}{\rho^2}.$$

Using Lemma B.5 to lower bound $\nu$ and replacing $\rho$ with $\left(g(\log\frac{1}{\mathcal{L}})/\tilde{\gamma}\right)^{1/L}$ by the definition of $\tilde{\gamma}$, we have

$$-\frac{d\mathcal{L}}{dt} \geq L^2 \cdot \left(\frac{g(\log\frac{1}{\mathcal{L}})}{g'(\log\frac{1}{\mathcal{L}})}\mathcal{L}\right)^2 \cdot \left(\frac{\tilde{\gamma}(t)}{g(\log\frac{1}{\mathcal{L}})}\right)^{2/L} \geq L^2\tilde{\gamma}(t_0)^{2/L} \cdot \frac{g(\log\frac{1}{\mathcal{L}})^{2-2/L}}{g'(\log\frac{1}{\mathcal{L}})^2}\mathcal{L},$$

where the last inequality uses the monotonicity of $\tilde{\gamma}$. So the following holds for a.e. $t \geq t_0$,

$$\frac{g'(\log\frac{1}{\mathcal{L}})^2}{g(\log\frac{1}{\mathcal{L}})^{2-2/L}} \cdot \frac{d}{dt}\frac{1}{\mathcal{L}} \geq L^2\tilde{\gamma}(t_0)^{2/L}.$$

Integrating on both sides from $t_0$ to $t$, we can conclude that

$$G(1/\mathcal{L}) \geq L^2\tilde{\gamma}(t_0)^{2/L}(t - t_0).$$

Note that $1/\mathcal{L}$ is non-decreasing. If $1/\mathcal{L}$ does not grow to $+\infty$, then neither does $G(1/\mathcal{L})$. But the RHS grows to $+\infty$, which leads to a contradiction. So $\mathcal{L} \to 0$.

To make $\mathcal{L} \to 0$, $q_{\min}$ must converge to $+\infty$. So $\rho \to +\infty$. □

## C CONVERGENCE TO THE MAX-MARGIN SOLUTION

In this section, we analyze the convergent direction of $\boldsymbol{\theta}$ and prove Theorem A.8 and A.9, assuming (A1), (A2), (B3), (B4) as mentioned in Section A.

We follow the notations in Section 5 to define $\rho := \|\boldsymbol{\theta}\|_2$ and $\hat{\boldsymbol{\theta}} := \frac{\boldsymbol{\theta}}{\|\boldsymbol{\theta}\|_2} \in \mathcal{S}^{d-1}$, and sometimes we view the functions of $\boldsymbol{\theta}$ as functions of $t$.

### C.1 PRELIMINARIES FOR KKT CONDITIONS

We first review the definition of Karush-Kuhn-Tucker (KKT) conditions for non-smooth optimization problems following from (Dutta et al., 2013).

Consider the following optimization problem (P) for $\boldsymbol{x} \in \mathbb{R}^d$:

$$\begin{aligned}
\min \quad &f(\boldsymbol{x}) \\
\text{s.t.} \quad &g_n(\boldsymbol{x}) \leq 0 \qquad \forall n \in [N]
\end{aligned}$$

where $f, g_1, \ldots, g_n : \mathbb{R}^d \to \mathbb{R}$ are locally Lipschitz functions. We say that $\boldsymbol{x} \in \mathbb{R}^d$ is a feasible point of (P) if $\boldsymbol{x}$ satisfies $g_n(\boldsymbol{x}) \leq 0$ for all $n \in [N]$.

**Definition C.1** (KKT Point). A feasible point $\boldsymbol{x}$ of (P) is a KKT point if $\boldsymbol{x}$ satisfies KKT conditions: there exists $\lambda_1, \ldots, \lambda_N \geq 0$ such that

1. $0 \in \partial^\circ f(\boldsymbol{x}) + \sum_{n \in [N]} \lambda_n \partial^\circ g_n(\boldsymbol{x})$;

2. $\forall n \in [N] : \lambda_n g_n(\boldsymbol{x}) = 0$.

It is important to note that a global minimum of (P) may not be a KKT point, but under some regularity assumptions, the KKT conditions become a necessary condition for global optimality. The regularity condition we shall use in this paper is the non-smooth version of Mangasarian-Fromovitz Constraint Qualification (MFCQ) (see, e.g., the constraint qualification (C.Q.5) in (Giorgi et al., 2004)):

**Definition C.2** (MFCQ). For a feasible point $\boldsymbol{x}$ of (P), (P) is said to satisfy MFCQ at $\boldsymbol{x}$ if there exists $\boldsymbol{v} \in \mathbb{R}^d$ such that for all $n \in [N]$ with $g_n(\boldsymbol{x}) = 0$,

$$\forall \boldsymbol{h} \in \partial^\circ g_n(\boldsymbol{x}) : \langle \boldsymbol{h}, \boldsymbol{v} \rangle > 0.$$

Following from (Dutta et al., 2013), we define an approximate version of KKT point, as shown below. Note that this definition is essentially the modified $\epsilon$-KKT point defined in their paper, but these two definitions differ in the following two ways: (1) First, in their paper, the subdifferential is allowed to be evaluated in a neighborhood of $\boldsymbol{x}$, so our definition is slightly stronger; (2) Second, their paper fixes $\delta = \epsilon^2$, but in our definition we make them independent.

**Definition C.3** (Approximate KKT Point). For $\epsilon, \delta > 0$, a feasible point $\boldsymbol{x}$ of (P) is an $(\epsilon, \delta)$-KKT point if there exists $\lambda_n \geq 0, \boldsymbol{k} \in \partial^\circ f(\boldsymbol{x}), \boldsymbol{h}_n \in \partial^\circ g_n(\boldsymbol{x})$ for all $n \in [N]$ such that

1. $\left\| \boldsymbol{k} + \sum_{n \in [N]} \lambda_n \boldsymbol{h}_n(\boldsymbol{x}) \right\|_2 \leq \epsilon$;

2. $\forall n \in [N] : \lambda_n g_n(\boldsymbol{x}) \geq -\delta$.

As shown in (Dutta et al., 2013), $(\epsilon, \delta)$-KKT point is an approximate version of KKT point in the sense that a series of $(\epsilon, \delta)$-KKT points can converge to a KKT point. We restate their theorem in our setting:

**Theorem C.4** (Corollary of Theorem 3.6 in (Dutta et al., 2013)). *Let $\{\boldsymbol{x}_k \in \mathbb{R}^d : k \in \mathbb{N}\}$ be a sequence of feasible points of (P), $\{\epsilon_k > 0 : k \in \mathbb{N}\}$ and $\{\delta_k > 0 : k \in \mathbb{N}\}$ be two sequences. $\boldsymbol{x}_k$ is an $(\epsilon_k, \delta_k)$-KKT point for every $k$, and $\epsilon_k \to 0, \delta_k \to 0$. If $\boldsymbol{x}_k \to \boldsymbol{x}$ as $k \to +\infty$ and MFCQ holds at $\boldsymbol{x}$, then $\boldsymbol{x}$ is a KKT point of (P).*

## C.2 KKT CONDITIONS FOR (P)

Recall that for a homogeneous neural network, the optimization problem (P) is defined as follows:

$$\begin{aligned} \min \quad & \frac{1}{2} \|\boldsymbol{\theta}\|_2^2 \\ \text{s.t.} \quad & q_n(\boldsymbol{\theta}) \geq 1 \qquad \forall n \in [N] \end{aligned}$$

Using the terminologies and notations in Appendix C.1, the objective and constraints are $f(\boldsymbol{x}) = \frac{1}{2}\|\boldsymbol{x}\|_2^2$ and $g_n(\boldsymbol{x}) = 1 - q_n(\boldsymbol{x})$. The KKT points and approximate KKT points for (P) are defined as follows:

**Definition C.5** (KKT Point of (P)). A feasible point $\boldsymbol{\theta}$ of (P) is a KKT point if there exist $\lambda_1, \ldots, \lambda_N \geq 0$ such that

1. $\boldsymbol{\theta} - \sum_{n=1}^N \lambda_n \boldsymbol{h}_n = \boldsymbol{0}$ for some $\boldsymbol{h}_1, \ldots, \boldsymbol{h}_N$ satisfying $\boldsymbol{h}_n \in \partial^\circ q_n(\boldsymbol{\theta})$;

2. $\forall n \in [N] : \lambda_n(q_n(\boldsymbol{\theta}) - 1) = 0$.

**Definition C.6** (Approximate KKT Point of (P)). A feasible point $\boldsymbol{\theta}$ of (P) is an $(\epsilon, \delta)$-KKT point of (P) if there exists $\lambda_1, \ldots, \lambda_N \geq 0$ such that

1. $\left\| \boldsymbol{\theta} - \sum_{n=1}^N \lambda_n \boldsymbol{h}_n \right\|_2 \leq \epsilon$ for some $\boldsymbol{h}_1, \ldots, \boldsymbol{h}_N$ satisfying $\boldsymbol{h}_n \in \partial^\circ q_n(\boldsymbol{\theta})$;

    2. $\forall n \in [N] : \lambda_n(q_n(\boldsymbol{\theta}) - 1) \leq \delta$.

By the homogeneity of $q_n$, it is easy to see that (P) satisfies MFCQ, and thus KKT conditions are first-order necessary condition for global optimality.

**Lemma C.7.** *(P) satisfies MFCQ at every feasible point $\boldsymbol{\theta}$.*

*Proof.* Take $\boldsymbol{v} := \boldsymbol{\theta}$. For all $n \in [N]$ satisfying $q_n = 1$, by homogeneity of $q_n$,

$$\langle \boldsymbol{v}, \boldsymbol{h} \rangle = Lq_n(\boldsymbol{\theta}) = L > 0$$

holds for any $\boldsymbol{h} \in -\partial^\circ q_n(\boldsymbol{\theta}) = \partial^\circ(1 - q_n(\boldsymbol{\theta}))$. $\qquad\square$

## C.3 KEY LEMMAS

Define $\beta(t) := \frac{1}{\left\| \frac{d\boldsymbol{\theta}}{dt} \right\|_2} \left\langle \hat{\boldsymbol{\theta}}, \frac{d\boldsymbol{\theta}}{dt} \right\rangle$ to be the cosine of the angle between $\boldsymbol{\theta}$ and $\frac{d\boldsymbol{\theta}}{dt}$. Here $\beta(t)$ is only defined for a.e. $t > 0$. Since $q_n$ is locally Lipschitz, it can be shown that $q_n$ is (globally) Lipschitz on the compact set $\mathcal{S}^{d-1}$, which is the unit sphere in $\mathbb{R}^d$. Define

$$B_0 := \sup \left\{ \frac{q_n}{\rho^L} : \boldsymbol{\theta} \in \mathbb{R}^d \setminus \{\boldsymbol{0}\} \right\}$$
$$= \sup \left\{ q_n : \boldsymbol{\theta} \in \mathcal{S}^{d-1} \right\} < \infty.$$
$$B_1 := \sup \left\{ \frac{\|\boldsymbol{h}\|_2}{\rho^{L-1}} : \boldsymbol{\theta} \in \mathbb{R}^d \setminus \{\boldsymbol{0}\}, \boldsymbol{h} \in \partial^\circ q_n, n \in [N] \right\}$$
$$= \sup \left\{ \|\boldsymbol{h}\|_2 : \boldsymbol{\theta} \in \mathcal{S}^{d-1}, \boldsymbol{h} \in \partial^\circ q_n, n \in [N] \right\} < \infty.$$

For showing Theorem A.8 and Theorem A.9, we first prove Lemma C.8. In light of this lemma, if we aim to show that $\boldsymbol{\theta}$ is along the direction of an approximate KKT point, we only need to show $\beta \to 1$ (which makes $\epsilon \to 0$) and $\mathcal{L} \to 0$ (which makes $\delta \to 0$).

**Lemma C.8.** *Let $C_1, C_2$ be two constants defined as*

$$C_1 := \frac{\sqrt{2}}{\tilde{\gamma}(t_0)^{1/L}}, \qquad C_2 := \frac{2eNK^2}{L\tilde{\gamma}(t_0)^{2/L}} \left( \frac{B_1}{\tilde{\gamma}(t_0)} \right)^{\log_2 K},$$

*where $K, b_g$ are constants specified in (B3.4). If $\log \frac{1}{\mathcal{L}} \geq b_g$ at time $t > t_0$, then $\tilde{\boldsymbol{\theta}} := \boldsymbol{\theta}/q_{\min}(\boldsymbol{\theta})^{1/L}$ is an $(\epsilon, \delta)$-KKT point of (P), where $\epsilon := C_1\sqrt{1 - \beta}$ and $\delta := C_2/(\log \frac{1}{\mathcal{L}})$.*

*Proof.* Let $\boldsymbol{h}(t) := \frac{d\boldsymbol{\theta}}{dt}(t)$ for a.e. $t > 0$. By the chain rule, there exist $\boldsymbol{h}_1, \ldots, \boldsymbol{h}_N$ such that $\boldsymbol{h}_n \in \partial^\circ q_n$ and $\boldsymbol{h} = \sum_{n \in [N]} e^{-f(q_n)} f'(q_n)\boldsymbol{h}_n$. Let $\tilde{\boldsymbol{h}}_n := \boldsymbol{h}_n/q_{\min}^{1-1/L} \in \partial^\circ q_n(\tilde{\boldsymbol{\theta}})$ (Recall that $\tilde{\boldsymbol{\theta}} := \boldsymbol{\theta}/q_{\min}(\boldsymbol{\theta})^{1/L}$). Construct $\lambda_n := q_{\min}^{1-2/L} \rho \cdot e^{-f(q_n)} f'(q_n)/\|\boldsymbol{h}\|_2$. Now we only need to show

$$\left\| \tilde{\boldsymbol{\theta}} - \sum_{n=1}^N \lambda_n \tilde{\boldsymbol{h}}_n \right\|_2^2 \leq \frac{2}{\tilde{\gamma}^{2/L}}(1 - \beta) \tag{12}$$

$$\sum_{n=1}^N \lambda_n(q_n(\tilde{\boldsymbol{\theta}}) - 1) \leq \left( \frac{2eNK^2}{L\tilde{\gamma}^{2/L}} \left( \frac{B_1}{\tilde{\gamma}} \right)^{\log_2 K} \right) \frac{1}{f(\tilde{\gamma}\rho^L)}. \tag{13}$$

Then $\tilde{\boldsymbol{\theta}}$ can be shown to be an $(\epsilon, \delta)$-KKT point by the monotonicity $\tilde{\gamma}(t) \geq \tilde{\gamma}(t_0)$ for $t > t_0$.

**Proof of** (12). From our construction, $\sum_{n=1}^N \lambda_n \tilde{\boldsymbol{h}}_n = q_{\min}^{-1/L} \rho \boldsymbol{h}/\|\boldsymbol{h}\|_2$. So

$$\left\| \tilde{\boldsymbol{\theta}} - \sum_{n=1}^N \lambda_n \tilde{\boldsymbol{h}}_n \right\|_2^2 = q_{\min}^{-2/L} \rho^2 \left\| \hat{\boldsymbol{\theta}} - \frac{\boldsymbol{h}}{\|\boldsymbol{h}\|_2} \right\|_2^2 = q_{\min}^{-2/L} \rho^2(2 - 2\beta) \leq \frac{2}{\tilde{\gamma}^{2/L}}(1 - \beta),$$

where the last equality is by Lemma A.5.

**Proof for** (13). According to our construction,

$$\sum_{n=1}^{N} \lambda_n (q_n(\tilde{\boldsymbol{\theta}}) - 1) = \frac{q_{\min}^{-2/L} \rho}{\|\boldsymbol{h}\|_2} \sum_{n=1}^{N} e^{-f(q_n)} f'(q_n)(q_n - q_{\min}).$$

Note that $\|\boldsymbol{h}\|_2 \geq \langle \boldsymbol{h}, \hat{\boldsymbol{\theta}} \rangle = L\nu/\rho$. By Lemma B.5 and Lemma D.1, we have

$$\nu \geq \frac{g(\log \frac{1}{\mathcal{L}})}{g'(\log \frac{1}{\mathcal{L}})} \mathcal{L} \geq \frac{1}{2K} \log \frac{1}{\mathcal{L}} \cdot \mathcal{L} \geq \frac{1}{2K} f(\tilde{\gamma}\rho^L) e^{-f(q_{\min})},$$

where the last inequality uses $f(\tilde{\gamma}\rho^L) = \log \frac{1}{\mathcal{L}}$ and $\mathcal{L} \geq e^{-f(q_{\min})}$. Combining these gives

$$\sum_{n=1}^{N} \lambda_n (q_n(\tilde{\boldsymbol{\theta}}) - 1) \leq \frac{2K q_{\min}^{-2/L} \rho^2}{L f(\tilde{\gamma}\rho^L)} \sum_{n=1}^{N} e^{f(q_{\min}) - f(q_n)} f'(q_n)(q_n - q_{\min}).$$

If $q_n > q_{\min}$, then by the mean value theorem there exists $\xi_n \in (q_{\min}, q_n)$ such that $f(q_n) - f(q_{\min}) = f'(\xi_n)(q_n - q_{\min})$. By Assumption (B3.4), we know that $f'(q_n) \leq K^{\lceil \log_2(q_n/\xi_n) \rceil} f'(\xi_n)$. Note that $\lceil \log_2(q_n/\xi_n) \rceil \leq \log_2(2B_1\rho^L/q_{\min}) \leq \log_2(2B_1/\tilde{\gamma})$. Then we have

$$\sum_{n=1}^{N} \lambda_n (q_n(\tilde{\boldsymbol{\theta}}) - 1) \leq \frac{2K q_{\min}^{-2/L} \rho^2}{L f(\tilde{\gamma}\rho^L)} K^{\log_2(2B_1/\tilde{\gamma})} \sum_{n: \, q_n \neq q_{\min}} e^{-f'(\xi_n)(q_n - q_{\min})} f'(\xi_n)(q_n - q_{\min})$$

$$\leq \frac{2K \tilde{\gamma}^{-2/L} \rho^2}{L f(\tilde{\gamma}\rho^L)} K^{\log_2(2B_1/\tilde{\gamma})} \cdot Ne$$

where the second inequality uses $q_{\min}^{-2/L} \rho^2 \leq \tilde{\gamma}^{-2/L}$ by Lemma A.5 and the fact that the function $x \mapsto e^{-x} x$ on $(0, +\infty)$ attains the maximum value $e$ at $x = 1$. ☐

By Theorem A.7, we have already known that $\mathcal{L} \to 0$. So it remains to bound $\beta(t)$. For this, we first prove the following lemma to bound the integral of $\beta(t)$.

**Lemma C.9.** *For all $t_2 > t_1 \geq t_0$,*

$$\int_{t_1}^{t_2} \left( \beta(\tau)^{-2} - 1 \right) \cdot \frac{d}{d\tau} \log \rho(\tau) \cdot d\tau \leq \frac{1}{L} \log \frac{\tilde{\gamma}(t_2)}{\tilde{\gamma}(t_1)}.$$

*Proof.* By Lemma B.4, $\frac{d}{dt} \log \rho = \frac{1}{2\rho^2} \frac{d\rho^2}{dt} = \frac{L\nu}{\rho^2}$ for a.e. $t > 0$. By Lemma B.1, for a.e. $t \in (t_1, t_2)$,

$$\frac{d}{dt} \log \tilde{\gamma} \geq L \cdot \left\| \frac{\rho^2}{L\nu} \frac{d\hat{\boldsymbol{\theta}}}{dt} \right\|_2^2 \cdot \frac{d}{dt} \log \rho. \tag{14}$$

By the chain rule, $\frac{d\hat{\boldsymbol{\theta}}}{dt} = \frac{1}{\rho}(\boldsymbol{I} - \hat{\boldsymbol{\theta}}\hat{\boldsymbol{\theta}}^\top) \frac{d\boldsymbol{\theta}}{dt}$. So we have

$$\left\| \frac{\rho^2}{L\nu} \frac{d\hat{\boldsymbol{\theta}}}{dt} \right\|_2^2 = \left\| \frac{\rho}{L\nu(t)}(\boldsymbol{I} - \hat{\boldsymbol{\theta}}\hat{\boldsymbol{\theta}}^\top) \frac{d\boldsymbol{\theta}}{dt} \right\|_2^2 = \frac{\left\| \frac{d\boldsymbol{\theta}}{dt} \right\|_2^2 - \left\langle \frac{d\boldsymbol{\theta}}{dt}, \hat{\boldsymbol{\theta}} \right\rangle^2}{\left\langle \frac{d\boldsymbol{\theta}}{dt}, \hat{\boldsymbol{\theta}} \right\rangle^2} = \beta^{-2} - 1. \tag{15}$$

where the last equality follows from the definition of $\beta$. Combining 14 and 15, we have

$$\frac{d}{dt} \log \tilde{\gamma} \geq L \left( \beta^{-2} - 1 \right) \cdot \frac{d}{dt} \log \rho.$$

Integrating on both sides from $t_1$ to $t_2$ proves the lemma. ☐

A direct corollary of Lemma C.9 is the upper bound for the minimum $\beta^2 - 1$ within a time interval:

**Corollary C.10.** *For all $t_2 > t_1 \geq t_0$, then there exists $t_* \in (t_1, t_2)$ such that*

$$\beta(t_*)^{-2} - 1 \leq \frac{1}{L} \cdot \frac{\log \tilde{\gamma}(t_2) - \log \tilde{\gamma}(t_1)}{\log \rho(t_2) - \log \rho(t_1)}.$$

*Proof.* Denote the RHS as $C$. Assume to the contrary that $\beta(\tau)^{-2} - 1 > C$ for a.e. $\tau \in (t_1, t_2)$. By Lemma B.1, $\log \rho(\tau) > 0$ for a.e. $\tau \in (t_1, t_2)$. Then by Lemma C.9, we have

$$\frac{1}{L} \log \frac{\tilde{\gamma}(t_2)}{\tilde{\gamma}(t_1)} > \int_{t_1}^{t_2} C \cdot \frac{d}{d\tau} \log \rho(\tau) \cdot d\tau = C \cdot (\log \rho(t_2) - \log \rho(t_1)) = \frac{1}{L} \log \frac{\tilde{\gamma}(t_2)}{\tilde{\gamma}(t_1)},$$

which leads to a contradiction. $\qquad\square$

In the rest of this section, we present both asymptotic and non-asymptotic analyses for the directional convergence by using Corollary C.10 to bound $\beta(t)$.

## C.4 ASYMPTOTIC ANALYSIS

We first prove an auxiliary lemma which gives an upper bound for the change of $\hat{\boldsymbol{\theta}}$.

**Lemma C.11.** *For a.e. $t > t_0$,*

$$\left\| \frac{d\hat{\boldsymbol{\theta}}}{dt} \right\|_2 \le \frac{B_1}{L\tilde{\gamma}} \cdot \frac{d}{dt} \log \rho.$$

*Proof.* Observe that $\left\| \frac{d\hat{\boldsymbol{\theta}}}{dt} \right\|_2 = \frac{1}{\rho} \left\| (\boldsymbol{I} - \hat{\boldsymbol{\theta}}\hat{\boldsymbol{\theta}}^\top) \frac{d\boldsymbol{\theta}}{dt} \right\|_2 \le \frac{1}{\rho} \left\| \frac{d\boldsymbol{\theta}}{dt} \right\|_2$. It is sufficient to bound $\left\| \frac{d\boldsymbol{\theta}}{dt} \right\|_2$. By the chain rule, there exists $\boldsymbol{h}_1, \dots, \boldsymbol{h}_N : [0, +\infty) \to \mathbb{R}^d$ satisfying that for a.e. $t > 0$, $\boldsymbol{h}_n(t) \in \partial^\circ q_n$ and $\frac{d\boldsymbol{\theta}}{dt} = \sum_{n \in [N]} e^{-f(q_n)} f'(q_n) \boldsymbol{h}_n(t)$. By definition of $B_1$, $\|\boldsymbol{h}_n\|_2 \le B_1 \rho^{L-1}$ for a.e. $t > 0$. So we have

$$\left\| \frac{d\boldsymbol{\theta}}{dt} \right\|_2 \le \sum_{n \in [N]} e^{-f(q_n)} f'(q_n) \|\boldsymbol{h}_n\|_2$$

$$\le \sum_{n \in [N]} e^{-f(q_n)} f'(q_n) q_n \cdot \frac{1}{q_n} \cdot B_1 \rho^{L-1}.$$

Note that every summand is positive. By Lemma A.5, $q_n$ is lower-bounded by $q_n \ge q_{\min} \ge g(\log \frac{1}{\mathcal{L}})$, so we can replace $q_n$ with $g(\log \frac{1}{\mathcal{L}})$ in the above inequality. Combining with the fact that $\sum_{n \in [N]} e^{-f(q_n)} f'(q_n) q_n$ is just $\nu$, we have

$$\left\| \frac{d\boldsymbol{\theta}}{dt} \right\|_2 \le \frac{\nu}{g(\log \frac{1}{\mathcal{L}})} \cdot B_1 \rho^{L-1} = \frac{B_1 \nu}{\tilde{\gamma}\rho}. \tag{16}$$

So we have $\left\| \frac{d\hat{\boldsymbol{\theta}}}{dt} \right\|_2 \le \frac{1}{\rho} \left\| \frac{d\boldsymbol{\theta}}{dt} \right\|_2 \le \frac{B_1}{L\tilde{\gamma}} \cdot \frac{L\nu}{\rho^2} = \frac{B_1}{L\tilde{\gamma}} \cdot \frac{d}{dt} \log \rho.$ $\qquad\square$

To prove Theorem A.8, we consider each limit point $\bar{\boldsymbol{\theta}}/q_{\min}(\bar{\boldsymbol{\theta}})^{1/L}$, and construct a series of approximate KKT points converging to it. Then $\bar{\boldsymbol{\theta}}/q_{\min}(\bar{\boldsymbol{\theta}})^{1/L}$ can be shown to be a KKT point by Theorem C.4. The following lemma ensures that such construction exists.

**Lemma C.12.** *For every limit point $\bar{\boldsymbol{\theta}}$ of $\left\{ \hat{\boldsymbol{\theta}}(t) : t \ge 0 \right\}$, there exists a sequence of $\{t_m : m \in \mathbb{N}\}$ such that $t_m \uparrow +\infty, \hat{\boldsymbol{\theta}}(t_m) \to \bar{\boldsymbol{\theta}}$, and $\beta(t_m) \to 1$.*

*Proof.* Let $\{\epsilon_m > 0 : m \in \mathbb{N}\}$ be an arbitrary sequence with $\epsilon_m \to 0$. Now we construct $\{t_m\}$ by induction. Suppose $t_1 < t_2 < \cdots < t_{m-1}$ have already been constructed. Since $\bar{\boldsymbol{\theta}}$ is a limit point and $\tilde{\gamma}(t) \uparrow \tilde{\gamma}_\infty$ (recall that $\tilde{\gamma}_\infty := \lim_{t \to +\infty} \tilde{\gamma}(t)$), there exists $s_m > t_{m-1}$ such that

$$\left\| \hat{\boldsymbol{\theta}}(s_m) - \bar{\boldsymbol{\theta}} \right\|_2 \le \epsilon_m \quad \text{and} \quad \frac{1}{L} \log \frac{\tilde{\gamma}_\infty}{\tilde{\gamma}(s_m)} \le \epsilon_m^3.$$

Let $s'_m > s_m$ be a time such that $\log \rho(s'_m) = \log \rho(s_m) + \epsilon_m$. According to Theorem A.7, $\log \rho \to +\infty$, so $s'_m$ must exist. We construct $t_m \in (s_m, s'_m)$ to be a time that $\beta(t_m)^{-2} - 1 \le \epsilon_m^2$, where the existence can be shown by Corollary C.10.

Now we show that this construction meets our requirement. It follows from $\beta(t_m)^{-2} - 1 \leq \epsilon_m^2$ that $\beta(t_m) \geq 1/\sqrt{1 + \epsilon_m^2} \to 1$. By Lemma C.11, we also know that

$$\left\|\hat{\boldsymbol{\theta}}(t_m) - \bar{\boldsymbol{\theta}}\right\|_2 \leq \left\|\hat{\boldsymbol{\theta}}(t_m) - \hat{\boldsymbol{\theta}}(s_m)\right\|_2 + \left\|\hat{\boldsymbol{\theta}}(s_m) - \bar{\boldsymbol{\theta}}\right\|_2 \leq \frac{B_1}{L\tilde{\gamma}(t_0)} \cdot \epsilon_m + \epsilon_m \to 0.$$

This completes the proof. $\qquad\square$

*Proof of Theorem A.8.* Let $\tilde{\boldsymbol{\theta}} := \bar{\boldsymbol{\theta}}/q_{\min}(\bar{\boldsymbol{\theta}})^{1/L}$ for short. Let $\{t_m : m \in \mathbb{N}\}$ be the sequence constructed as in Lemma C.12. For each $t_m$, define $\epsilon(t_m)$ and $\delta(t_m)$ as in Lemma C.8. Then we know that $\boldsymbol{\theta}(t_m)/q_{\min}(t_m)^{1/L}$ is an $(\epsilon(t_m), \delta(t_m))$-KKT point and $\boldsymbol{\theta}(t_m)/q_{\min}(t_m)^{1/L} \to \tilde{\boldsymbol{\theta}}, \epsilon(t_m) \to 0, \delta(t_m) \to 0$. By Lemma C.7, (P) satisfies MFCQ. Applying Theorem C.4 proves the theorem. $\quad\square$

## C.5 NON-ASYMPTOTIC ANALYSIS

*Proof of Theorem A.9.* Let $C_0 := \frac{1}{L} \log \frac{\tilde{\gamma}_\infty}{\tilde{\gamma}(t_0)}$. Without loss of generality, we assume $\epsilon < \frac{\sqrt{6}}{2} C_1$ and $\delta < C_2/f(b_f)$.

Let $t_1$ be the time such that $\log \rho(t_1) = \frac{1}{L} \log \frac{g(C_2 \delta^{-1})}{\tilde{\gamma}(t_0)} = \Theta(\log \delta^{-1})$ and $t_2$ be the time such that $\log \rho(t_2) - \log \rho(t_1) = \frac{1}{2} C_0 C_1^2 \epsilon^{-2} = \Theta(\epsilon^{-2})$. By Corollary C.10, there exists $t_* \in (t_1, t_2)$ such that $\beta(t_*)^{-2} - 1 \leq 2\epsilon^2 C_1^{-2}$.

Now we argue that $\tilde{\boldsymbol{\theta}}(t_*)$ is an $(\epsilon, \delta)$-KKT point. By Lemma C.8, we only need to show $C_1 \sqrt{1 - \beta(t_*)} \leq \epsilon$ and $C_2/f(\tilde{\gamma}(t_*)\rho(t_*)^L) \leq \delta$. For the first inequality, by assumption $\epsilon < \frac{\sqrt{6}}{2} C_1$, we know that $\beta(t_*)^{-2} - 1 < 3$, which implies $|\beta(t_*)| < \frac{1}{2}$. Then we have $1 - \beta(t_*) \leq 2\epsilon^2 C_1^{-2} \cdot \frac{\beta(t_*)^2}{1+\beta(t_*)} \leq \epsilon^2 C_1^{-2}$. Therefore, $C_1 \sqrt{1 - \beta(t_*)} \leq \epsilon$ holds. For the second inequality, $\tilde{\gamma}(t_*)\rho(t_*)^L \geq \tilde{\gamma}(t_*) \cdot \frac{g(C_2 \delta^{-1})}{\tilde{\gamma}(t_0)} \geq g(C_2 \delta^{-1})$. Therefore, $C_2/f(\tilde{\gamma}(t_*)\rho(t_*)^L) \leq \delta$ holds. $\qquad\square$

## C.6 PROOF FOR COROLLARY 4.5

By the homogeneity of $q_n$, we can characterize KKT points using kernel SVM.

**Lemma C.13.** *If $\boldsymbol{\theta}_*$ is KKT point of (P), then there exists $\boldsymbol{h}_n \in \partial^\circ \Phi_{\boldsymbol{x}_n}(\boldsymbol{\theta}_*)$ for $n \in [N]$ such that $\frac{1}{L}\boldsymbol{\theta}_*$ is an optimal solution for the following constrained optimization problem (Q):*

$$\min \quad \frac{1}{2} \|\boldsymbol{\theta}\|_2^2$$
$$s.t. \quad y_n \langle \boldsymbol{\theta}, \boldsymbol{h}_n \rangle \geq 1 \qquad \forall n \in [N]$$

*Proof.* It is easy to see that (Q) is a convex optimization problem. For $\boldsymbol{\theta} = \frac{2}{L}\boldsymbol{\theta}_*$, from Theorem B.2, we can see that $y_n \langle \boldsymbol{\theta}, \boldsymbol{h}_n \rangle = 2q_n(\boldsymbol{\theta}_*) \geq 2 > 1$, which implies Slater's condition. Thus, we only need to show that $\frac{1}{L}\boldsymbol{\theta}_*$ satisfies KKT conditions for (Q).

By the KKT conditions for (P), we can construct $\boldsymbol{h}_n \in \partial^\circ q_n(\boldsymbol{\theta}_*)$ for $n \in [N]$ such that $\boldsymbol{\theta}_* - \sum_{n=1}^N \lambda_n y_n \boldsymbol{h}_n = \boldsymbol{0}$ for some $\lambda_1, \ldots, \lambda_N \geq 0$ and $\lambda_n(q_n(\boldsymbol{\theta}_*) - 1) = 0$. Thus, $\frac{1}{L}\boldsymbol{\theta}_*$ satisfies

1. $\frac{1}{L}\boldsymbol{\theta}_* - \sum_{n=1}^N \frac{1}{L}\lambda_n y_n \boldsymbol{h}_n = \boldsymbol{0}$;

2. $\frac{1}{L}\lambda_n \left(y_n \left\langle \frac{1}{L}\boldsymbol{\theta}_*, \boldsymbol{h}_n \right\rangle - 1\right) = \frac{1}{L}\lambda_n \left(q_n(\boldsymbol{\theta}) - 1\right) \geq 0.$

So $\frac{1}{L}\boldsymbol{\theta}_*$ satisfies KKT conditions for (Q). $\qquad\square$

Now we prove Corollary 4.5 in Section 4.3.

*Proof.* By Theorem A.8, every limit point $\bar{\boldsymbol{\theta}}$ is along the direction of a KKT point of (P). Combining this with Lemma C.13, we know that every limit point $\bar{\boldsymbol{\theta}}$ is also along the max-margin direction of (Q).

For smooth models, $h_n$ in (Q) is exactly the gradient $\nabla \Phi_{\boldsymbol{x}_n}(\bar{\boldsymbol{\theta}})$. So, (Q) is the optimization problem for SVM with kernel $K_{\bar{\boldsymbol{\theta}}}(\boldsymbol{x}, \boldsymbol{x}') = \langle \nabla \Phi_{\boldsymbol{x}}(\bar{\boldsymbol{\theta}}), \nabla \Phi_{\boldsymbol{x}'}(\bar{\boldsymbol{\theta}}) \rangle$. For non-smooth models, we can construct an arbitrary function $\boldsymbol{h}(\boldsymbol{x}) \in \partial^\circ \Phi_{\boldsymbol{x}}(\bar{\boldsymbol{\theta}})$ that ensures $\boldsymbol{h}(\boldsymbol{x}_n) = \boldsymbol{h}_n$. Then, (Q) is the optimization problem for SVM with kernel $K_{\bar{\boldsymbol{\theta}}}(\boldsymbol{x}, \boldsymbol{x}') = \langle \boldsymbol{h}(\boldsymbol{x}), \boldsymbol{h}(\boldsymbol{x}') \rangle$. □

## D  TIGHT BOUNDS FOR LOSS CONVERGENCE AND WEIGHT GROWTH

In this section, we give proof for Theorem A.10, which gives tight bounds for loss convergence and weight growth under Assumption (A1), (A2), (B3), (B4).

### D.1  CONSEQUENCES OF (B3.4)

Before proving Theorem A.10, we show some consequences of (B3.4).

**Lemma D.1.** *For $f(\cdot)$ and $g(\cdot)$, we have*

1. *For all $x \in [b_g, +\infty)$, $\frac{g(x)}{g'(x)} \in [\frac{1}{2K}x, 2Kx]$;*

2. *For all $y \in [g(b_g), +\infty)$, $\frac{f(y)}{f'(y)} \in [\frac{1}{2K}y, 2Ky]$.*

*Thus, $g(x) = \Theta(xg'(x))$, $f(y) = \Theta(yf'(y))$.*

*Proof.* To prove Item 1, it is sufficient to show that

$$g(x) = b_f + \int_{f(b_f)}^x g'(u)du \geq \int_{x/2}^x g'(u)du$$

$$\geq (x/2) \cdot \frac{g'(x)}{K} = \frac{1}{2K} \cdot xg'(x).$$

$$x = f(b_f) + \int_{f(b_f)}^x g'(u)f'(g(u))du \geq \frac{f'(g(x))}{K} \int_{f(g(x)/2)}^x g'(u)du$$

$$= (g(x)/2) \cdot \frac{f'(g(x))}{K} = \frac{1}{2K} \cdot \frac{g(x)}{g'(x)}.$$

To prove Item 2, we only need to notice that Item 1 implies $yf'(y) = \frac{g(f(y))}{g'(f(y))} \in [\frac{1}{2K}f(y), 2Kf(y)]$ for all $y \in [g(b_g), +\infty)$. □

Recall that (B3.4) directly implies that $f'(\Theta(x)) = \Theta(f'(x))$ and $g'(\Theta(x)) = \Theta(g'(x))$. Combining this with Lemma D.1, we have the following corollary:

**Corollary D.2.** $f(\Theta(x)) = \Theta(f(x))$ *and* $g(\Theta(x)) = \Theta(g(x))$.

Also, note that Lemma D.1 essentially shows that $(\log f(x))' = \Theta(1/x)$ and $(\log g(x))' = \Theta(1/x)$. So $\log f(x) = \Theta(\log x)$ and $\log g(x) = \Theta(\log x)$, which means that $f$ and $g$ grow at most polynomially.

**Corollary D.3.** $f(x) = x^{\Theta(1)}$ *and* $g(x) = x^{\Theta(1)}$.

### D.2  PROOF FOR THEOREM A.10

We follow the notations in Section 5 to define $\rho := \|\boldsymbol{\theta}\|_2$ and $\hat{\boldsymbol{\theta}} := \frac{\boldsymbol{\theta}}{\|\boldsymbol{\theta}\|_2} \in \mathcal{S}^{d-1}$, and sometimes we view the functions of $\boldsymbol{\theta}$ as functions of $t$. And we use the notations $B_0, B_1$ from Appendix C.3.

The key idea to prove Theorem A.10 is to utilize Lemma B.6, in which $\mathcal{L}(t)$ is bounded from above by $\frac{1}{G^{-1}(\Omega(t))}$. So upper bounding $\mathcal{L}(t)$ reduces to lower bounding $G^{-1}$. In the following lemma, we obtain tight asymptotic bounds for $G(\cdot)$ and $G^{-1}(\cdot)$:

**Lemma D.4.** *For function $G(\cdot)$ defined in Lemma B.6 and its inverse function $G^{-1}(\cdot)$, we have the following bounds:*

$$G(x) = \Theta\left(\frac{g(\log x)^{2/L}}{(\log x)^2}x\right) \quad and \quad G^{-1}(y) = \Theta\left(\frac{(\log y)^2}{g(\log y)^{2/L}}y\right).$$

*Proof.* We first prove the bounds for $G(x)$, and then prove the bounds for $G^{-1}(y)$.

**Bounding for $G(x)$.** Let $C_G = \int_{1/\mathcal{L}(t_0)}^{\exp(b_g)} \frac{g'(\log u)^2}{g(\log u)^{2-2/L}}du$. For $x \geq \exp(b_g)$,

$$G(x) = C_G + \int_{\exp(b_g)}^{x} \frac{g'(\log u)^2}{g(\log u)^{2-2/L}}du$$

$$= C_G + \int_{\exp(b_g)}^{x} \left(\frac{g'(\log u)\log u}{g(\log u)}\right)^2 \frac{g(\log u)^{2/L}}{(\log u)^2}du,$$

By Lemma D.1,

$$G(x) \leq C_G + 4K^2 g(\log x)^{2/L}\int_{\exp(b_g)}^{x} \frac{1}{(\log u)^2}du = O\left(\frac{g(\log x)^{2/L}}{(\log x)^2}x\right).$$

On the other hand, for $x \geq \exp(2b_g)$, we have

$$G(x) \geq \frac{1}{4K^2}\int_{\sqrt{x}}^{x} \frac{g(\log u)^{2/L}}{(\log u)^2}du \geq \frac{g((\log x)/2)^{2/L}}{4K^2}\int_{\sqrt{x}}^{x} \frac{1}{(\log u)^2}du = \Omega\left(\frac{g(\log x)^{2/L}}{(\log x)^2}x\right).$$

**Bounding for $G^{-1}(y)$.** Let $x = G^{-1}(y)$ for $y \geq 0$. $G(x)$ always has a finite value whenever $x$ is finite. So $x \to +\infty$ when $y \to +\infty$. According to the first part of the proof, we know that $y = \Theta\left(\frac{g(\log x)^{2/L}}{(\log x)^2}x\right)$. Taking logarithm on both sides and using Corollary D.3, we have $\log y = \Theta(\log x)$. By Corollary D.2, $g(\log y) = g(\Theta(\log x)) = \Theta(g(\log x))$. Therefore,

$$\frac{(\log y)^2}{g(\log y)^{2/L}}y = \Theta\left(\frac{(\log x)^2}{g(\log x)^{2/L}}y\right) = \Theta(x).$$

This implies that $x = \Theta\left(\frac{(\log y)^2}{g(\log y)^{2/L}}y\right)$. $\qquad\square$

For other bounds, we derive them as follows. We first show that $g(\log\frac{1}{\mathcal{L}}) = \Theta(\rho^L)$. With this equivalence, we derive an upper bound for the gradient at each time $t$ in terms of $\mathcal{L}$, and take an integration to bound $\mathcal{L}(t)$ from below. Now we have both lower and upper bounds for $\mathcal{L}(t)$. Plugging these two bounds to $g(\log\frac{1}{\mathcal{L}}) = \Theta(\rho^L)$ gives the lower and upper bounds for $\rho(t)$.

*Proof for Theorem A.10.* We first prove the upper bound for $\mathcal{L}$. Then we derive lower and upper bounds for $\rho$ in terms of $\mathcal{L}$, and use these bounds to give a lower bound for $\mathcal{L}$. Finally, we plug in the tight bounds for $\mathcal{L}$ to obtain the lower and upper bounds for $\rho$ in terms of $t$.

**Upper Bounding $\mathcal{L}$.** By Lemma B.6, we have $\frac{1}{\mathcal{L}} \geq G^{-1}(\Omega(t))$. Using Lemma D.4, we have $\frac{1}{\mathcal{L}} = \Omega\left(\frac{(\log t)^2}{g(\log t)^{2/L}}t\right)$, which completes the proof.

**Bounding $\rho$ in Terms of $\mathcal{L}$.** $\tilde{\gamma}(t) \geq \tilde{\gamma}(t_0)$, so $\rho^L \leq \frac{1}{\tilde{\gamma}(t_0)}g(\log\frac{1}{\mathcal{L}}) = O(g(\log\frac{1}{\mathcal{L}}))$. On the other hand, $g(\log\frac{1}{\mathcal{L}}) \leq q_{\min} \leq B_0\rho^L$. So $\rho^L = \Omega(g(\log\frac{1}{\mathcal{L}}))$. Therefore, we have the following relationship between $\rho^L$ and $g(\log\frac{1}{\mathcal{L}})$:

$$\rho^L = \Theta(g(\log\frac{1}{\mathcal{L}})). \tag{17}$$

**Lower Bounding $\mathcal{L}$.** Let $\boldsymbol{h}_1, \ldots, \boldsymbol{h}_N$ be a set of vectors such that $\boldsymbol{h}_n \in \frac{\partial q_n}{\partial \boldsymbol{\theta}}$ and

$$\frac{d\boldsymbol{\theta}}{dt} = \sum_{n=1}^{N} e^{-f(q_n)} f'(q_n) \boldsymbol{h}_n.$$

By (17) and Corollary D.2, $f'(q_n) = f'(O(\rho^L)) = f'(O(g(\log \frac{1}{\mathcal{L}}))) = O(f'(g(\log \frac{1}{\mathcal{L}}))) = O(1/g'(\log \frac{1}{\mathcal{L}}))$. Again by (17), we have $\|\boldsymbol{h}_n\|_2 \leq B_1 \rho^{L-1} = O(g(\log \frac{1}{\mathcal{L}})^{1-1/L})$. Combining these two bounds together, it follows from Corollary D.2 that

$$f'(q_n) \|\boldsymbol{h}_n\|_2 = O\left(\frac{g(\log \frac{1}{\mathcal{L}})^{1-1/L}}{g'(\log \frac{1}{\mathcal{L}})}\right) = O\left(\frac{\log \frac{1}{\mathcal{L}}}{g(\log \frac{1}{\mathcal{L}})^{1/L}}\right).$$

Thus,

$$-\frac{d\mathcal{L}}{dt} = \left\|\frac{d\boldsymbol{\theta}}{dt}\right\|_2^2 = \left\|\sum_{n=1}^{N} e^{-f(q_n)} f'(q_n) \boldsymbol{h}_n\right\|_2^2$$
$$\leq \left(\sum_{n=1}^{N} e^{-f(q_n)} \cdot \max_{n \in [N]} \{f'(q_n) \|\boldsymbol{h}_n\|_2\}\right)^2 \leq \mathcal{L}^2 \cdot O\left(\frac{(\log \frac{1}{\mathcal{L}})^2}{g(\log \frac{1}{\mathcal{L}})^{2/L}}\right).$$

By definition of $G(\cdot)$, this implies that there exists a constant $c$ such that $\frac{d}{dt} G(\frac{1}{\mathcal{L}}) \leq c$ for any $\mathcal{L}$ that is small enough. We can complete our proof by applying Lemma D.4.

**Bounding $\rho$ in Terms of $t$.** By (17) and the tight bound for $\mathcal{L}(t)$, $\rho^L = \Theta(g(\log \frac{1}{\mathcal{L}})) = \Theta(g(\Theta(\log t)))$. Using Corollary D.2, we can conclude that $\rho^L = \Theta(g(\log t))$. $\qquad\square$

# E  GRADIENT DESCENT: SMOOTH HOMOGENEOUS MODELS WITH EXPONENTIAL LOSS

In this section, we discretize our proof to prove similar results for gradient descent on smooth homogeneous models. As usual, the update rule of gradient descent is defined as

$$\boldsymbol{\theta}(t+1) = \boldsymbol{\theta}(t) - \eta(t)\nabla\mathcal{L}(t) \tag{18}$$

Here $\eta(t)$ is the learning rate, and $\nabla\mathcal{L}(t) := \nabla\mathcal{L}(\boldsymbol{\theta}(t))$ is the gradient of $\mathcal{L}$ at $\boldsymbol{\theta}(t)$.

We first focus on the exponential loss. At the end of this section (Appendix F), we discuss how to extend the proof to general loss functions with a similar assumption as (B3).

The main difficulty for discretizing our previous analysis comes from the fact that the original version of the smoothed normalized margin $\tilde{\gamma}(\boldsymbol{\theta}) := \rho^{-L} \log \frac{1}{\mathcal{L}}$ becomes less smooth when $\rho \to +\infty$. Thus, if we take a Taylor expansion for $\tilde{\gamma}(\boldsymbol{\theta}(t+1))$ from the point $\boldsymbol{\theta}(t)$, although one can show that the first-order term is positive as in the gradient flow analysis, the second-order term is unlikely to be bounded during gradient descent with a constant step size. To get a smoothed version of the normalized margin that is monotone increasing, we need to define another one that is even smoother than $\tilde{\gamma}$.

Technically, recall that $\frac{d\mathcal{L}}{dt} = -\|\nabla\mathcal{L}\|_2^2$ does not hold exactly for gradient descent. However, if the smoothness can be bounded by $s(t)$, then it is well-known that

$$\mathcal{L}(t+1) - \mathcal{L}(t) \leq -\eta(t)(1 - s(t)\eta(t))\|\nabla\mathcal{L}\|_2^2.$$

By analyzing the landscape of $\mathcal{L}$, one can easily find that the smoothness is bounded locally by $O(\mathcal{L} \cdot \text{polylog}(\frac{1}{\mathcal{L}}))$. Thus, if we set $\eta(t)$ to a constant or set it appropriately according to the loss, then this discretization error becomes negligible. Using this insight, we define the new smoothed normalized margin $\hat{\gamma}$ in a way that it increases slightly slower than $\tilde{\gamma}$ during training to cancel the effect of discretization error.

## E.1 Assumptions

As stated in Section 4.1, we assume (A2), (A3), (A4) similarly as for gradient flow, and two additional assumptions (S1) and (S5).

**(S1).** (Smoothness). For any fixed $x$, $\Phi(\,\cdot\,;x)$ is $\mathcal{C}^2$-smooth on $\mathbb{R}^d \setminus \{0\}$.

**(A2).** (Homogeneity). There exists $L > 0$ such that $\forall \alpha > 0 : \Phi(\alpha\boldsymbol{\theta};x) = \alpha^L\Phi(\boldsymbol{\theta};x)$;

**(A3).** (Exponential Loss). $\ell(q) = e^{-q}$;

**(A4).** (Separability). There exists a time $t_0$ such that $\mathcal{L}(\boldsymbol{\theta}(t_0)) < 1$.

**(S5).** (Learing rate condition). $\sum_{t\geq 0} \eta(t) = +\infty$ and $\eta(t) \leq H(\mathcal{L}(\boldsymbol{\theta}(t)))$.

Here $H(\mathcal{L})$ is a function of the current training loss. The explicit formula of $H(\mathcal{L})$ is given below:

$$H(x) := \frac{\mu(x)}{C_\eta \kappa(x)} = \Theta\left(\frac{1}{x(\log\frac{1}{x})^{3-2/L}}\right),$$

where $C_\eta$ is a constant, and $\kappa(x), \mu(x)$ are two non-decreasing functions. For constant learning rate $\eta(t) = \eta_0$, (S5) is satisfied when $\eta_0$ if sufficiently small.

Roughly speaking, $C_\eta\kappa(x)$ is an upper bound for the smoothness of $\mathcal{L}$ in a neighborhood of $\boldsymbol{\theta}$ when $x = \mathcal{L}(\boldsymbol{\theta})$. And we set the learning rate $\eta(t)$ to be the inverse of the smoothness multiplied by a factor $\mu(x) = o(1)$. In our analysis, $\mu(x)$ can be any non-decreasing function that maps $(0, \mathcal{L}(t_0)]$ to $(0, 1/2]$ and makes the integral $\int_0^{1/2} \mu(x)dx$ exist. But for simplicity, we define $\mu(x)$ as

$$\mu(x) := \frac{\log\frac{1}{\mathcal{L}(t_0)}}{2\log\frac{1}{x}} = \Theta\left(\frac{1}{\log\frac{1}{x}}\right).$$

The value of $C_\eta$ will be specified later. The definition of $\kappa(x)$ depends on $L$. For $0 < L \leq 1$, we define $\kappa(x)$ as

$$\kappa(x) := x\left(\log\frac{1}{x}\right)^{2-2/L}$$

For $L > 1$, we define $\kappa(x)$ as

$$\kappa(x) := \begin{cases} x\left(\log\frac{1}{x}\right)^{2-2/L} & x \in (0, e^{2/L-2}] \\ \kappa_{\max} & x \in (e^{2/L-2}, 1) \end{cases}$$

where $\kappa_{\max} := e^{(2-2/L)(\ln(2-2/L)-1)}$. The specific meaning of $C_\eta, \kappa(x)$ and $\mu(x)$ will become clear in our analysis.

## E.2 Smoothed Normalized Margin

Now we define the smoothed normalized margins. As usual, we define $\tilde{\gamma}(\boldsymbol{\theta}) := \frac{\log\frac{1}{\mathcal{L}}}{\rho^L}$. At the same time, we also define

$$\hat{\gamma}(\boldsymbol{\theta}) := \frac{e^{\phi(\mathcal{L})}}{\rho^L}.$$

Here $\phi : (0, \mathcal{L}(t_0)] \to (0, +\infty)$ is constructed as follows. Construct the first-order derivative of $\phi(x)$ as

$$\phi'(x) = -\sup\left\{\frac{1 + 2(1 + \lambda(w)/L)\mu(w)}{w\log\frac{1}{w}} : w \in [x, \mathcal{L}(t_0)]\right\},$$

where $\lambda(x) := (\log\frac{1}{x})^{-1}$. And then we set $\phi(x)$ to be

$$\phi(x) = \log\log\frac{1}{x} + \int_0^x \left(\phi'(w) + \frac{1}{w\log\frac{1}{w}}\right)dw.$$

It can be verified that $\phi(x)$ is well-defined and $\phi'(x)$ is indeed the first-order derivative of $\phi(x)$. Moreover, we have the following relationship among $\hat{\gamma}, \tilde{\gamma}$ and $\bar{\gamma}$.

**Lemma E.1.** $\hat{\gamma}(\boldsymbol{\theta})$ *is well-defined for $\mathcal{L}(\boldsymbol{\theta}) \leq \mathcal{L}(t_0)$ and has the following properties:*

*(a) If $\mathcal{L}(\boldsymbol{\theta}) \leq \mathcal{L}(t_0)$, then $\hat{\gamma}(\boldsymbol{\theta}) < \tilde{\gamma}(\boldsymbol{\theta}) \leq \bar{\gamma}(\boldsymbol{\theta})$.*

*(b) Let $\{\boldsymbol{\theta}_m \in \mathbb{R}^d : m \in \mathbb{N}\}$ be a sequence of parameters. If $\mathcal{L}(\boldsymbol{\theta}_m) \to 0$, then*

$$\frac{\hat{\gamma}(\boldsymbol{\theta}_m)}{\bar{\gamma}(\boldsymbol{\theta}_m)} = 1 - O\left(\left(\log \frac{1}{\mathcal{L}(\boldsymbol{\theta}_m)}\right)^{-1}\right) \to 1.$$

*Proof.* First we verify that $\hat{\gamma}$ is well-defined. To see this, we only need to verify that

$$I(x) := \int_0^x \left(\phi'(w) + \frac{1}{w \log \frac{1}{w}}\right) dw$$

exists for all $x \in (0, \mathcal{L}(t_0)]$, then it is trivial to see that $\phi'(w)$ is indeed the derivative of $\phi(w)$ by $I'(x) = \phi'(x) + \frac{1}{x \log \frac{1}{x}}$.

Note that $I(x)$ exists for all $x \in (0, \mathcal{L}(t_0)]$ as long as $I(x)$ exists for a small enough $x > 0$. By definition, it is easy to verify that $r(w) := \frac{1 + 2(1 + \lambda(w)/L)\mu(w)}{w \log \frac{1}{w}}$ is decreasing when $w$ is small enough. Thus, for a small enough $w > 0$, we have

$$-\phi'(w) = r(w) = \frac{1 + 2(1 + \frac{1}{L}(\log \frac{1}{w})^{-1})\frac{\log \frac{1}{\mathcal{L}(t_0)}}{2 \log \frac{1}{w}}}{w \log \frac{1}{w}} = \frac{1}{w \log \frac{1}{w}} + \frac{\log \frac{1}{\mathcal{L}(t_0)}(1 + \frac{1}{L}(\log \frac{1}{w})^{-1})}{w(\log \frac{1}{w})^2}.$$

So we have the following for small enough $x$:

$$I(x) = \int_0^x -\frac{\log \frac{1}{\mathcal{L}(t_0)}(1 + \frac{1}{L}(\log \frac{1}{w})^{-1})}{w(\log \frac{1}{w})^2} dw = -\log \frac{1}{\mathcal{L}(t_0)}\left(\frac{1}{\log \frac{1}{w}} + \frac{1}{2L(\log \frac{1}{w})^2}\right)\Big|_0^x$$

$$= -\log \frac{1}{\mathcal{L}(t_0)}\left(\frac{1}{\log \frac{1}{x}} + \frac{1}{2L(\log \frac{1}{x})^2}\right). \quad (19)$$

This proves the existence of $I(x)$.

Now we prove (a). By Lemma A.5, $\tilde{\gamma}(\boldsymbol{\theta}) \leq \bar{\gamma}(\boldsymbol{\theta})$, so we only need to prove that $\hat{\gamma}(\boldsymbol{\theta}) < \tilde{\gamma}(\boldsymbol{\theta})$. To see this, note that for all $w \leq \mathcal{L}(t_0)$, $r(w) > \frac{1}{w \log \frac{1}{w}}$. So we have $-\phi'(w) > \frac{1}{w \log \frac{1}{w}}$, and this implies $I(x) < 0$ for all $x \leq \mathcal{L}(t_0)$. Then it holds for all $\boldsymbol{\theta}$ with $\mathcal{L}(\boldsymbol{\theta}) \leq \mathcal{L}(t_0)$ that

$$\frac{\hat{\gamma}(\boldsymbol{\theta})}{\tilde{\gamma}(\boldsymbol{\theta})} = \frac{e^{\phi(\mathcal{L}(\boldsymbol{\theta}))}}{\log \frac{1}{\mathcal{L}(\boldsymbol{\theta})}} = \frac{e^{\log \log \frac{1}{\mathcal{L}(\boldsymbol{\theta})} + I(\mathcal{L}(\boldsymbol{\theta}))}}{e^{\log \log \frac{1}{\mathcal{L}(\boldsymbol{\theta})}}} = e^{I(\mathcal{L}(\boldsymbol{\theta}))} < 0. \quad (20)$$

To prove (b), we combine (19) and (20), then for small enough $\mathcal{L}(\boldsymbol{\theta}_m)$, we have

$$\frac{\hat{\gamma}(\boldsymbol{\theta})}{\bar{\gamma}(\boldsymbol{\theta})} = \frac{\hat{\gamma}(\boldsymbol{\theta})}{\tilde{\gamma}(\boldsymbol{\theta})} \cdot \frac{\tilde{\gamma}(\boldsymbol{\theta})}{\bar{\gamma}(\boldsymbol{\theta})}$$

$$= \exp\left(-\log \frac{1}{\mathcal{L}(t_0)}\left(\frac{1}{\log \frac{1}{\mathcal{L}(\boldsymbol{\theta}_m)}} + \frac{1}{2L(\log \frac{1}{\mathcal{L}(\boldsymbol{\theta}_m)})^2}\right)\right) \cdot \frac{\log \frac{1}{\mathcal{L}(\boldsymbol{\theta}_m)}}{q_{\min}(\boldsymbol{\theta}_m)}$$

$$= \exp\left(-O\left(\frac{1}{\log \frac{1}{\mathcal{L}(\boldsymbol{\theta}_m)}}\right)\right) \cdot \frac{\log \frac{1}{\mathcal{L}(\boldsymbol{\theta}_m)}}{\log \frac{1}{\mathcal{L}(\boldsymbol{\theta}_m)} + O(1)}$$

$$= 1 - O\left(\left(\log \frac{1}{\mathcal{L}(\boldsymbol{\theta}_m)}\right)^{-1}\right).$$

So $\frac{\hat{\gamma}(\boldsymbol{\theta})}{\bar{\gamma}(\boldsymbol{\theta})} = 1 - O\left((\log \frac{1}{\mathcal{L}(\boldsymbol{\theta}_m)})^{-1}\right) \to 1$. $\square$

Now we specify the value of $C_\eta$. By (S1) and (S2), we can define $B_0, B_1, B_2$ as follows:

$$B_0 := \sup\left\{q_n : \boldsymbol{\theta} \in \mathcal{S}^{d-1}, n \in [N]\right\},$$
$$B_1 := \sup\left\{\|\nabla q_n\|_2 : \boldsymbol{\theta} \in \mathcal{S}^{d-1}, n \in [N]\right\},$$
$$B_2 := \sup\left\{\left\|\nabla^2 q_n\right\|_2 : \boldsymbol{\theta} \in \mathcal{S}^{d-1}, n \in [N]\right\}.$$

Then we set $C_\eta := \frac{1}{2}\left(B_1^2 + \rho(t_0)^{-L}B_2\right)\min\left\{\hat{\gamma}(t_0)^{-2+2/L}, B_0^{-2+2/L}\right\}$.

### E.3 THEOREMS

Now we state our main theorems for the monotonicity of the normalized margin and the convergence to KKT points. We will prove Theorem E.2 in Appendix E.4, and prove Theorem E.3 and E.4 in Appendix E.5.

**Theorem E.2.** *Under assumptions (S1), (A2) - (A4), (S5), the following are true for gradient descent:*

1. *For all $t \geq t_0$, $\hat{\gamma}(t+1) \geq \hat{\gamma}(t)$;*

2. *For all $t \geq t_0$, either $\hat{\gamma}(t+1) > \hat{\gamma}(t)$ or $\hat{\boldsymbol{\theta}}(t+1) = \hat{\boldsymbol{\theta}}(t)$;*

3. *$\mathcal{L}(t) \to 0$ and $\rho(t) \to \infty$ as $t \to +\infty$; therefore, $|\bar{\gamma}(t) - \tilde{\gamma}(t)| \to 0$.*

**Theorem E.3.** *Consider gradient flow under assumptions (S1), (A2) - (A4), (S5). For every limit point $\bar{\boldsymbol{\theta}}$ of $\left\{\hat{\boldsymbol{\theta}}(t) : t \geq 0\right\}$, $\bar{\boldsymbol{\theta}}/q_{\min}(\bar{\boldsymbol{\theta}})^{1/L}$ is a KKT point of (P).*

**Theorem E.4.** *Consider gradient descent under assumptions (S1), (A2) - (A4), (S5). For any $\epsilon, \delta > 0$, there exists $r := \Theta(\log \delta^{-1})$ and $\Delta := \Theta(\epsilon^{-2})$ such that $\boldsymbol{\theta}/q_{\min}(\boldsymbol{\theta})^{1/L}$ is an $(\epsilon, \delta)$-KKT point at some time $t_*$ satisfying $\log \rho(t_*) \in (r, r+\Delta)$.*

With a refined analysis, we can also derive tight rates for loss convergence and weight growth. We defer the proof to Appendix E.6.

**Theorem E.5.** *Under assumptions (S1), (A2) - (A4), (S5), we have the following tight rates for training loss and weight norm:*

$$\mathcal{L}(t) = \Theta\left(\frac{1}{T(\log T)^{2-2/L}}\right) \quad and \quad \rho(t) = \Theta((\log T)^{1/L}),$$

*where $T = \sum_{\tau=t_0}^{t-1} \eta(\tau)$.*

### E.4 PROOF FOR THEOREM E.2

We define $\nu(t) := \sum_{n=1}^N e^{-q_n(t)}q_n(t)$ as we do for gradient flow. Then we can get a closed form for $\langle \boldsymbol{\theta}(t), -\nabla\mathcal{L}(t)\rangle$ easily from Corollary B.3. Also, we can get a lower bound for $\nu(t)$ using Lemma B.5 for exponential loss directly.

**Corollary E.6.** $\langle \boldsymbol{\theta}(t), -\nabla\mathcal{L}(t)\rangle = L\nu(t)$. *If $\mathcal{L}(t) < 1$, then $\nu(t) \geq \frac{\mathcal{L}(t)}{\lambda(\mathcal{L}(t))}$.*

As we are analyzing gradient descent, the norm of $\nabla\mathcal{L}$ and $\nabla^2\mathcal{L}$ appear naturally in discretization error terms. To bound them, we have the following lemma when $\tilde{\gamma}(\boldsymbol{\theta})$ and $\rho(\boldsymbol{\theta})$ have lower bounds:

**Lemma E.7.** *For any $\boldsymbol{\theta}$, if $\tilde{\gamma}(\boldsymbol{\theta}) \geq \hat{\gamma}(t_0), \rho(\boldsymbol{\theta}) \geq \rho(t_0)$, then*

$$\|\nabla\mathcal{L}(\boldsymbol{\theta})\|_2^2 \leq 2C_\eta\kappa(\mathcal{L}(\boldsymbol{\theta}))\mathcal{L}(\boldsymbol{\theta}) \quad and \quad \left\|\nabla^2\mathcal{L}(\boldsymbol{\theta})\right\|_2 \leq 2C_\eta\kappa(\mathcal{L}(\boldsymbol{\theta})).$$

*Proof.* By the chain rule and the definitions of $B_1$, $B_2$, we have

$$\|\nabla \mathcal{L}\|_2 = \left\|-\sum_{n=1}^{N} e^{-q_n} \nabla q_n\right\|_2 \le \mathcal{L}\rho^{L-1} B_1$$

$$\|\nabla^2 \mathcal{L}\|_2 = \left\|\sum_{n=1}^{N} e^{-q_n} (\nabla q_n \nabla q_n^\top - \nabla^2 q_n)\right\|_2$$

$$\le \sum_{n=1}^{N} e^{-q_n} \left(B_1^2 \rho^{2L-2} + B_2 \rho^{L-2}\right) \le \mathcal{L}\rho^{2L-2}\left(B_1^2 + \rho^{-L} B_2\right).$$

Note that $\hat{\gamma}(t_0)\rho^L \le \tilde{\gamma}\rho^L \le \log \frac{1}{\mathcal{L}} \le B_0 \rho^L$. So $\hat{\gamma}(t_0)^{-1} \log \frac{1}{\mathcal{L}} \le \rho^L \le B_0^{-1} \log \frac{1}{\mathcal{L}}$. Combining all these formulas together gives

$$\|\nabla \mathcal{L}\|_2^2 \le B_1^2 \min\left\{\hat{\gamma}(t_0)^{-2+2/L}, B_0^{-2+2/L}\right\} \mathcal{L}^2 \cdot \left(\log \frac{1}{\mathcal{L}}\right)^{2-2/L} \le 2C_\eta \kappa(\mathcal{L})\mathcal{L}$$

$$\|\nabla^2 \mathcal{L}\|_2 \le \left(B_1^2 + \rho^{-L} B_2\right) \min\left\{\hat{\gamma}(t_0)^{-2+2/L}, B_0^{-2+2/L}\right\} \mathcal{L} \cdot \left(\log \frac{1}{\mathcal{L}}\right)^{2-2/L} \le 2C_\eta \kappa(\mathcal{L}),$$

which completes the proof. $\qquad\square$

For proving the first two propositions in Theorem E.2, we only need to prove Lemma E.8. (P1) gives a lower bound for $\tilde{\gamma}$. (P2) gives both lower and upper bounds for the weight growth using $\nu(t)$. (P3) gives a lower bound for the decrement of training loss. Finally, (P4) shows the monotonicity of $\hat{\gamma}$, and it is trivial to deduce the first two propositions in Theorem E.2 from (P4).

**Lemma E.8.** *For all $t = t_0, t_0 + 1, \ldots$, we interpolate between $\boldsymbol{\theta}(t)$ and $\boldsymbol{\theta}(t+1)$ by defining $\boldsymbol{\theta}(t+\alpha) = \boldsymbol{\theta}(t) - \alpha\eta(t)\nabla\mathcal{L}(t)$ for $\alpha \in (0,1)$. Then for all integer $t \ge t_0$, $\nu(t) > 0$, and the following holds for all $\alpha \in [0,1]$:*

**(P1).** $\tilde{\gamma}(t+\alpha) > \hat{\gamma}(t_0)$.

**(P2).** $2L\alpha\eta(t)\nu(t) \le \rho(t+\alpha)^2 - \rho(t)^2 \le 2L\alpha\eta(t)\nu(t)\left(1 + \frac{\lambda(\mathcal{L}(t))\mu(\mathcal{L}(t))}{L}\right)$.

**(P3).** $\mathcal{L}(t+\alpha) - \mathcal{L}(t) \le -\alpha\eta(t)(1 - \mu(\mathcal{L}(t)))\|\nabla\mathcal{L}(t)\|_2^2$.

**(P4).** $\log\hat{\gamma}(t+\alpha) - \log\hat{\gamma}(t) \ge \frac{\rho(t)^2}{L\nu(t)^2}\left\|\left(\boldsymbol{I} - \hat{\boldsymbol{\theta}}(t)\hat{\boldsymbol{\theta}}(t)^\top\right)\nabla\mathcal{L}(t)\right\|_2^2 \cdot \log\frac{\rho(t+\alpha)}{\rho(t)}$.

To prove Lemma E.8, we only need to prove the following lemma and then use an induction:

**Lemma E.9.** *Fix an integer $T \ge t_0$. Suppose that (P1), (P2), (P3), (P4) hold for any $t + \alpha \le T$. Then if (P1) holds for $(t, \alpha) \in \{T\} \times [0, A)$ for some $A \in (0, 1]$, then all of (P1), (P2), (P3), (P4) hold for $(t, \alpha) \in \{T\} \times [0, A]$.*

*Proof for Lemma E.8.* We prove this lemma by induction. For $t = t_0, \alpha = 0$, $\nu(t) > 0$ by (S4) and Corollary E.6. (P2), (P3), (P4) hold trivially since $\log\hat{\gamma}(t+\alpha) = \log\hat{\gamma}(t)$, $\mathcal{L}(t+\alpha) = \mathcal{L}(t)$ and $\log\hat{\gamma}(t+\alpha) = \log\hat{\gamma}(t)$. By Lemma E.1, (P1) also holds trivially.

Now we fix an integer $T \ge t_0$ and assume that (P1), (P2), (P3), (P4) hold for any $t + \alpha \le T$ (where $t \ge t_0$ is an integer and $\alpha \in [0, 1]$). By (P3), $\mathcal{L}(t) \le \mathcal{L}(t_0) < 1$, so $\nu(t) > 0$. We only need to show that (P1), (P2), (P3), (P4) hold for $t = T$ and $\alpha \in [0, 1]$.

Let $A := \inf\{\alpha \in [0, 1] : \alpha = 1$ or (P1) does not hold for $(T, \alpha)\}$. If $A = 0$, then (P1) holds for $(T, A)$ since (P1) holds for $(T - 1, 1)$; if $A > 0$, we can also know that (P1) holds for $(T, A)$ by Lemma E.9. Suppose that $A < 1$. Then by the continuity of $\tilde{\gamma}(T + \alpha)$ (with respect to $\alpha$), we know that there exists $A' > A$ such that $\tilde{\gamma}(T + \alpha) > \hat{\gamma}(t_0)$ for all $\alpha \in [A, A']$, which contradicts to the definition of $A$. Therefore, $A = 1$. Using Lemma E.9 again, we can conclude that (P1), (P2), (P3), (P4) hold for $t = T$ and $\alpha \in [0, 1]$. $\qquad\square$

Now we turn to prove Lemma E.9.

*Proof for Lemma E.9.* Applying (P3) on $(t, \alpha) \in \{t_0, \ldots, T-1\} \times 1$, we have $\mathcal{L}(t) \leq \mathcal{L}(t_0) < 1$. Then by Corollary E.6, we have $\nu(t) > 0$. Applying (P2) on $(t, \alpha) \in \{t_0, \ldots, T-1\} \times 1$, we can get $\rho(t) \geq \rho(t_0)$.

Fix $t = T$. By (P2) with $\alpha \in [0, A)$ and the continuity of $\tilde{\gamma}$, we have $\tilde{\gamma}(t + A) \geq \hat{\gamma}(t_0)$. Thus, $\tilde{\gamma}(t + \alpha) \geq \hat{\gamma}(t_0)$ for all $\alpha \in [0, A]$.

**Proof for (P2).** By definition, $\rho(t+\alpha)^2 = \|\boldsymbol{\theta}(t) - \alpha\eta(t)\nabla\mathcal{L}(t)\|_2^2 = \rho(t)^2 + \alpha^2\eta(t)^2 \|\nabla\mathcal{L}(t)\|_2^2 - 2\alpha\eta(t) \langle\boldsymbol{\theta}(t), \nabla\mathcal{L}(t)\rangle$. By Corollary E.6, we have

$$\rho(t+\alpha)^2 - \rho(t)^2 = \alpha^2\eta(t)^2 \|\nabla\mathcal{L}(t)\|_2^2 + 2L\alpha\eta(t)\nu(t).$$

So $\rho(t+\alpha)^2 - \rho(t)^2 \geq 2L\alpha\eta(t)\nu(t)$. For the other direction, we have the following using Corollary E.6 and Lemma E.7,

$$\rho(t+\alpha)^2 - \rho(t)^2 = 2L\alpha\eta(t)\nu(t) \left(1 + \frac{\alpha\eta(t) \|\nabla\mathcal{L}(t)\|_2^2}{2L\nu(t)}\right)$$

$$\leq 2L\alpha\eta(t)\nu(t) \left(1 + \frac{C_\eta^{-1}\kappa(\mathcal{L}(t))^{-1}\mu(\mathcal{L}(t)) \cdot 2C_\eta\kappa(\mathcal{L}(t))\mathcal{L}(t)}{2L\mathcal{L}(t)\lambda(\mathcal{L}(t))^{-1}}\right)$$

$$= 2L\alpha\eta(t)\nu(t) \left(1 + \lambda(\mathcal{L}(t))\mu(\mathcal{L}(t))/L\right).$$

**Proof for (P3).** (P3) holds trivially for $\alpha = 0$ or $\nabla\mathcal{L}(t) = \mathbf{0}$. So now we assume that $\alpha \neq 0$ and $\nabla\mathcal{L}(t) \neq \mathbf{0}$. By the update rule (18) and Taylor expansion, there exists $\xi \in (0, \alpha)$ such that

$$\mathcal{L}(t+\alpha) = \mathcal{L}(t) + (\boldsymbol{\theta}(t+\alpha) - \boldsymbol{\theta}(t))^\top \nabla\mathcal{L}(t) + \frac{1}{2}(\boldsymbol{\theta}(t+\alpha) - \boldsymbol{\theta}(t))^\top \nabla^2\mathcal{L}(t+\xi)(\boldsymbol{\theta}(t+\alpha) - \boldsymbol{\theta}(t))$$

$$\leq \mathcal{L}(t) - \alpha\eta(t) \|\nabla\mathcal{L}(t)\|_2^2 + \frac{1}{2}\alpha^2\eta(t)^2 \|\nabla^2\mathcal{L}(t+\xi)\|_2 \|\nabla\mathcal{L}(t)\|_2^2$$

$$= \mathcal{L}(t) - \alpha\eta(t) \left(1 - \frac{1}{2}\alpha\eta(t) \|\nabla^2\mathcal{L}(t+\xi)\|_2\right) \|\nabla\mathcal{L}(t)\|_2^2.$$

By Lemma E.7, $\|\nabla^2\mathcal{L}(t+\xi)\|_2 \leq 2C_\eta \cdot \kappa(\mathcal{L}(t+\xi))$, so we have

$$\mathcal{L}(t+\alpha) \leq \mathcal{L}(t) - \alpha\eta(t) \left(1 - \alpha C_\eta\eta(t)\kappa(\mathcal{L}(t+\xi))\right) \|\nabla\mathcal{L}(t)\|_2^2. \tag{21}$$

Now we only need to show that $\mathcal{L}(t+\alpha) < \mathcal{L}(t)$ for all $\alpha \in (0, A]$. Assuming this, we can have $\kappa(\mathcal{L}(t+\xi)) \leq \kappa(\mathcal{L}(t))$ by the monotonicity of $\kappa$, and thus

$$\mathcal{L}(t+\alpha) \leq \mathcal{L}(t) - \alpha\eta(t) \left(1 - \alpha C_\eta\eta(t)\kappa(\mathcal{L}(t))\right) \|\nabla\mathcal{L}(t)\|_2^2$$

$$\leq \mathcal{L}(t) - \alpha\eta(t) \left(1 - \mu(\mathcal{L}(t))\right) \|\nabla\mathcal{L}(t)\|_2^2.$$

Now we show that $\mathcal{L}(t+\alpha) < \mathcal{L}(t)$ for all $\alpha \in (0, A]$. Assume to the contrary that $\alpha_0 := \inf\{\alpha' \in (0, A] : \mathcal{L}(t+\alpha') \geq \mathcal{L}(t)\}$ exists. If $\alpha_0 = 0$, let $p : [0, 1] \to \mathbb{R}, \alpha \mapsto \mathcal{L}(t+\alpha)$, then $p'(0) = \lim_{\alpha\downarrow 0} \frac{\mathcal{L}(t+\alpha)-\mathcal{L}(t)}{\alpha} \geq 0$, but it contradicts to $p'(0) = -\eta(t) \|\nabla\mathcal{L}(t)\|_2^2 < 0$. If $\alpha_0 > 0$, then by the monotonicity of $\kappa$ we have $\kappa(\mathcal{L}(t+\xi)) < \kappa(\mathcal{L}(t))$ for all $\xi \in (0, \alpha_0)$, and thus

$$\mathcal{L}(t+\alpha) \leq \mathcal{L}(t) - \alpha\eta(t) \left(1 - \mu(\mathcal{L}(t))\right) \|\nabla\mathcal{L}(t)\|_2^2 < \mathcal{L}(t),$$

which leads to a contradiction.

**Proof for (P4).** We define $\boldsymbol{v}(t) := \hat{\boldsymbol{\theta}}(t)\hat{\boldsymbol{\theta}}(t)^\top(-\nabla\mathcal{L}(t))$ and $\boldsymbol{u}(t) := \left(\boldsymbol{I} - \hat{\boldsymbol{\theta}}(t)\hat{\boldsymbol{\theta}}(t)^\top\right)(-\nabla\mathcal{L}(t))$ similarly as in the analysis for gradient flow. For $\boldsymbol{v}(t)$, we have

$$\|\boldsymbol{v}(t)\|_2 = \frac{1}{\rho(t)} \langle\boldsymbol{\theta}(t), -\mathcal{L}(t)\rangle = \frac{1}{\rho(t)} \cdot L\nu(t).$$

Decompose $\|\nabla\mathcal{L}(t)\|_2^2 = \|\boldsymbol{v}(t)\|_2^2 + \|\boldsymbol{u}(t)\|_2^2$. Then by (P3), we have

$$\mathcal{L}(t+\alpha) - \mathcal{L}(t) \leq -\alpha\eta(t)(1-\mu(\mathcal{L}(t)))\left(\frac{1}{\rho(t)^2}\cdot L^2\nu(t)^2 + \|\boldsymbol{u}\|_2^2\right).$$

Multiplying $\frac{1+\lambda(\mathcal{L}(t))\mu(\mathcal{L}(t))/L}{(1-\mu(\mathcal{L}(t)))\nu(t)}$ on both sides, we have

$$\frac{1+\lambda(\mathcal{L}(t))\mu(\mathcal{L}(t))/L}{(1-\mu(\mathcal{L}(t)))\nu(t)}(\mathcal{L}(t+\alpha)-\mathcal{L}(t)) \leq -\alpha\eta(t)\left(1+\frac{\lambda(\mathcal{L}(t))\mu(\mathcal{L}(t))}{L}\right)\left(\frac{L^2\nu(t)}{\rho(t)^2}+\frac{\|\boldsymbol{u}\|_2^2}{\nu(t)}\right)$$

By Corollary E.6, we can bound $\nu(t)$ by $\nu(t) \geq \mathcal{L}(t)/\lambda(\mathcal{L}(t))$. By (P2), we have the inequality $\alpha\eta(t)\nu(t)\left(1+\frac{\lambda(\mathcal{L}(t))\mu(\mathcal{L}(t))}{L}\right) \geq \frac{\rho(t+\alpha)^2-\rho(t)^2}{2L}$. So we further have

$$\frac{1+\lambda(\mathcal{L}(t))\mu(\mathcal{L}(t))/L}{(1-\mu(\mathcal{L}(t)))\mathcal{L}(t)/\lambda(\mathcal{L}(t))}(\mathcal{L}(t+\alpha)-\mathcal{L}(t)) \leq -\frac{1}{\rho(t)^2}(\rho(t+\alpha)^2-\rho(t)^2)\left(\frac{L}{2}+\frac{\rho(t)^2}{2L\nu(t)^2}\|\boldsymbol{u}\|_2^2\right)$$

From the definition $\phi$, it is easy to see that $-\phi'(\mathcal{L}(t)) \geq \frac{1+\lambda(\mathcal{L}(t))\mu(\mathcal{L}(t))/L}{(1-\mu(\mathcal{L}(t)))\mathcal{L}(t)/\lambda(\mathcal{L}(t))}$. Let $\psi(x) = -\log x$, then $\psi'(x) = -\frac{1}{x}$. Combining these together gives

$$\phi'(\mathcal{L}(t))(\mathcal{L}(t+\alpha)-\mathcal{L}(t)) + \psi'(\rho(t)^2)(\rho(t+\alpha)^2-\rho(t)^2)\left(\frac{L}{2}+\frac{\rho(t)^2}{2L\nu(t)^2}\|\boldsymbol{u}\|_2^2\right) \geq 0$$

Then by convexity of $\phi$ and $\psi$, we have

$$(\phi(\mathcal{L}(t+\alpha))-\phi(\mathcal{L}(t))) + \left(\log\frac{1}{\rho(t+\alpha)^2}-\log\frac{1}{\rho(t)^2}\right)\left(\frac{L}{2}+\frac{\rho(t)^2}{2L\nu(t)^2}\|\boldsymbol{u}\|_2^2\right) \geq 0$$

And by definition of $\hat\gamma$, this can be re-written as

$$\log\hat\gamma(t+\alpha) - \log\hat\gamma(t) = (\phi(\mathcal{L}(t+\alpha))-\phi(\mathcal{L}(t))) + L\left(\log\frac{1}{\rho(t+\alpha)}-\log\frac{1}{\rho(t)}\right)$$
$$\geq -\left(\log\frac{1}{\rho(t+\alpha)^2}-\log\frac{1}{\rho(t)^2}\right)\frac{\rho(t)^2}{2L\nu(t)^2}\|\boldsymbol{u}\|_2^2$$
$$= \frac{\rho(t)^2}{L\nu(t)^2}\|\boldsymbol{u}\|_2^2 \cdot \log\frac{\rho(t+\alpha)}{\rho(t)}.$$

**Proof for (P1).** By (P4), $\log\hat\gamma(t+\alpha) \geq \log\hat\gamma(t) \geq \log\hat\gamma(t_0)$. Note that $\phi(x) \geq \log\log\frac{1}{x}$. So we have
$$\tilde\gamma(t+\alpha) > \hat\gamma(t+\alpha) \geq \hat\gamma(t_0),$$
which completes the proof. $\qquad\square$

For showing the third proposition in Theorem E.2, we use (P1) to give a lower bound for $\|\nabla\mathcal{L}(t)\|_2$, and use (P3) to show the speed of loss decreasing. Then it can be seen that $\mathcal{L}(t) \to 0$ and $\rho(t) \to +\infty$. By Lemma E.1, we then have $|\bar\gamma - \hat\gamma| \to 0$.

**Lemma E.10.** *Let* $E_0 := \mathcal{L}(t_0)^2(\log\frac{1}{\mathcal{L}(t_0)})^{2-2/L}$. *Then for all* $t > t_0$,

$$E(\mathcal{L}(t)) \geq \frac{1}{2}L^2\hat\gamma(t_0)^{2/L}\sum_{\tau=t_0}^{t-1}\eta(\tau) \quad \text{for } E(x) := \int_x^{\mathcal{L}(t_0)}\frac{1}{\min\{u^2(\log\frac{1}{u})^{2-2/L}, E_0\}}du.$$

*Therefore,* $\mathcal{L}(t) \to 0$ *and* $\rho(t) \to +\infty$ *as* $t \to \infty$.

*Proof.* For any integer $t \geq t_0$, $\mu(\mathcal{L}(t)) \leq \frac{1}{2}$ and $\|\nabla\mathcal{L}(t)\|_2 \geq \|\boldsymbol{v}(t)\|_2$. Combining these with (P3), we have

$$\mathcal{L}(t+1) - \mathcal{L}(t) \leq -\frac{1}{2}\eta(t)\|\boldsymbol{v}(t)\|_2^2 = -\frac{1}{2}\eta(t)\frac{L^2\nu(t)^2}{\rho(t)^2}.$$

By (P1), $\rho(t)^{-2} \le \hat{\gamma}(t_0)^{2/L} \left( \log \frac{1}{\mathcal{L}(t)} \right)^{-2/L}$. By Corollary E.6, $\nu(t)^2 \ge \mathcal{L}(t)^2 \left( \log \frac{1}{\mathcal{L}(t)} \right)^2$. Thus we have

$$\mathcal{L}(t+1) - \mathcal{L}(t) \le -\frac{1}{2}\eta(t) \cdot L^2 \hat{\gamma}(t_0)^{2/L} \mathcal{L}(t)^2 \left( \log \frac{1}{\mathcal{L}(t)} \right)^{2-2/L}.$$

It is easy to see that $\frac{1}{u^2 (\log \frac{1}{u})^{2-2/L}}$ is unimodal in $(0,1)$, so $E'(x)$ is non-decreasing and $E(x)$ is convex. So we have

$$E(\mathcal{L}(t+1)) - E(\mathcal{L}(t)) \ge E'(\mathcal{L}(t))\left(\mathcal{L}(t+1) - \mathcal{L}(t)\right)$$

$$\ge \frac{1}{2}\eta(t) \cdot L^2 \hat{\gamma}(t_0)^{2/L} \cdot \frac{\mathcal{L}(t)^2 \left( \log \frac{1}{\mathcal{L}(t)} \right)^{2-2/L}}{\min\{\mathcal{L}(t)^2 (\log \frac{1}{\mathcal{L}(t)})^{2-2/L}, E_0\}}$$

$$\ge \frac{1}{2}\eta(t) \cdot L^2 \hat{\gamma}(t_0)^{2/L},$$

which proves $E(\mathcal{L}(t)) \ge \frac{1}{2}L^2 \hat{\gamma}(t_0)^{2/L} \sum_{\tau=t_0}^{t-1} \eta(\tau)$. Note that $\mathcal{L}$ is non-decreasing. If $\mathcal{L}$ does not decreases to 0, then neither does $E(\mathcal{L})$. But the RHS grows to $+\infty$, which leads to a contradiction. So $\mathcal{L} \to 0$. To make $\mathcal{L} \to 0$, $q_{\min}$ must converge to $+\infty$. So $\rho \to +\infty$. $\qquad\square$

### E.5 PROOF FOR THEOREM E.3 AND E.4

The proofs for Theorem E.3 and E.4 are similar as those for Theorem A.8 and A.9 in Appendix C.

Define $\beta(t) := \frac{1}{\|\nabla\mathcal{L}(t)\|_2}\left\langle \hat{\boldsymbol{\theta}}, -\nabla\mathcal{L}(t) \right\rangle$ as we do in Appendix C. It is easy to see that Lemma C.8 still holds if we replace $\tilde{\gamma}(t_0)$ with $\hat{\gamma}(t_0)$. So we only need to show $\mathcal{L} \to 0$ and $\beta \to 1$ for proving convergence to KKT points. $\mathcal{L} \to 0$ can be followed from Theorem E.2. Similar as the proof for Lemma C.9, it follows from Lemma E.8 and (15) that for all $t_2 > t_1 \ge t_0$,

$$\sum_{t=t_1}^{t_2-1} \left( \beta(t)^{-2} - 1 \right) \cdot \log \frac{\rho(t+1)}{\rho(t)} \le \frac{1}{L} \log \frac{\hat{\gamma}(t_2)}{\hat{\gamma}(t_1)}. \tag{22}$$

Now we prove Theorem E.4.

*Proof for Theorem E.4.* We make the following changes in the proof for Theorem A.9. First, we replace $\tilde{\gamma}(t_0)$ with $\hat{\gamma}(t_0)$, since $\tilde{\gamma}(t)$ ($t \ge t_0$) is lower bounded by $\hat{\gamma}(t_0)$ rather than $\tilde{\gamma}(t_0)$. Second, when choosing $t_1$ and $t_2$, we make $\log \rho(t_1)$ and $\log \rho(t_2)$ equal to the chosen values approximately with an additive error $o(1)$, rather than make them equal exactly. This is possible because it can be shown from (P2) in Lemma E.8 that the following holds:

$$\rho(t+1)^2 - \rho(t)^2 = O(\eta(t)\nu(t))$$

$$= o(\kappa(\mathcal{L}(t))^{-1} \cdot \mathcal{L}(t)\rho(t)^L) = o\left( \frac{\rho(t)^L}{(\log \frac{1}{\mathcal{L}(t)})^{2-2/L}} \right) = o(\rho(t)^{-L+2}).$$

Dividing $\rho(t)^2$ on the leftmost and rightmost sides, we have

$$\rho(t+1)/\rho(t) = 1 + o(\rho(t)^{-L}) = 1 + o(1),$$

which implies that $\log \rho(t+1) - \log \rho(t) = o(1)$. Therefore, for any $R$, we can always find the minimum time $t$ such that $\log \rho(t) \ge R$, and it holds for sure that $\log \rho(t) - R \to 0$ as $R \to +\infty$. $\quad\square$

For proving Theorem E.3, we also need the following lemma as a variant of Lemma C.11.

**Lemma E.11.** *For all $t \ge t_0$,*

$$\left\| \hat{\boldsymbol{\theta}}(t+1) - \hat{\boldsymbol{\theta}}(t) \right\|_2 \le \left( \frac{B_1}{L\tilde{\gamma}(t)} \cdot \frac{\rho(t+1)}{\rho(t)} + 1 \right) \log \frac{\rho(t+1)}{\rho(t)}$$

*Proof.* According to the update rule, we have

$$\left\| \hat{\boldsymbol{\theta}}(t+1) - \hat{\boldsymbol{\theta}}(t) \right\|_2 = \frac{1}{\rho(t+1)} \left\| \boldsymbol{\theta}(t+1) - \frac{\rho(t+1)}{\rho(t)} \boldsymbol{\theta}(t) \right\|_2$$

$$\leq \frac{1}{\rho(t+1)} \left( \left\| \boldsymbol{\theta}(t+1) - \boldsymbol{\theta}(t) \right\|_2 + \left\| \left( \frac{\rho(t+1)}{\rho(t)} - 1 \right) \boldsymbol{\theta}(t) \right\|_2 \right)$$

$$= \frac{\eta(t)}{\rho(t+1)} \left\| \nabla \mathcal{L}(t) \right\|_2 + \left( 1 - \frac{\rho(t)}{\rho(t+1)} \right).$$

By (16), $\| \nabla \mathcal{L}(t) \|_2 \leq \frac{B_1 \nu(t)}{\tilde{\gamma}(t) \rho(t)}$. So we can bound the first term as

$$\frac{\eta(t)}{\rho(t+1)} \| \nabla \mathcal{L}(t) \|_2 \leq \frac{B_1}{\tilde{\gamma}(t)} \cdot \frac{\eta(t) \nu(t)}{\rho(t+1) \rho(t)} \leq \frac{B_1}{\tilde{\gamma}(t)} \cdot \frac{\rho(t+1)^2 - \rho(t)^2}{2 L \rho(t+1) \rho(t)}$$

$$= \frac{B_1}{2 L \tilde{\gamma}(t)} \cdot \frac{\rho(t+1)^2 - \rho(t)^2}{\rho(t+1)^2} \cdot \frac{\rho(t+1)}{\rho(t)}$$

$$\leq \frac{B_1}{L \tilde{\gamma}(t)} \cdot \frac{\rho(t+1)}{\rho(t)} \log \frac{\rho(t+1)}{\rho(t)}$$

where the last inequality uses the inequality $\frac{a-b}{a} \leq \log(a/b)$. Using this inequality again, we can bound the second term by

$$1 - \frac{\rho(t)}{\rho(t+1)} \leq \log \frac{\rho(t+1)}{\rho(t)}.$$

Combining these together gives $\left\| \hat{\boldsymbol{\theta}}(t+1) - \hat{\boldsymbol{\theta}}(t) \right\|_2 \leq \left( \frac{B_1}{L\tilde{\gamma}(t)} \cdot \frac{\rho(t+1)}{\rho(t)} + 1 \right) \log \frac{\rho(t+1)}{\rho(t)}.$ $\qquad \square$

Now we are ready to prove Theorem E.3.

*Proof for Theorem E.3.* As discussed above, we only need to show a variant of Lemma C.12 for gradient descent: for every limit point $\bar{\boldsymbol{\theta}}$ of $\left\{ \hat{\boldsymbol{\theta}}(t) : t \geq 0 \right\}$, there exists a sequence of $\{t_m : m \in \mathbb{N}\}$ such that $t_m \uparrow +\infty, \hat{\boldsymbol{\theta}}(t_m) \to \bar{\boldsymbol{\theta}}$, and $\beta(t_m) \to 1$.

We only need to change the choices of $s_m, s'_m, t_m$ in the proof for Lemma C.12. We choose $s_m > t_{m-1}$ to be a time such that

$$\left\| \hat{\boldsymbol{\theta}}(s_m) - \bar{\boldsymbol{\theta}} \right\|_2 \leq \epsilon_m \quad \text{and} \quad \frac{1}{L} \log \left( \frac{\lim_{t \to +\infty} \hat{\gamma}(t)}{\hat{\gamma}(s_m)} \right) \leq \epsilon_m^3.$$

Then we let $s'_m > s_m$ be the minimum time such that $\log \rho(s'_m) \geq \log \rho(s_m) + \epsilon_m$. According to Theorem E.2, $s_m$ and $s'_m$ must exist. Finally, we construct $t_m \in \{s_m, \ldots, s'_m - 1\}$ to be a time that $\beta(t_m)^{-2} - 1 \leq \epsilon_m^2$, where the existence can be shown by (22).

To see that this construction meets our requirement, note that $\beta(t_m)^{-2} - 1 \leq \epsilon_m^2 \to 0$ and

$$\left\| \hat{\boldsymbol{\theta}}(t_m) - \bar{\boldsymbol{\theta}} \right\|_2 \leq \left\| \hat{\boldsymbol{\theta}}(t_m) - \hat{\boldsymbol{\theta}}(s_m) \right\|_2 + \left\| \hat{\boldsymbol{\theta}}(s_m) - \bar{\boldsymbol{\theta}} \right\|_2 \leq \left( \frac{B_1}{L \hat{\gamma}(t_0)} e^{\epsilon_m} + 1 \right) \cdot \epsilon_m + \epsilon_m \to 0,$$

where the last inequality is by Lemma E.11. $\qquad \square$

### E.6 PROOF FOR THEOREM E.5

*Proof.* By a similar analysis as Lemma D.4, we have

$$E(x) = \int_x^{\mathcal{L}(t_0)} \Theta \left( \frac{1}{u^2 (\log \frac{1}{u})^{2-2/L}} \right) du = \int_{1/\mathcal{L}(t_0)}^{1/x} \Theta \left( \frac{1}{(\log u)^{2-2/L}} \right) du$$

$$= \Theta \left( \frac{1}{x (\log \frac{1}{x})^{2-2/L}} \right).$$

We can also bound the inverse function $E^{-1}(y)$ by $\Theta\left(\frac{1}{y(\log y)^{2-2/L}}\right)$. With these, we can use a similar analysis as Theorem A.10 to prove Theorem E.5.

First, using a similar proof as for (17), we have $\rho^L = \Theta(\log\frac{1}{\mathcal{L}})$. So we only need to show $\mathcal{L}(t) = \Theta(\frac{1}{T(\log T)^{2-2/L}})$. With a similar analysis as for (P3) in Lemma E.9, we have the following bound for $\mathcal{L}(\tau+1) - \mathcal{L}(\tau)$:

$$\mathcal{L}(\tau+1) - \mathcal{L}(\tau) \geq -\eta(\tau)(1 + \mu(\mathcal{L}(\tau)))\|\nabla\mathcal{L}(\tau)\|_2^2.$$

Using the fact that $\mu \leq 1/2$, we have $\mathcal{L}(\tau+1) - \mathcal{L}(\tau) \geq -\frac{3}{2}\eta(\tau)\|\nabla\mathcal{L}(\tau)\|_2^2$. By Lemma E.7, $\|\nabla\mathcal{L}(\tau)\|_2 \leq 2C_\eta\kappa(\mathcal{L}(\tau))\mathcal{L}(\tau)$. Using a similar proof as for Lemma E.10, we can show that $E(\mathcal{L}(t)) \leq O(T)$. Combining this with Lemma E.10, we have $E(\mathcal{L}(t)) = \Theta(T)$. Therefore, $\mathcal{L}(t) = \Theta(\frac{1}{T(\log T)^{2-2/L}})$. $\qquad\square$

## F  GENERAL LOSS FUNCTIONS: GENERAL LOSS FUNCTIONS

It is worth to note that the above analysis can be extended to other loss functions. For this, we need to replace (B3) with a strong assumption (S3), which takes into account the second-order derivatives of $f$.

**(S3).** The loss function $\ell(q)$ can be expressed as $\ell(q) = e^{-f(q)}$ such that

(S3.1). $f : \mathbb{R} \to \mathbb{R}$ is $\mathcal{C}^2$-smooth.

(S3.2). $f'(q) > 0$ for all $q \in \mathbb{R}$.

(S3.3). There exists $b_f \geq 0$ such that $f'(q)q$ is non-decreasing for $q \in (b_f, +\infty)$, and $f'(q)q \to +\infty$ as $q \to +\infty$.

(S3.4). Let $g : [f(b_f), +\infty) \to [b_f, +\infty)$ be the inverse function of $f$ on the domain $[b_f, +\infty)$. There exists $p \geq 0$ such that for all $x > f(b_f), y > b_f$,

$$\left|\frac{g''(x)}{g'(x)}\right| \leq \frac{p}{x} \quad \text{and} \quad \left|\frac{f''(y)}{f'(y)}\right| \leq \frac{p}{y}.$$

It can be verified that (S3) is satisfied by exponential loss and logistic loss. Now we explain each of the assumptions in (S3). (S3.2) and (S3.3) are essentially the same as (B3.2) and (B3.3). (S3.1) and (B3.1) are the same except that (S3.1) assumes $f$ is $\mathcal{C}^2$-smooth rather than $\mathcal{C}^1$-smooth.

(S3.4) can also be written in the following form:

$$\left|\frac{d}{dx}\log g'(x)\right| \leq p \cdot \frac{d}{dx}\log x \quad \text{and} \quad \left|\frac{d}{dy}\log f'(y)\right| \leq p \cdot \frac{d}{dy}\log y. \tag{23}$$

That is, $\log g'(x)$ and $\log f'(y)$ grow no faster than $O(\log x)$ and $O(\log y)$, respectively. In fact, (B3.4) can be deduced from (S3.4). Recall that (B3.4) ensures that $\Theta(g'(x)) = g'(\Theta(x))$ and $\Theta(f'(y)) = f'(\Theta(y))$. Thus, (S3.4) also gives us the interchangeability between $f', g'$ and $\Theta$.

**Lemma F.1.** *(S3.4) implies (B3.4) with* $b_g = \max\{2f(b_f), f(2b_f)\}$ *and* $K = 2^p$.

*Proof.* Fix $x \in (b_g, +\infty), y \in (g(b_g), +\infty)$ and $\theta \in [1/2, 1)$. Integrating (23) on both sides of the inequalities from $\theta x$ to $x$ and $\theta y$ to $y$, we have

$$|\log g'(x) - \log g'(\theta x)| \leq p \cdot \log\frac{1}{\theta} \quad \text{and} \quad |\log f'(y) - \log f'(\theta y)| \leq p \cdot \log\frac{1}{\theta}.$$

Therefore, we have $g'(x) \leq \theta^{-p}g'(\theta x) \leq Kg'(\theta x)$ and $f'(y) \leq \theta^{-p}f'(\theta y) \leq Kf'(\theta x)$. $\qquad\square$

To extend our results to general loss functions, we need to redefine $\kappa(x), \lambda(x), \phi(x), C_\eta, H(x)$ in order. For $\kappa(x)$ and $\lambda(x)$, we redefine them as follows:

$$\kappa(x) := \sup\left\{\frac{w(\log\frac{1}{w})^{2-2/L}}{g'(\log\frac{1}{w})^2} : w \in (0, x]\right\} \qquad \lambda(x) := \frac{g'(\log\frac{1}{x})}{g(\log\frac{1}{x})}.$$

By Lemma D.1, $\frac{w(\log\frac{1}{w})^{2-2/L}}{g'(\log\frac{1}{w})^2} = O\left(w(\log\frac{1}{w})^{4-2/L}/g(\log\frac{1}{w})^2\right) \to 0$. So $\kappa(x)$ is well-defined. Using $\lambda(x)$, we can define $\phi(x)$ and $\hat{\gamma}(\boldsymbol{\theta})$ as follows.

$$\phi'(x) := -\sup\left\{\frac{\lambda(w)}{w}\left(1 + 2(1 + \lambda(w)/L)\mu(w)\right) : w \in [x, \mathcal{L}(t_0)]\right\}$$

$$\phi(x) := \log\left(g(\log\frac{1}{x})\right) + \int_0^x \left(\phi'(w) + \frac{\lambda(w)}{w}\right) dw$$

$$\hat{\gamma}(\boldsymbol{\theta}) := \frac{e^{\phi(\mathcal{L}(\boldsymbol{\theta}))}}{\rho^L}.$$

Using a similar argument as in Lemma E.1, we can show that $\hat{\gamma}(\boldsymbol{\theta})$ is well-defined and $\hat{\gamma}(\boldsymbol{\theta}) < \tilde{\gamma}(\boldsymbol{\theta}) := g(\log\frac{1}{\mathcal{L}})/\rho^L \le \bar{\gamma}(\boldsymbol{\theta})$. When $\mathcal{L}(\boldsymbol{\theta}) \to 0$, we also have $\hat{\gamma}(\boldsymbol{\theta})/\bar{\gamma}(\boldsymbol{\theta}) \to 1$.

For $C_\eta$, we define it to be the following.

$$C_\eta := \frac{1}{2}B_1\left(\left(\frac{B_0}{\hat{\gamma}(t_0)}\right)^p B_1 + \frac{2^{p+1}}{\log\frac{1}{\mathcal{L}(t_0)}}(pB_1 + B_2)\right)\left(\frac{B_0}{\hat{\gamma}(t_0)}\right)^p \min\left\{\hat{\gamma}(t_0)^{-2+2/L}, B_0^{-2+2/L}\right\}.$$

Finally, the definitions for $\mu(x) := (\log\frac{1}{\mathcal{L}(t_0)})/(2\log\frac{1}{x})$ and $H(x) := \mu(x)/(C_\eta\kappa(x))$ in (S5) remain unchanged except that $\kappa(x)$ and $C_\eta$ now use the new definitions.

Similar as gradient flow, we define $\nu(t) := \sum_{n=1}^N e^{-f(q_n(t))}f'(q_n(t))q_n(t)$. The key idea behind the above definitions is that we can prove similar bounds for $\nu(t), \|\nabla\mathcal{L}\|_2, \|\nabla^2\mathcal{L}\|_2$ as Corollary E.6 and Lemma E.7.

**Lemma F.2.** $\langle\boldsymbol{\theta}(t), -\nabla\mathcal{L}(t)\rangle = L\nu(t)$. If $\mathcal{L}(t) < e^{-f(b_f)}$, then $\nu(t) \ge \frac{\mathcal{L}(t)}{\lambda(\mathcal{L}(t))}$ and $\lambda(\mathcal{L}(t))$ has the lower bound $\lambda(\mathcal{L}(t)) \le 2^{p+1}\left(\log\frac{1}{\mathcal{L}(t)}\right)^{-1}$.

*Proof.* It can be easily proved by combining Theorem B.4, Lemma B.5 and Lemma D.1 together. $\square$

**Lemma F.3.** *For any $\boldsymbol{\theta}$, if $\mathcal{L}(\boldsymbol{\theta}) \le \mathcal{L}(t_0), \tilde{\gamma}(\boldsymbol{\theta}) \ge \hat{\gamma}(t_0)$, then*

$$\|\nabla\mathcal{L}(\boldsymbol{\theta})\|_2^2 \le 2C_\eta\kappa(\mathcal{L}(\boldsymbol{\theta}))\mathcal{L}(\boldsymbol{\theta}) \quad and \quad \left\|\nabla^2\mathcal{L}(\boldsymbol{\theta})\right\|_2 \le 2C_\eta\kappa(\mathcal{L}(\boldsymbol{\theta})).$$

*Proof.* Note that a direct corollary of (23) is that $f'(x_1) \le f'(x_2)(x_1/x_2)^p$ for $x_1 \ge x_2 > b_f$. So we have

$$f'(q_n) \le f'\left(g(\log\frac{1}{\mathcal{L}})\right)\left(\frac{q_n}{g(\log\frac{1}{\mathcal{L}})}\right)^p = \frac{1}{g'(\log\frac{1}{\mathcal{L}})}\left(\frac{q_n/\rho^L}{\tilde{\gamma}}\right)^p \le \frac{R}{g'(\log\frac{1}{\mathcal{L}})},$$

where $R := (B_0/\hat{\gamma}(t_0))^p$. Applying (S3.4), we can also deduce that

$$|f''(q_n)| \le \frac{p}{q_n} \cdot f'(q_n) \le \frac{pR}{g(\log\frac{1}{\mathcal{L}})g'(\log\frac{1}{\mathcal{L}})}.$$

Now we bound $\|\nabla\mathcal{L}\|_2$ and $\left\|\nabla^2\mathcal{L}\right\|_2$. By the chain rule, we have

$$\|\nabla\mathcal{L}\|_2 = \left\|-\sum_{n=1}^N e^{-f(q_n)}f'(q_n)\nabla q_n\right\|_2 \le B_1R \cdot \frac{\mathcal{L}\rho^{L-1}}{g'(\log\frac{1}{\mathcal{L}})}$$

$$\left\|\nabla^2\mathcal{L}\right\|_2 = \left\|\sum_{n=1}^N e^{-f(q_n)}\left((f'(q_n)^2 - f''(q_n))\nabla q_n\nabla q_n^\top - f'(q_n)\nabla^2 q_n\right)\right\|_2$$

$$\le \sum_{n=1}^N e^{-f(q_n)}\left(\left(\frac{R^2}{g'(\log\frac{1}{\mathcal{L}})^2} + \frac{pR}{g(\log\frac{1}{\mathcal{L}})g'(\log\frac{1}{\mathcal{L}})}\right)B_1^2\rho^{2L-2} + \frac{R}{g'(\log\frac{1}{\mathcal{L}})}B_2\rho^{L-2}\right)$$

$$\le \frac{\mathcal{L}\rho^{2L-2}}{g'(\log\frac{1}{\mathcal{L}})^2}\left((R + p\lambda(\mathcal{L}))B_1^2 + \frac{g'(\log\frac{1}{\mathcal{L}})}{\rho^L}B_2\right)R.$$

For $\rho$ we have $\hat{\gamma}(t_0)\rho^L \leq \log\frac{1}{\mathcal{L}} \leq B_0\rho^L$, so we can bound $\rho^{2L-2}$ by $\rho^{2L-2} \leq M(\log\frac{1}{\mathcal{L}})^{2L-2}$, where $M := \min\left\{\hat{\gamma}(t_0)^{-2+2/L}, B_0^{-2+2/L}\right\}$. By Lemma F.2, we have $\lambda(\mathcal{L}) \leq 2^{p+1}\frac{1}{\log\frac{1}{\mathcal{L}}} \leq \frac{2^{p+1}}{\log\frac{1}{\mathcal{L}(t_0)}}$, and also $\frac{g'(\log\frac{1}{\mathcal{L}})}{\rho^L} = \lambda(\mathcal{L})\cdot\tilde{\gamma} \leq \frac{2^{p+1}B_1}{\log\frac{1}{\mathcal{L}(t_0)}}$. Combining all these formulas together gives

$$\|\nabla\mathcal{L}\|_2^2 \leq B_1^2 R^2 M \frac{\mathcal{L}^2}{g'(\log\frac{1}{\mathcal{L}})^2}\left(\log\frac{1}{\mathcal{L}}\right)^{2-2/L} \leq 2C_\eta\kappa(\mathcal{L})\mathcal{L}$$

$$\|\nabla^2\mathcal{L}\|_2 \leq \left(\left(R + \frac{p2^{p+1}}{\log\frac{1}{\mathcal{L}(t_0)}}\right)B_1^2 + \frac{2^{p+1}B_1B_2}{\log\frac{1}{\mathcal{L}(t_0)}}\right)RM\frac{\mathcal{L}}{g'(\log\frac{1}{\mathcal{L}})^2}\left(\log\frac{1}{\mathcal{L}}\right)^{2-2/L} \leq 2C_\eta\kappa(\mathcal{L}),$$

which completes the proof. $\qquad\qquad\qquad\qquad\qquad\qquad\qquad\qquad\qquad\qquad\qquad\qquad\qquad\quad\square$

With Corollary E.6 and Lemma E.7, we can prove Lemma E.8 with exactly the same argument. Then Theorem E.2, E.3, E.4 can also be proved similarly. For Theorem E.5, we can follow the argument for gradient flow to show that it holds with slightly different tight bounds:

**Theorem F.4.** *Under assumptions (S1), (A2), (S3), (A4), (S5), we have the following tight rates for training loss and weight norm:*

$$\mathcal{L}(t) = \Theta\left(\frac{g(\log T)^{2/L}}{T(\log T)^2}\right) \quad and \quad \rho(t) = \Theta(g(\log T)^{1/L}),$$

*where $T = \sum_{\tau=t_0}^{t-1}\eta(\tau)$.*

## G  EXTENSION: MULTI-CLASS CLASSIFICATION

In this section, we generalize our results to multi-class classification with cross-entropy loss. This part of analysis is inspired by Theorem 1 in (Zhang et al., 2019), which gives a lower bound for the gradient in terms of the loss $\mathcal{L}$.

Since now a neural network has multiple outputs, we need to redefine our notations. Let $C$ be the number of classes. The output of a neural network $\Phi$ is a vector $\boldsymbol{\Phi}(\boldsymbol{\theta};\boldsymbol{x}) \in \mathbb{R}^C$. We use $\Phi_j(\boldsymbol{\theta};\boldsymbol{x}) \in \mathbb{R}$ to denote the $j$-th output of $\Phi$ on the input $\boldsymbol{x} \in \mathbb{R}^{d_\times}$. A dataset is denoted by $\mathcal{D} = \{\boldsymbol{x}_n, y_n\}_{n=1}^N = \{(\boldsymbol{x}_n, y_n) : n \in [N]\}$, where $\boldsymbol{x}_n \in \mathbb{R}^{d_\times}$ is a data input and $y_n \in [C]$ is the corresponding label. The loss function of $\Phi$ on the dataset $\mathcal{D}$ is defined as

$$\mathcal{L}(\boldsymbol{\theta}) := \sum_{n=1}^N -\log\frac{e^{\Phi_{y_n}(\boldsymbol{\theta};\boldsymbol{x}_n)}}{\sum_{j=1}^C e^{\Phi_j(\boldsymbol{\theta};\boldsymbol{x}_n)}}.$$

The *margin* for a single data point $(\boldsymbol{x}_n, y_n)$ is defined to be $q_n(\boldsymbol{\theta}) := \Phi_{y_n}(\boldsymbol{\theta};\boldsymbol{x}_n) - \max_{j\neq y_n}\{\Phi_j(\boldsymbol{\theta};\boldsymbol{x}_n)\}$, and the margin for the entire dataset is defined to be $q_{\min}(\boldsymbol{\theta}) = \min_{n\in[N]}q_n(\boldsymbol{\theta})$. We define the *normalized margin* to be $\bar{\gamma}(\boldsymbol{\theta}) := q_{\min}(\hat{\boldsymbol{\theta}}) = q_{\min}(\boldsymbol{\theta})/\rho^L$, where $\rho := \|\boldsymbol{\theta}\|_2$ and $\hat{\boldsymbol{\theta}} := \boldsymbol{\theta}/\rho \in \mathcal{S}^{d-1}$ as usual.

Let $\ell(q) := \log(1+e^{-q})$ be the logistic loss. Recall that $\ell(q)$ satisfies (B3). Let $f(q) = -\log\ell(q) = -\log\log(1+e^{-q})$. Let $g$ be the inverse function of $f$. So $g(q) = -\log(e^{e^{-q}} - 1)$.

The cross-entropy loss can be rewritten in other ways. Let $s_{nj} := \Phi_{y_n}(\boldsymbol{\theta};\boldsymbol{x}_n) - \Phi_j(\boldsymbol{\theta};\boldsymbol{x}_n)$. Let $\tilde{q}_n := -\mathrm{LSE}(\{-s_{nj} : j \neq y_n\}) = -\log\left(\sum_{j\neq y_n}e^{-s_{nj}}\right)$. Then

$$\mathcal{L}(\boldsymbol{\theta}) := \sum_{n=1}^N \log\left(1 + \sum_{j\neq y_n}e^{-s_{nj}}\right) = \sum_{n=1}^N \log(1 + e^{-\tilde{q}_n}) = \sum_{n=1}^N \ell(\tilde{q}_n) = \sum_{n=1}^N e^{-f(\tilde{q}_n)}.$$

**Gradient Flow.**  For gradient flow, we assume the following:

**(M1).** (Regularity). For any fixed $\boldsymbol{x}$ and $j \in [C]$, $\Phi_j(\cdot\,;\boldsymbol{x})$ is locally Lipschitz and admits a chain rule;

**(M2).** (Homogeneity). There exists $L > 0$ such that $\forall j \in [C], \forall \alpha > 0 : \Phi_j(\alpha\boldsymbol{\theta}; \boldsymbol{x}) = \alpha^L \Phi_j(\boldsymbol{\theta}; \boldsymbol{x})$;

**(M3).** (Cross-entropy Loss). $\mathcal{L}(\boldsymbol{\theta})$ is defined as the cross-entropy loss on the training set;

**(M4).** (Separability). There exists a time $t_0$ such that $\mathcal{L}(t_0) < \log 2$.

If $\mathcal{L} < \log 2$, then $\sum_{j \neq y_n} e^{-s_{nj}} < 1$ for all $n \in [N]$, and thus $s_{nj} > 0$ for all $n \in [N], j \in [C]$. So (M4) ensures the separability of training data.

**Definition G.1.** For cross-entropy loss, the smoothed normalized margin $\tilde{\gamma}(\boldsymbol{\theta})$ of $\boldsymbol{\theta}$ is defined as

$$\tilde{\gamma}(\boldsymbol{\theta}) := \frac{\ell^{-1}(\mathcal{L})}{\rho^L} = \frac{-\log(e^{\mathcal{L}} - 1)}{\rho^L},$$

where $\ell^{-1}(\cdot)$ is the inverse function of the logistic loss $\ell(\cdot)$.

Theorem 4.1 and 4.4 still hold. Here we redefine the optimization problem (P) to be

$$\min \quad \frac{1}{2} \|\boldsymbol{\theta}\|_2^2$$
$$\text{s.t.} \quad s_{nj}(\boldsymbol{\theta}) \geq 1 \qquad \forall n \in [N], \forall j \in [C] \setminus \{y_n\}$$

Most of our proofs are very similar as before. Here we only show the proof for the generalized version of Lemma 5.1.

**Lemma G.2.** *Lemma 5.1 is also true for the smoothed normalized margin $\tilde{\gamma}$ defined in Definition G.1.*

*Proof.* Define $\nu(t)$ by the following formula:

$$\nu(t) := \sum_{n=1}^{N} \frac{\sum_{j \neq y_n} e^{-s_{nj}} s_{nj}}{1 + \sum_{j \neq y_n} e^{-s_{nj}}}.$$

Using a similar argument as in Theorem B.4, it can be proved that $\frac{1}{2}\frac{d\rho^2}{dt} = L\nu(t)$ for a.e. $t > 0$.

It can be shown that Lemma B.5, which asserts that $\nu(t) \geq \frac{g(\log \frac{1}{\mathcal{L}})}{g'(\log \frac{1}{\mathcal{L}})}\mathcal{L}$, still holds for this new definition of $\nu(t)$. By definition, $s_{nj} \geq q_n$ for $j \neq y_n$. Also note that $e^{-\tilde{q}_n} \geq e^{-q_n}$. So $s_{nj} \geq q_n \geq \tilde{q}_n$. Then we have

$$\nu(t) \geq \sum_{n=1}^{N} \frac{\sum_{j \neq y_n} e^{-s_{nj}}}{1 + \sum_{j \neq y_n} e^{-s_{nj}}} \cdot \tilde{q}_n = \sum_{n=1}^{N} \frac{e^{-\tilde{q}_n}}{1 + e^{-\tilde{q}_n}} \cdot \tilde{q}_n = \sum_{n=1}^{N} e^{-f(\tilde{q}_n)} f'(\tilde{q}_n) \tilde{q}_n.$$

Note that $\mathcal{L} = \sum_{n=1}^{N} e^{-f(\tilde{q}_n)}$. Then using Lemma B.5 for logistic loss can conclude that $\nu(t) \geq \frac{g(\log \frac{1}{\mathcal{L}})}{g'(\log \frac{1}{\mathcal{L}})}\mathcal{L}$.

The rest of the proof for this lemma is exactly the same as that for Lemma 5.1. $\square$

**Gradient Descent.** For gradient descent, we only need to replace (M1) with (S1) and make the assumption (S5) on the learning rate.

**(S1).** (Smoothness). For any fixed $\boldsymbol{x}$ and $j \in [C]$, $\Phi_j(\cdot; \boldsymbol{x})$ is $\mathcal{C}^2$-smooth;

**(S5).** (Learing rate condition). $\sum_{t \geq 0} \eta(t) = +\infty$ and $\eta(t) \leq H(\mathcal{L}(\boldsymbol{\theta}(t)))$.

Here $H(x) := \mu(x)/(C_\eta \kappa(x))$ is defined to be the same as in Appendix F (when $\ell(\cdot)$ is set to logistic loss) except that we use another $C_\eta$ which will be specified later.

We only need to show that Lemma F.2 and Lemma F.3 continue to hold. Using the same argument as we do for gradient flow, we can show that $\gamma(t)$ and $\lambda(x)$ do satisfy the propositions in Lemma F.2.

For Lemma F.3, we first note the original definition of $C_\eta$ involves $B_0, B_1, B_2$, which are undefined in the multi-class setting. So now we redefine them as

$$B_0 := \sup\left\{s_{nj} : \boldsymbol{\theta} \in \mathcal{S}^{d-1}, n \in [N], j \in [C]\right\},$$
$$B_1 := \sup\left\{\left\|\nabla s_{nj}\right\|_2 : \boldsymbol{\theta} \in \mathcal{S}^{d-1}, n \in [N], j \in [C]\right\},$$
$$B_2 := \sup\left\{\left\|\nabla^2 s_{nj}\right\|_2 : \boldsymbol{\theta} \in \mathcal{S}^{d-1}, n \in [N], j \in [C]\right\}.$$

By the property of LSE, we can use $B_0, B_1, B_2$ to bound $\tilde{q}_n, \nabla \tilde{q}_n, \nabla^2 \tilde{q}_n$.

$$\tilde{q}_n \leq q_n \leq B_0 \rho^L,$$

$$\left\|\nabla \tilde{q}_n\right\|_2 = \left\|\sum_{j=1}^C \frac{e^{-s_{nj}}}{\sum_{k=1}^C e^{-s_{nk}}} \nabla s_{nj}\right\|_2 \leq \sum_{j=1}^C \frac{e^{-s_{nj}}}{\sum_{k=1}^C e^{-s_{nk}}} \left\|\nabla s_{nj}\right\|_2 \leq B_1 \rho^{L-1},$$

$$\left\|\nabla^2 \tilde{q}_n\right\|_2 = \left\|\sum_{j=1}^C \left(\frac{e^{-s_{nj}}}{\sum_{k=1}^C e^{-s_{nk}}} (\nabla^2 s_{nj} - \nabla s_{nj} \nabla s_{nj}^\top) + \frac{e^{-2s_{nj}}}{\left(\sum_{k=1}^C e^{-s_{nk}}\right)^2} \nabla s_{nj} \nabla s_{nj}^\top\right)\right\|_2$$
$$\leq B_2 \rho^{L-2} + 2B_1^2 \rho^{2L-2}.$$

Using these bounds, we can prove with the constant $C_\eta$ defined as follows:

$$C_\eta := \frac{1}{2}\left(\left(\left(\frac{B_0}{\hat{\gamma}(t_0)}\right)^p + \frac{p2^{p+1}}{\log \frac{1}{\mathcal{L}(t_0)}}\right) B_1^2 + (2\log 2)\left(\frac{B_2}{\rho(t_0)^L} + 2B_1^2\right)\right)\left(\frac{B_0}{\hat{\gamma}(t_0)}\right)^p M,$$

where $M := \min\left\{\hat{\gamma}(t_0)^{-2+2/L}, B_0^{-2+2/L}\right\}$.

## H EXTENSION: MULTI-HOMOGENEOUS MODELS

In this section, we extend our results to multi-homogeneous models. For this, the main difference from the proof for homogeneous models is that now we have to separate the norm of each homogeneous parts of the parameter, rather than consider them as a whole. So only a small part to proof needs to be changed. We focus on gradient flow, but it is worth to note that following the same argument, it is not hard to extend the results to gradient descent.

Let $\Phi(\boldsymbol{w}_1, \ldots, \boldsymbol{w}_m; \boldsymbol{x})$ be $(k_1, \ldots, k_m)$-homogeneous. Let $\rho_i = \left\|\boldsymbol{w}_i\right\|_2$ and $\hat{\boldsymbol{w}}_i = \frac{\boldsymbol{w}_i}{\|\boldsymbol{w}_i\|_2}$. The smoothed normalized margin defined in (5) can be rewritten as follows:

**Definition H.1.** For a multi-homogeneous model with loss function $\ell(\cdot)$ satisfying (B3), the smoothed normalized margin $\tilde{\gamma}(\boldsymbol{\theta})$ of $\boldsymbol{\theta}$ is defined as

$$\tilde{\gamma}(\boldsymbol{\theta}) := \frac{g(\log \frac{1}{\mathcal{L}})}{\prod_{i=1}^m \rho_i^{k_i}} = \frac{\ell^{-1}(\mathcal{L})}{\prod_{i=1}^m \rho_i^{k_i}}.$$

We only prove the generalized version of Lemma 5.1 here. The other proofs are almost the same.

**Lemma H.2.** For all $t > t_0$,

$$\frac{d}{dt}\log \rho_i > 0 \quad \text{for all } i \in [m] \quad \text{and} \quad \frac{d}{dt}\log \tilde{\gamma} \geq \sum_{i=1}^m k_i \left(\frac{d}{dt}\log \rho_i\right)^{-1} \left\|\frac{d\hat{\boldsymbol{w}}_i}{dt}\right\|_2^2. \quad (24)$$

*Proof.* Note that $\frac{d}{dt}\log \rho_i = \frac{1}{2\rho_i^2}\frac{d\rho_i^2}{dt} = \frac{k_i \nu(t)}{\rho_i^2}$ by Theorem B.4. It simply follows from Lemma B.5 that $\frac{d}{dt}\log \rho > 0$ for a.e. $t > t_0$. And it is easy to see that $\log \tilde{\gamma} = \log\left(g(\log \frac{1}{\mathcal{L}})/\rho^L\right)$ exists for all

$t \geq t_0$. By the chain rule and Lemma B.5, we have

$$\frac{d}{dt} \log \tilde{\gamma} = \frac{d}{dt} \left( \log \left( g(\log \frac{1}{\mathcal{L}}) \right) - \sum_{i=1}^{m} k_i \log \rho_i \right) = \frac{g'(\log \frac{1}{\mathcal{L}})}{g(\log \frac{1}{\mathcal{L}})} \cdot \frac{1}{\mathcal{L}} \cdot \left( -\frac{d\mathcal{L}}{dt} \right) - \sum_{i=1}^{m} \frac{k_i^2 \nu(t)}{\rho_i^2}$$

$$\geq \frac{1}{\nu(t)} \cdot \left( -\frac{d\mathcal{L}}{dt} \right) - \sum_{i=1}^{m} \frac{k_i^2 \nu(t)}{\rho_i^2}$$

$$\geq \frac{1}{\nu(t)} \cdot \left( -\frac{d\mathcal{L}}{dt} - \sum_{i=1}^{m} \frac{k_i^2 \nu(t)^2}{\rho_i^2} \right).$$

On the one hand, $-\frac{d\mathcal{L}}{dt} = \sum_{i=1}^{m} \left\| \frac{d\boldsymbol{w}_i}{dt} \right\|_2^2$ for a.e. $t > 0$ by Lemma I.3; on the other hand, $k_i \nu(t) = \left\langle \boldsymbol{w}_i, \frac{d\boldsymbol{w}_i}{dt} \right\rangle$ by Theorem B.4. Combining these together yields

$$\frac{d}{dt} \log \tilde{\gamma} \geq \frac{1}{\nu(t)} \sum_{i=1}^{m} \left( \left\| \frac{d\boldsymbol{w}_i}{dt} \right\|_2^2 - \left\langle \hat{\boldsymbol{w}}_i, \frac{d\boldsymbol{w}_i}{dt} \right\rangle^2 \right) = \frac{1}{\nu(t)} \left\| (\boldsymbol{I} - \hat{\boldsymbol{w}}_i \hat{\boldsymbol{w}}_i^\top) \frac{d\boldsymbol{w}_i}{dt} \right\|_2^2.$$

By the chain rule, $\frac{d\hat{\boldsymbol{w}}_i}{dt} = \frac{1}{\rho_i} (\boldsymbol{I} - \hat{\boldsymbol{w}}_i \hat{\boldsymbol{w}}_i^\top) \frac{d\boldsymbol{w}_i}{dt}$ for a.e. $t > 0$. So we have

$$\frac{d}{dt} \log \tilde{\gamma} \geq \sum_{i=1}^{m} \frac{\rho_i^2}{\nu(t)} \left\| \frac{d\hat{\boldsymbol{w}}_i}{dt} \right\|_2^2 = \sum_{i=1}^{m} k_i \left( \frac{d}{dt} \log \rho_i \right)^{-1} \left\| \frac{d\hat{\boldsymbol{w}}_i}{dt} \right\|_2^2.$$

$\square$

For cross-entropy loss, we can combine the proofs in Appendix G to show that Lemma H.2 holds if we use the following definition of the smoothed normalized margin:

**Definition H.3.** For a multi-homogeneous model with cross-entropy, the smoothed normalized margin $\tilde{\gamma}(\boldsymbol{\theta})$ of $\boldsymbol{\theta}$ is defined as

$$\tilde{\gamma} = \frac{\ell^{-1}(\mathcal{L})}{\prod_{i=1}^{m} \rho_i^{k_i}} = \frac{-\log(e^{\mathcal{L}} - 1)}{\prod_{i=1}^{m} \rho_i^{k_i}}.$$

where $\ell^{-1}(\cdot)$ is the inverse function of the logistic loss $\ell(\cdot)$.

The only place we need to change in the proof for Lemma H.2 is that instead of using Lemma B.5, we need to prove $\nu(t) \geq \frac{g(\log \frac{1}{\mathcal{L}})}{g'(\log \frac{1}{\mathcal{L}})} \mathcal{L}$ in a similar way as in Lemma G.2. The other parts of the proof are exactly the same as before.

# I    CHAIN RULES FOR NON-DIFFERENTIABLE FUNCTIONS

In this section, we provide some background on the chain rule for non-differentiable functions. The ordinary chain rule for differentiable functions is a very useful formula for computing derivatives in calculus. However, for non-differentiable functions, it is difficult to find a natural definition of subdifferential so that the chain rule equation holds exactly. To solve this issue, Clarke proposed Clarke's subdifferential (Clarke, 1975; 1990; Clarke et al., 2008) for locally Lipschitz functions, for which the chain rule holds as an inclusion rather than an equation:

**Theorem I.1** (Theorem 2.3.9 and 2.3.10 of (Clarke, 1990)). *Let* $z_1, \ldots, z_n : \mathbb{R}^d \to \mathbb{R}$ *and* $f : \mathbb{R}^n \to \mathbb{R}$ *be locally Lipschitz functions. Let* $(f \circ \boldsymbol{z})(\boldsymbol{x}) = f(z_1(\boldsymbol{x}), \ldots, z_n(\boldsymbol{x}))$ *be the composition of* $f$ *and* $\boldsymbol{z}$. *Then,*

$$\partial^\circ (f \circ \boldsymbol{z})(\boldsymbol{x}) \subseteq \text{conv} \left\{ \sum_{i=1}^{n} \alpha_i \boldsymbol{h}_i : \boldsymbol{\alpha} \in \partial^\circ f(z_1(\boldsymbol{x}), \ldots, z_n(\boldsymbol{x})), \boldsymbol{h}_i \in \partial^\circ z_i(\boldsymbol{x}) \right\}.$$

For analyzing gradient flow, the chain rule is crucial. For a differentiable loss function $\mathcal{L}(\boldsymbol{\theta})$, we can see from the chain rule that the function value keeps decreasing along the gradient flow $\frac{d\boldsymbol{\theta}(t)}{dt} = -\nabla \mathcal{L}(\boldsymbol{\theta}(t))$:

$$\frac{d\mathcal{L}(\boldsymbol{\theta}(t))}{dt} = \left\langle \nabla \mathcal{L}(\boldsymbol{\theta}(t)), \frac{d\boldsymbol{\theta}(t)}{dt} \right\rangle = - \left\| \frac{d\boldsymbol{\theta}(t)}{dt} \right\|_2^2. \tag{25}$$

But for locally Lipschitz functions which could be non-differentiable, (25) may not hold in general since Theorem I.1 only holds for an inclusion.

Following (Davis et al., 2020; Drusvyatskiy et al., 2015), we consider the functions that admit a chain rule for any arc.

**Definition I.2** (Chain Rule). A locally Lipschitz function $f : \mathbb{R}^d \to \mathbb{R}$ admits a chain rule if for any arc $\boldsymbol{z} : [0, +\infty) \to \mathbb{R}^d$,

$$\forall \boldsymbol{h} \in \partial^\circ f(z(t)) : (f \circ \boldsymbol{z})'(t) = \langle \boldsymbol{h}, \boldsymbol{z}'(t) \rangle \tag{26}$$

holds for a.e. $t > 0$.

It is shown in (Davis et al., 2020; Drusvyatskiy et al., 2015) that a generalized version of (25) holds for such functions:

**Lemma I.3** (Lemma 5.2 (Davis et al., 2020)). *Let $\mathcal{L} : \mathbb{R}^d \to \mathbb{R}$ be a locally Lipschitz function that admits a chain rule. Let $\boldsymbol{\theta} : [0, +\infty) \to \mathbb{R}^d$ be the gradient flow on $\mathcal{L}$:*

$$\frac{d\boldsymbol{\theta}(t)}{dt} \in -\partial^\circ \mathcal{L}(\boldsymbol{\theta}(t)) \qquad \text{for a.e. } t \geq 0,$$

*Then*

$$\frac{d\mathcal{L}(\boldsymbol{\theta}(t))}{dt} = - \left\| \frac{d\boldsymbol{\theta}(t)}{dt} \right\|_2^2 = -\min\{\|\boldsymbol{h}\|_2^2 : \boldsymbol{h} \in \partial^\circ \mathcal{L}(\boldsymbol{\theta}(t))\}$$

*holds for a.e. $t > 0$.*

We can see that $\mathcal{C}^1$-smooth functions admit chain rules. As shown in (Davis et al., 2020), if a locally Lipschitz function is subdifferentiablly regular or Whitney $\mathcal{C}^1$-stratifiable, then it admits a chain rule. The latter one includes a large family of functions, e.g., semi-algebraic functions, semi-analytic functions, and definable functions in an o-minimal structure (Coste, 2002; van den Dries & Miller, 1996).

It is worth noting that the class of functions that admits chain rules is closed under composition. This is indeed a simple corollary of Theorem I.1.

**Theorem I.4.** *Let $z_1, \ldots, z_n : \mathbb{R}^d \to \mathbb{R}$ and $f : \mathbb{R}^n \to \mathbb{R}$ be locally Lipschitz functions and assume all of them admit chain rules. Let $(f \circ \boldsymbol{z})(\boldsymbol{x}) = f(z_1(\boldsymbol{x}), \ldots, z_n(\boldsymbol{x}))$ be the composition of $f$ and $\boldsymbol{z}$. Then $f \circ \boldsymbol{z}$ also admits a chain rule.*

*Proof.* We can see that $f \circ \boldsymbol{z}$ is locally Lipschitz. Let $\boldsymbol{x} : [0, +\infty) \to \mathbb{R}^d, t \mapsto \boldsymbol{x}(t)$ be an arc. First, we show that $\boldsymbol{z} \circ \boldsymbol{x} : [0, +\infty) \to \mathbb{R}^d, t \mapsto \boldsymbol{z}(\boldsymbol{x}(t))$ is also an arc. For any closed sub-interval $I$, $\boldsymbol{z}(\boldsymbol{x}(I))$ must be contained in a compact set $U$. Then it can be shown that the locally Lipschitz continuous function $\boldsymbol{z}$ is (globally) Lipschitz continuous on $U$. By the fact that the composition of a Lipschitz continuous and an absolutely continuous function is absolutely continuous, $\boldsymbol{z} \circ \boldsymbol{x}$ is absolutely continuous on $I$, and thus it is an arc.

Since $f$ and $\boldsymbol{z}$ admit chain rules on arcs $\boldsymbol{z} \circ \boldsymbol{x}$ and $\boldsymbol{x}$ respectively, the following holds for a.e. $t > 0$,

$$\forall \boldsymbol{\alpha} \in \partial^\circ f(\boldsymbol{z}(\boldsymbol{x}(t))) : \quad (f \circ (\boldsymbol{z} \circ \boldsymbol{x}))'(t) = \langle \boldsymbol{\alpha}, (\boldsymbol{z} \circ \boldsymbol{x})'(t) \rangle,$$
$$\forall \boldsymbol{h}_i \in \partial^\circ z_i(\boldsymbol{x}(t)) : \quad (z_i \circ \boldsymbol{x})'(t) = \langle \boldsymbol{h}_i, \boldsymbol{x}'(t) \rangle.$$

Combining these we obtain that for a.e. $t > 0$,

$$(f \circ \boldsymbol{z} \circ \boldsymbol{x})'(t) = \sum_{i=1}^n \alpha_i \langle \boldsymbol{h}_i, \boldsymbol{x}'(t) \rangle,$$

for all $\boldsymbol{\alpha} \in \partial^\circ f(\boldsymbol{z}(\boldsymbol{x}(t)))$ and for all $\boldsymbol{h}_i \in \partial^\circ z_i(\boldsymbol{x}(t))$. The RHS can be rewritten as $\langle \sum_{i=1}^n \alpha_i \boldsymbol{h}_i, \boldsymbol{x}'(t) \rangle$. By Theorem I.1, every $\boldsymbol{k} \in \partial^\circ (f \circ \boldsymbol{z})(\boldsymbol{x}(t))$ can be written as a convex combination of a finite set of points in the form of $\sum_{i=1}^n \alpha_i \boldsymbol{h}_i$. So $(f \circ \boldsymbol{z} \circ \boldsymbol{x})'(t) = \langle \boldsymbol{k}, \boldsymbol{x}'(t) \rangle$ holds for a.e. $t > 0$. $\qquad \square$

## J    MEXICAN HAT

In this section, we give an example to illustrate that gradient flow does not necessarily converge in direction, even for $C^\infty$-smooth homogeneous models.

It is known that gradient flow (or gradient descent) may not converge to any point even when optimizing an $C^\infty$ function (Curry, 1944; Zoutendijk, 1976; Palis & De Melo, 2012; Absil et al., 2005). One famous counterexample is the "Mexican Hat" function described in (Absil et al., 2005):

$$f(u,v) := f(r\cos\varphi, r\sin\varphi) := \begin{cases} e^{-\frac{1}{1-r^2}}\left(1 - C(r)\sin\left(\varphi - \frac{1}{1-r^2}\right)\right) & r < 1 \\ 0 & r \geq 1 \end{cases}$$

where $C(r) = \frac{4r^4}{4r^4+(1-r^2)^4} \in [0,1]$. It can be shown that $f$ is $C^\infty$-smooth on $\mathbb{R}^2$ but not analytic. See Figure 2 for a plot for $f(u,v)$.

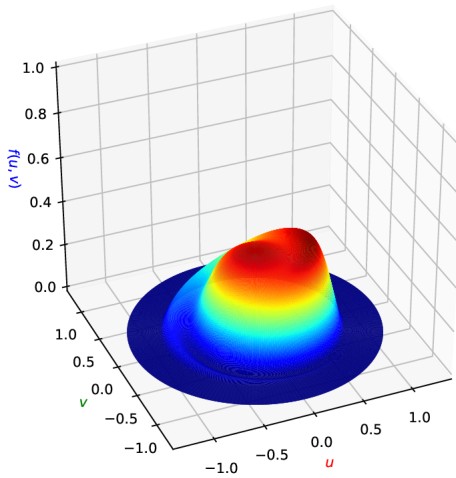

Figure 2: A plot for the Mexican Hat function $f(u,v)$.

However, the Maxican Hat function is not homogeneous, and Absil et al. (2005) did not consider the directional convergence, either. To make it homogeneous, we introduce an extra variable $z$, and normalize the parameter before evaluate $f$. In particular, we fix $L > 0$ and define

$$h(\boldsymbol{\theta}) = h(x,y,z) = \rho^L(1 - f(u,v)) \qquad \text{where} \quad u = x/\rho, \quad v = y/\rho, \quad \rho = \sqrt{x^2+y^2+z^2}.$$

We can show the following theorem.

**Theorem J.1.** *Consider gradient flow on $\mathcal{L}(\boldsymbol{\theta}) = \sum_{n=1}^N e^{-q_n(\boldsymbol{\theta})}$, where $q_n(\boldsymbol{\theta}) = h(\boldsymbol{\theta})$ for all $n \in [N]$. Suppose the polar representation of $(u,v)$ is $(r\cos\varphi, r\sin\varphi)$. If $0 < r < 1$ and $\varphi = \frac{1}{1-r^2}$ holds at time $t = 0$, then $\frac{\boldsymbol{\theta}(t)}{\|\boldsymbol{\theta}(t)\|_2}$ does not converge to any point, and the limit points of $\{\frac{\boldsymbol{\theta}(t)}{\|\boldsymbol{\theta}(t)\|_2} : t > 0\}$ form a circle $\{(x,y,z) \in \mathcal{S}^2 : x^2 + y^2 = 1, z = 0\}$.*

*Proof.* Define $\psi = \varphi - \frac{1}{1-r^2}$. Our proof consists of two parts, following from the idea in (Absil et al., 2005). First, we show that $\frac{d\psi}{dt} = 0$ as long as $\psi = 0$. Then we can infer that $\psi = 0$ for all $t \geq 0$. Next, we show that $r \to 1$ as $t \to +\infty$. Using $\psi = 0$, we know that the polar angle $\varphi \to +\infty$ as $t \to +\infty$. Therefore, $(u,v)$ circles around $\{(u,v) : u^2 + v^2 = 1\}$, and thus it does not converge.

**Proof for $\frac{d\psi}{dt} = 0$.** For convenience, we use $w$ to denote $z/\rho$. By simple calculation, we have the following formulas for partial derivatives:

$$\begin{cases} \frac{\partial h}{\partial x} = \rho^{L-1}\left(Lu(1-f) - (1-u^2)\frac{\partial f}{\partial u} + uv\frac{\partial f}{\partial v}\right) \\ \frac{\partial h}{\partial y} = \rho^{L-1}\left(Lv(1-f) + uv\frac{\partial f}{\partial u} - (1-v^2)\frac{\partial f}{\partial v}\right) \\ \frac{\partial h}{\partial z} = \rho^{L-1}\left(Lw(1-f) + uw\frac{\partial f}{\partial u} + vw\frac{\partial f}{\partial v}\right) \end{cases} \qquad \begin{cases} \frac{\partial f}{\partial u} = \frac{u}{r}\cdot\frac{\partial f}{\partial r} - \frac{v}{r^2}\cdot\frac{\partial f}{\partial \varphi} \\ \frac{\partial f}{\partial v} = \frac{v}{r}\cdot\frac{\partial f}{\partial r} + \frac{u}{r^2}\cdot\frac{\partial f}{\partial \varphi} \end{cases}$$

For gradient flow, we have

$$\frac{d\boldsymbol{\theta}}{dt} = Ne^{-h}\nabla h$$

$$\frac{du}{dt} = \frac{1}{\rho}\left((1-u^2)\frac{dx}{dt} - uv\frac{dy}{dt} - uw\frac{dz}{dt}\right) = Ne^{-h}\rho^{L-2}\left(-(1-u^2)\frac{\partial f}{\partial u} + uv\frac{\partial f}{\partial v}\right)$$

$$\frac{dv}{dt} = \frac{1}{\rho}\left((1-v^2)\frac{dy}{dt} - uv\frac{dx}{dt} - vw\frac{dz}{dt}\right) = Ne^{-h}\rho^{L-2}\left(-(1-v^2)\frac{\partial f}{\partial v} + uv\frac{\partial f}{\partial u}\right)$$

By writing down the movement of $(u,v)$ in the polar coordinate system, we have

$$\frac{dr}{dt} = \frac{u}{r}\cdot\frac{du}{dt} + \frac{v}{r}\cdot\frac{dv}{dt}$$

$$= -\frac{Ne^{-h}\rho^{L-2}}{r}\left(u(1-r^2)\frac{\partial f}{\partial u} + v(1-r^2)\frac{\partial f}{\partial v}\right)$$

$$= -Ne^{-h}\rho^{L-2}(1-r^2)\frac{\partial f}{\partial r}$$

$$\frac{d\varphi}{dt} = -\frac{v}{r^2}\cdot\frac{du}{dt} + \frac{u}{r^2}\cdot\frac{dv}{dt}$$

$$= \frac{Ne^{-h}\rho^{L-2}}{r^2}\left(v\frac{\partial f}{\partial u} - u\frac{\partial f}{\partial v}\right)$$

$$= -Ne^{-h}\rho^{L-2}\frac{1}{r^2}\cdot\frac{\partial f}{\partial \varphi}$$

For $\psi = 0$, the partial derivatives of $f$ with respect to $r$ and $\varphi$ can be evaluated as follows:

$$\frac{\partial f}{\partial r} = \frac{d}{dr}\left(e^{-\frac{1}{1-r^2}}\right) - e^{-\frac{1}{1-r^2}}C'(r)\sin\psi - e^{-\frac{1}{1-r^2}}C(r)\cos\psi\cdot\frac{\partial\psi}{\partial r}$$

$$= \frac{d}{dr}\left(e^{-\frac{1}{1-r^2}}\right) + e^{-\frac{1}{1-r^2}}C(r)\frac{d}{dr}\left(\frac{1}{1-r^2}\right)$$

$$= -\frac{2r}{(1-r^2)^2}(1-C(r))e^{-\frac{1}{1-r^2}}.$$

$$\frac{\partial f}{\partial \varphi} = -e^{-\frac{1}{1-r^2}}C(r)\cos\psi\cdot\frac{\partial\psi}{\partial\varphi}$$

$$= -e^{-\frac{1}{1-r^2}}C(r).$$

So if $\psi = 0$, then $\frac{d\psi}{dt} = 0$ by the direct calculation below:

$$\frac{d\psi}{dt} = \frac{d\varphi}{dt} - \frac{d}{dr}\left(\frac{1}{1-r^2}\right)\cdot\frac{dr}{dt}$$

$$= Ne^{-h}\rho^{L-2}\left(-\frac{1}{r^2}\frac{\partial f}{\partial\varphi} + \frac{2r}{(1-r^2)^2}\frac{\partial f}{\partial r}\right)$$

$$= Ne^{-h}\rho^{L-2}e^{-\frac{1}{1-r^2}}\left(\left(\frac{1}{r^2} + \frac{4r^2}{(1-r^2)^4}\right)C(r) - \frac{4r^2}{(1-r^2)^4}\right) = 0.$$

**Proof for $r \to 1$.** Let $(\bar{u},\bar{v})$ be a convergent point of $\{(u,v) : t \geq 0\}$. Define $\bar{r} = \sqrt{u^2 + v^2}$. It is easy to see that $r \leq 1$ from the normalization of $\boldsymbol{\theta}$ in the definition. According to Theorem 4.4, we know that $(\bar{u},\bar{v})$ is a stationary point of $\bar{\gamma}(u(t),v(t)) = 1 - f(u(t),v(t))$. If $\bar{r} = 0$, then $f(\bar{u},\bar{v}) > f(u(0),v(0))$, which contradicts to the monotonicity of $\tilde{\gamma}(t) = \bar{\gamma}(t) = 1 - f(u(t),v(t))$. If $\bar{r} < 1$, then $\bar{\psi} = 0$ (defined as $\psi$ for $(\bar{u},\bar{v})$). So $\frac{\partial f}{\partial r} = -\frac{2r}{(1-r^2)^2}(1-C(r))e^{-\frac{1}{1-r^2}} \neq 0$, which again leads to a contradiction. Therefore, $\bar{r} = 1$, and thus $r \to 1$. □

## K   EXPERIMENTS

To validate our theoretical results, we conduct several experiments. We mainly focus on MNIST dataset. We trained two models with Tensorflow. The first one (called *the CNN with bias*) is a standard 4-layer CNN with exactly the same architecture as that used in MNIST Adversarial Examples

Challenge[5]. The layers of this model can be described as `conv`-32 with filter size $5 \times 5$, `max-pool`, `conv`-64 with filter size $3 \times 3$, `max-pool`, `fc`-1024, `fc`-10 in order. Notice that this model has bias terms in each layer, and thus does not satisfy homogeneity. To make its outputs homogeneous to its parameters, we also trained this model after removing all the bias terms except those in the first layer (the modified model is called *the CNN without bias*). Note that keeping the bias terms in the first layer prevents the model to be homogeneous in the input data while retains the homogeneity in parameters. We initialize all layer weights by He normal initializer (He et al., 2015) and all bias terms by zero. In training the models, we use SGD with batch size 100 without momentum. We normalize all the images to $[0, 1]^{32 \times 32}$ by dividing 255 for each pixel.

### K.1 EVALUATION FOR NORMALIZED MARGIN

In the first part of our experiments, we evaluate the normalized margin every few epochs to see how it changes over time. From now on, we view the bias term in the first layer as a part of the weight in the first layer for convenience. Observe that the CNN without bias is multi-homogeneous in layer weights (see (4) in Section 4.4). So for the CNN without bias, we define the normalized margin $\bar{\gamma}$ as the margin divided by the product of the $L^2$-norm of all layer weights. Here we compute the $L^2$-norm of a layer weight parameter after flattening it into a one-dimensional vector. For the CNN with bias, we still compute the smoothed normalized margin in this way. When computing the $L^2$-norm of every layer weight, we simply ignore the bias terms if they are not in the first layer. For completeness, we include the plots for the normalized margin using the original definition (2) in Figure 3 and 4.

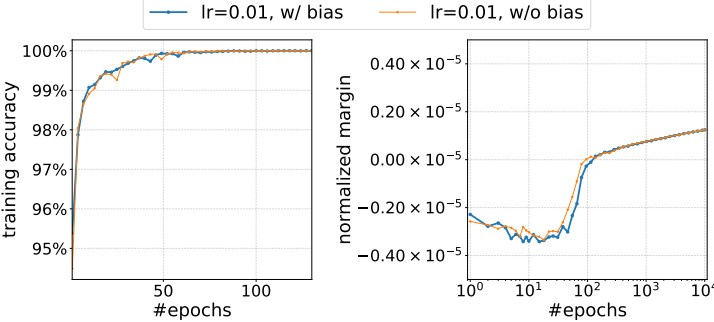

Figure 3: Training CNNs with and without bias on MNIST, using SGD with learning rate 0.01. The training accuracy (left) increases to 100% after about 100 epochs, and the normalized margin with the original definition (right) keeps increasing after the model is fitted.

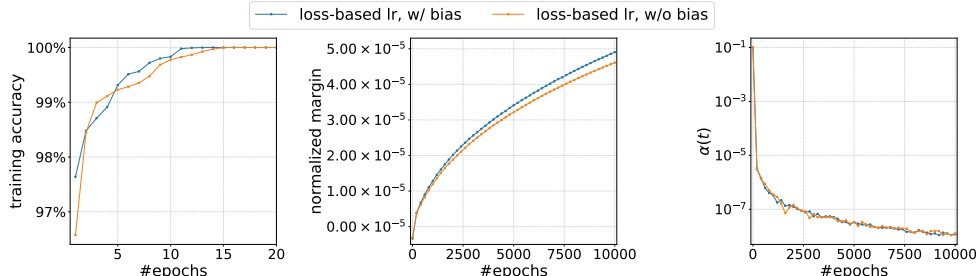

Figure 4: Training CNNs with and without bias on MNIST, using SGD with the loss-based learning rate scheduler. The training accuracy (left) increases to 100% after about 20 epochs, and the normalized margin with the original definition (middle) increases rapidly after the model is fitted. The right figure shows the change of the relative learning rate $\alpha(t)$ (see (27) for its definition) during training.

---

[5]`https://github.com/MadryLab/mnist_challenge`

**SGD with Constant Learning Rate.**    We first train the CNNs using SGD with constant learning rate $0.01$. After about $100$ epochs, both CNNs have fitted the training set. After that, we can see that the normalized margins of both CNNs increase. However, the growth rate of the normalized margin is rather slow. The results are shown in Figure 1 in Section 1. We also tried other learning rates other than $0.01$, and similar phenomena can be observed.

**SGD with Loss-based Learning Rate.**    Indeed, we can speed up the training by using a proper scheduling of learning rates for SGD. We propose a heuristic learning rate scheduling method, called the *loss-based learning rate scheduling*. The basic idea is to find the maximum possible learning rate at each epoch based on the current training loss (in a similar way as the line search method). See Appendix L.1 for the details. As shown in Figure 1, SGD with loss-based learning rate scheduling decreases the training loss exponentially faster than SGD with constant learning rate. Also, a rapid growth of normalized margin is observed for both CNNs. Note that with this scheduling the training loss can be as small as $10^{-800}$, which may lead to numerical issues. To address such issues, we applied some re-parameterization tricks and numerical tricks in our implementation. See Appendix L.2 for the details.

**Experiments on CIFAR-10.**    To verify whether the normalized margin is increasing in practice, we also conduct experiments on CIFAR-10. We use a modified version of VGGNet-16. The layers of this model can be described as `conv-64` $\times 2$, `max-pool`, `conv-128` $\times 2$, `max-pool`, `conv-256` $\times 3$, `max-pool`, `conv-512` $\times 3$, `max-pool`, `conv-512` $\times 3$, `max-pool`, `fc-10` in order, where each `conv` has filter size $3 \times 3$. We train two networks: one is exactly the same as the VGGNet we described, and the other one is the VGGNet without any bias terms except those in the first layer (similar as in the experiments on MNIST). The experiment results are shown in Figure 5 and 6. We can see that the normalize margin is increasing over time.

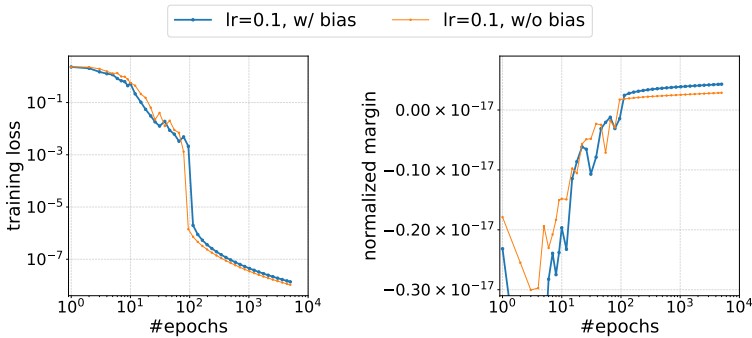

Figure 5: Training VGGNet with and without bias on CIFAR-10, using SGD with learning rate $0.1$.

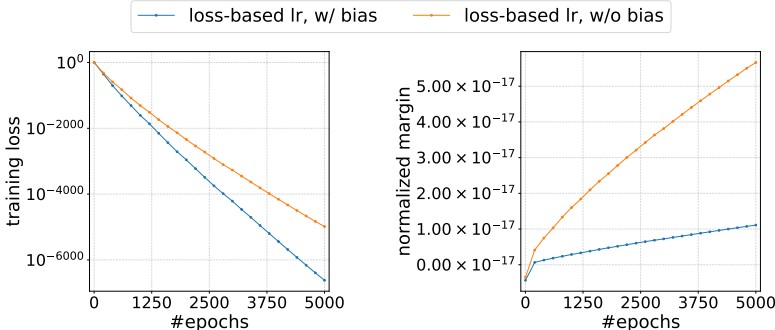

Figure 6: Training VGGNet with and without bias on CIFAR-10, using SGD with the loss-based learning rate scheduler.

**Test Accuracy.**    Previous works on margin-based generalization bounds (Neyshabur et al., 2018; Bartlett et al., 2017; Golowich et al., 2018; Li et al., 2018a; Wei et al., 2019; Banburski et al.,

2019) usually suggest that a larger margin implies a better generalization bound. To see whether the generalization error also gets smaller in practice, we plot train and test accuracy for both MNIST and CIFAR-10. As shown in Figure 7, the test accuracy changes only slightly after training with loss-based learning rate scheduling for 10000 epochs, although the normalized margin does increase a lot. We leave it as a future work to study this interesting gap between margin-based generalization bound and generalization error. Concurrent to this work, Wei & Ma (2020) proposed a generalization bound based on a new notion of margin called all-layer margin, and showed via experiments that enlarging all-layer margin can indeed improve generalization. It would be an interesting research direction to study how different definitions of margin may lead to different generalization abilities.

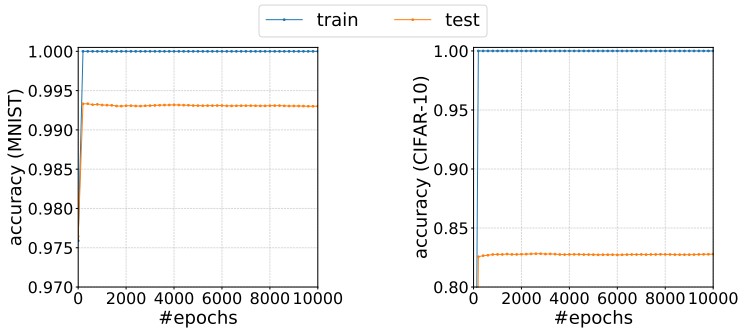

Figure 7: **(Left).** Training and test accuracy during training CNNs without bias on MNIST, using SGD with the loss-based learning rate scheduler. Every number is averaged over 10 runs. **(Right).** Training and test accuracy during training VGGNet without bias on CIFAR-10, using SGD with the loss-based learning rate scheduler. Every number is averaged over 3 runs.

### K.2 EVALUATION FOR ROBUSTNESS

Recently, robustness of deep learning has received considerable attention (Szegedy et al., 2013; Biggio et al., 2013; Athalye et al., 2018), since most state-of-the-arts deep neural networks are found to be very vulnerable against small but adversarial perturbations of the input points. In our experiments, we found that enlarging the normalized margin can improve the robustness. In particular, by simply training the neural network for a longer time with our loss-based learning rate, we observe noticeable improvements of $L^2$-robustness on both the training set and test set.

We first elaborate the relationship between the normalized margin and the robustness from a theoretical perspective. For a data point $z = (x, y)$, we can define the robustness (with respect to some norm $\| \cdot \|$) of a neural network $\Phi$ for $z$ to be

$$R_{\theta}(z) := \inf_{x' \in X} \{\|x - x'\| : (x', y) \text{ is misclassified}\}.$$

where $X$ is the data domain (which is $[0, 1]^{32 \times 32}$ for MNIST). It is well-known that the normalized margin is a lower bound of $L^2$-robustness for fully-connected networks (See, e.g., Theorem 4 in (Sokolic et al., 2017)). Indeed, a general relationship between those two quantities can be easily shown. Note that a data point $z$ is correctly classified iff the margin for $z$, denoted as $q_{\theta}(z)$, is larger than 0. For homogeneous models, the margin $q_{\theta}(z)$ and the normalized margin $q_{\hat{\theta}}(z)$ for $x$ have the same sign. If $q_{\hat{\theta}}(\cdot) : \mathbb{R}^{d_x} \to \mathbb{R}$ is $\beta$-Lipschitz (with respect to some norm $\| \cdot \|$), then it is easy to see that $R_{\theta}(z) \geq q_{\hat{\theta}}(z)/\beta$. This suggests that improving the normalize margin on the training set can improve the robustness on the training set. Therefore, our theoretical analysis suggests that training longer can improve the robustness of the model on the training set.

This observation does match with our experiment results. In the experiments, we measure the $L^2$-robustness of the CNN without bias for the first time its loss decreases below $10^{-10}$, $10^{-15}$, $10^{-20}$, $10^{-120}$ (labelled as `model-1` to `model-4` respectively). We also measure the $L^2$-robustness for the final model after training for 10000 epochs (labelled as `model-5`), whose training loss is about $10^{-882}$. The normalized margin of each model is monotone increasing with respect to the number of epochs, as shown in Table 1.

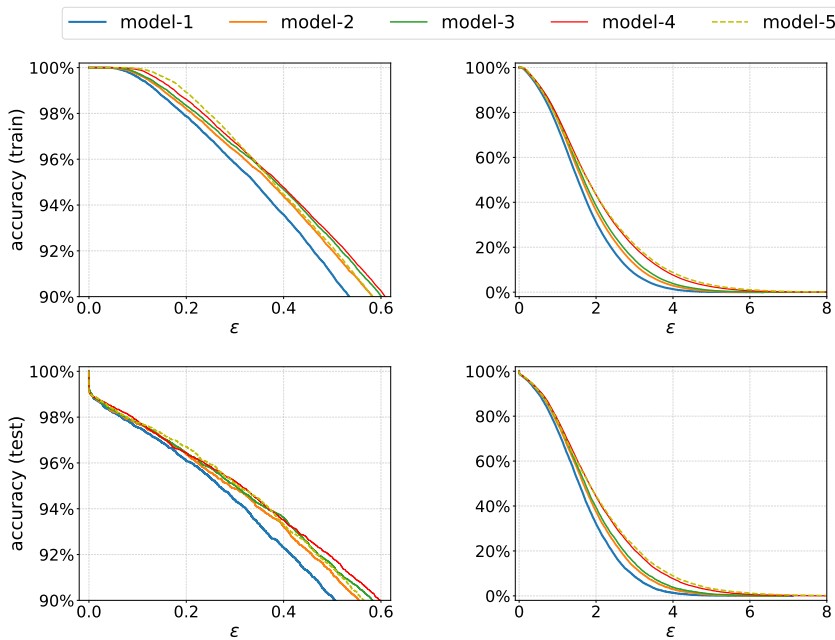

Figure 8: $L^2$-robustness of the models of CNNs without bias trained for different number of epochs (see Table 1 for the statistics of each model). Figures on the first row show the robust accuracy on the training set, and figures on the second row show that on the test set. On every row, the left figure and the right figure plot the same curves but they are in different scales. From `model-1` to `model-4`, noticeable robust accuracy improvements can be observed. The improvement of `model-5` upon `model-4` is marginal or nonexistent for some $\epsilon$, but the improvement upon `model-1` is always significant.

Table 1: Statistics of the CNN without bias after training for different number of epochs.

| model name | number of epochs | train loss | normalized margin | train acc | test acc |
|---|---|---|---|---|---|
| model-1 | 38 | $10^{-10.04}$ | $5.65 \times 10^{-5}$ | 100% | 99.3% |
| model-2 | 75 | $10^{-15.12}$ | $9.50 \times 10^{-5}$ | 100% | 99.3% |
| model-3 | 107 | $10^{-20.07}$ | $1.30 \times 10^{-4}$ | 100% | 99.3% |
| model-4 | 935 | $10^{-120.01}$ | $4.61 \times 10^{-4}$ | 100% | 99.2% |
| model-5 | 10000 | $10^{-881.51}$ | $1.18 \times 10^{-3}$ | 100% | 99.1% |

We use the standard method for evaluating $L^2$-robustness in (Carlini & Wagner, 2017) and the source code from the authors with default hyperparameters[6]. We plot the robust accuracy (the percentage of data with robustness $> \epsilon$) for the training set in the figures on the first row of Figure 8. It can be seen from the figures that for small $\epsilon$ (e.g., $\epsilon < 0.3$), the relative order of robust accuracy is just the order of `model-1` to `model-5`. For relatively large $\epsilon$ (e.g., $\epsilon > 0.3$), the improvement of `model-5` upon `model-2` to `model-4` becomes marginal or nonexistent in certain intervals of $\epsilon$, but `model-1` to `model-4` still have an increasing order of robust accuracy and the improvement of `model-5` upon `model-1` is always significant. This shows that training longer can help to improve the $L^2$-robust accuracy on the training set.

We also evaluate the robustness on the test set, in which a misclassified test sample is considered to have robustness 0, and plot the robust accuracy in the figures on the second row of Figure 8. It can be seen from the figures that for small $\epsilon$ (e.g., $\epsilon < 0.2$), the curves of the robust accuracy of `model-1` to `model-5` are almost indistinguishable. However, for relatively large $\epsilon$ (e.g., $\epsilon > 0.2$), again, `model-1` to `model-4` have an increasing order of robust accuracy and the improvement of

---

[6]https://github.com/carlini/nn_robust_attacks

`model-5` upon `model-1` is always significant. This shows that training longer can also help to improve the $L^2$-robust accuracy on the test set.

We tried various different settings of hyperparameters for the evaluation method (including different learning rates, different binary search steps, etc.) and we observed that the shapes and relative positions of the curves in Figure 8 are stable across different hyperparameter settings.

It is worth to note that the normalized margin and robustness do not grow in the same speed in our experiments, although the theory suggests $R_{\boldsymbol{\theta}}(\boldsymbol{z}) \geq q_{\hat{\boldsymbol{\theta}}}(\boldsymbol{z})/\beta$. This may be because the Lipschitz constant $\beta$ (if defined locally) is also changing during training. Combining training longer with existing techniques for constraining Lipschitz number (Anil et al., 2019; Cisse et al., 2017) could potentially alleviate this issue, and we leave it as a future work.

## L ADDITIONAL EXPERIMENTAL DETAILS

In this section, we provide additional details of our experiments.

### L.1 LOSS-BASED LEARNING RATE SCHEDULING

The intuition of the loss-based learning rate scheduling is as follows. If the training loss is $\alpha$-smooth, then optimization theory suggests that we should set the learning rate to roughly $1/\alpha$. For a homogeneous model with cross-entropy loss, if the training accuracy is $100\%$ at $\boldsymbol{\theta}$, then a simple calculation can show that the smoothness (the $L^2$-norm of the Hessian matrix) at $\boldsymbol{\theta}$ is $O(\bar{\mathcal{L}} \cdot \mathrm{poly}(\rho))$, where $\bar{\mathcal{L}}$ is the average training loss and $\mathrm{poly}(\rho)$ is some polynomial. Motivated by this fact, we parameterize the learning rate $\eta(t)$ at epoch $t$ as

$$\eta(t) := \frac{\alpha(t)}{\bar{\mathcal{L}}(t-1)}, \tag{27}$$

where $\bar{\mathcal{L}}(t-1)$ is the average training loss at epoch $t-1$, and $\alpha(t)$ is a relative learning rate to be tuned (Similar parameterization has been considiered in (Nacson et al., 2019b) for linear model). The loss-based learning rate scheduling is indeed a variant of line search. In particular, we initialize $\alpha(0)$ by some value, and do the following at each epoch $t$:

Step 1. Initially $\alpha(t) \leftarrow \alpha(t-1)$; Let $\bar{\mathcal{L}}(t-1)$ be the training loss at the end of the last epoch;

Step 2. Run SGD through the whole training set with learning rate $\eta(t) := \alpha(t)/\mathcal{L}(t-1)$;

Step 3. Evaluate the training loss $\bar{\mathcal{L}}(t)$ on the whole training set;

Step 4. If $\bar{\mathcal{L}}(t) < \bar{\mathcal{L}}(t-1)$, $\alpha(t) \leftarrow \alpha(t) \cdot r_u$ and end this epoch; otherwise, $\alpha(t) \leftarrow \alpha(t)/r_d$ and go to Step 2.

In all our experiments, we set $\alpha(0) := 0.1, r_u := 2^{1/5} \approx 1.149, r_d := 2^{1/10} \approx 1.072$. This specific choice of those hyperparameters is not important; other choices can only affact the computational efficiency, but not the overall tendency of normalized margin.

### L.2 ADDRESSING NUMERICAL ISSUES

Since we are dealing with extremely small loss (as small as $10^{-800}$), the current Tensorflow implementation would run into numerical issues. To address the issues, we work as follows. Let $\bar{\mathcal{L}}_B(\boldsymbol{\theta})$ be the (average) training loss within a batch $B \subseteq [N]$. We use the notations $C, s_{nj}, \tilde{q}_n, q_n$ from Appendix G. We only need to show how to perform forward and backward passes for $\bar{\mathcal{L}}_B(\boldsymbol{\theta})$.

**Forward Pass.** Suppose we have a good estimate $\widetilde{\mathcal{F}}$ for $\log \bar{\mathcal{L}}_B(\boldsymbol{\theta})$ in the sense that

$$\mathcal{R}_B(\boldsymbol{\theta}) := \bar{\mathcal{L}}_B(\boldsymbol{\theta}) e^{-\widetilde{\mathcal{F}}} = \frac{1}{B} \sum_{n \in B} \log \left( 1 + \sum_{j \neq y_n} e^{-s_{nj}(\boldsymbol{\theta})} \right) e^{-\widetilde{\mathcal{F}}} \tag{28}$$

is in the range of `float64`. $\mathcal{R}_B(\boldsymbol{\theta})$ can be thought of a relative training loss with respect to $\widetilde{\mathcal{F}}$. Instead of evaluating the training loss $\bar{\mathcal{L}}_B(\boldsymbol{\theta})$ directly, we turn to evaluate this relative training loss in a numerically stable way:

Step 1. Perform forward pass to compute the values of $s_{nj}$ with `float32`, and convert them into `float64`;

Step 2. Let $Q := 30$. If $q_n(\boldsymbol{\theta}) > Q$ for all $n \in B$, then we compute

$$\mathcal{R}_B(\boldsymbol{\theta}) = \frac{1}{B} \sum_{n \in B} e^{-(\tilde{q}_n(\boldsymbol{\theta}) + \widetilde{\mathcal{F}})},$$

where $\tilde{q}_n(\boldsymbol{\theta}) := -\text{LSE}(\{-s_{nj} : j \neq y_n\}) = -\log\left(\sum_{j \neq y_n} e^{-s_{nj}}\right)$ is evaluated in a numerically stable way; otherwise, we compute

$$\mathcal{R}_B(\boldsymbol{\theta}) = \frac{1}{B} \sum_{n \in B} \text{log1p}\left(\sum_{j \neq y_n} e^{-s_{nj}(\boldsymbol{\theta})}\right) e^{-\widetilde{\mathcal{F}}},$$

where $\text{log1p}(x)$ is a numerical stable implementation of $\log(1 + x)$.

This algorithm can be explained as follows. Step 1 is numerically stable because we observe from the experiments that the layer weights and layer outputs grow slowly. Now we consider Step 2. If $q_n(\boldsymbol{\theta}) \leq Q$ for some $n \in [B]$, then $\bar{\mathcal{L}}_B(\boldsymbol{\theta}) = \Omega(e^{-Q})$ is in the range of `float64`, so we can compute $\mathcal{R}_B(\boldsymbol{\theta})$ by (28) directly except that we need to use a numerical stable implementation of $\log(1 + x)$. For $q_n(\boldsymbol{\theta}) > Q$, arithmetic underflow can occur. By Taylor expansion of $\log(1 + x)$, we know that when $x$ is small enough $\log(1 + x) \approx x$ in the sense that the relative error $\frac{|\log(1+x) - x|}{\log(1+x)} = O(x)$. Thus, we can do the following approximation

$$\log\left(1 + \sum_{j \neq y_n} e^{-s_{nj}(\boldsymbol{\theta})}\right) e^{-\widetilde{\mathcal{F}}} \approx \sum_{j \neq y_n} e^{-s_{nj}(\boldsymbol{\theta})} \cdot e^{-\widetilde{\mathcal{F}}} \tag{29}$$

for $q_n(\boldsymbol{\theta}) > Q$, and only introduce a relative error of $O(Ce^{-Q})$ (recall that $C$ is the number of classes). Using a numerical stable implementation of LSE, we can compute $\tilde{q}_n$ easily. Then the RHS of (29) can be rewritten as $e^{-(\tilde{q}_n(\boldsymbol{\theta}) + \widetilde{\mathcal{F}})}$. Note that computing $e^{-(\tilde{q}_n(\boldsymbol{\theta}) + \widetilde{\mathcal{F}})}$ does not have underflow or overflow problems if $\widetilde{\mathcal{F}}$ is a good approximation for $\log \bar{\mathcal{L}}_B(\boldsymbol{\theta})$.

**Backward Pass.** To perform backward pass, we build a computation graph in Tensorflow for the above forward pass for the relative training loss and use the automatic differentiation. We parameterize the learning rate as $\eta = \hat{\eta} \cdot e^{\widetilde{\mathcal{F}}}$. Then it is easy to see that taking a step of gradient descent for $\mathcal{L}_B(\boldsymbol{\theta})$ with learning rate $\eta$ is equivalent to taking a step for $\mathcal{R}_B(\boldsymbol{\theta})$ with $\hat{\eta}$. Thus, as long as $\hat{\eta}$ can fit into `float64`, we can perform gradient descent on $\mathcal{R}_B(\boldsymbol{\theta})$ to ensure numerical stability.

**The Choice of $\widetilde{\mathcal{F}}$.** The only question remains is how to choose $\widetilde{\mathcal{F}}$. In our experiments, we set $\widetilde{\mathcal{F}}(t) := \log \bar{\mathcal{L}}(t - 1)$ to be the training loss at the end of the last epoch, since the training loss cannot change a lot within one single epoch. For this, we need to maintain $\log \bar{\mathcal{L}}(t)$ during training. This can be done as follows: after evaluating the relative training loss $\mathcal{R}(t)$ on the whole training set, we can obtain $\log \bar{\mathcal{L}}(t)$ by adding $\widetilde{\mathcal{F}}(t)$ and $\log \mathcal{R}(t)$ together.

It is worth noting that with this choice of $\widetilde{\mathcal{F}}$, $\hat{\eta}(t) = \alpha(t)$ in the loss-based learning rate scheduling. As shown in the right figure of Figure 4, $\alpha(t)$ is always between $10^{-9}$ and $10^0$, which ensures the numerical stability of backward pass.

