# OpenReview forum: "Gradient Descent Maximizes the Margin of Homogeneous Neural Networks"
_ICLR.cc/2020/Conference — Accept (Talk)_

### Official Review · AnonReviewer2 · 2019-10-14
**Official Blind Review #2**

**Rating:** 6

**Review:**

This paper studies the implicit regularization phenomenon. More precisely, given separable data the authors ask whether homogenous functions (including neural networks) trained by gradient flow/descent converge to the max-margin solution.  The authors show that the limit points of gradient descent are KKT points of a constrained optimization problem.

-I think that the topic is important and the authors clearly made some interesting insights.
-The main results of this paper (Theorem 4.1 and Theorem 4.4) require that assumption (A4) is satisfied. Assumption (A4) essentially means, that gradient flow/descent is able to reach weights, such that every data x_n is classified correctly. To me this seems to be a quit restrictive assumption as due to the nonconvexity of the neural net there is a priori no reason to assume that such a point is reached. In this sense, the paper only studies the latter part of the training process.

I feel that Assumption (A4) clearly weakens the strength of the main results. However, because the topic studied by the paper is interesting and the authors have obtained some interesting insights, I decided to rate the paper as a weak accept.

Typos:
-p. 4: "Very Recently"
-p. 7 and p. 9: "homogenuous" (instead of "homogeneous")

----------

I want to thank the authors for their response. However, I will stand by me evaluation and will not change it.
I agree though that assumption (A4) is indeed reasonable, although of course very strong.


**Experience Assessment:**

I do not know much about this area.

**Review Assessment: Checking Correctness Of Derivations And Theory:**

I did not assess the derivations or theory.

**Review Assessment: Checking Correctness Of Experiments:**

N/A

**Review Assessment: Thoroughness In Paper Reading:**

I read the paper at least twice and used my best judgement in assessing the paper.

---

> ### Author Response · Authors · 2019-11-13
> **Response to Reviewer #2**
>
> Thanks for your reviews and for pointing out the typos!
>
> We admit that (A4) may not hold for all neural networks and all datasets. Indeed, the loss of a neural network is highly non-convex and (A4) seems to be a quite strong assumption. However, it is known that sufficiently overparameterized neural networks can fit the training set through (stochastic) gradient descent. As we discussed in the introduction of our paper, state-of-the-art neural networks are typically overparameterized, and they can perfectly fit not only normal data but also randomly labeled data easily in image classification tasks (Zhang et al., 2017). Theoretically, (Allen-Zhu et al., 2019; Du et al. 2018; Zou et al., 2018) showed that gradient descent can achieve 100% training accuracy if the width is large enough. Given the evidence from both theory and practice, we believe (A4) is a reasonable assumption (at least for many DL tasks).

---

### Official Review · AnonReviewer3 · 2019-10-21
**Official Blind Review #3**

**Rating:** 8

**Review:**

The goal of the paper is to formally prove that gradient flow / gradient descent performed on homogeneous neural network models maximizes the margin of the learnt function; assuming gradient flow/descent manages to separate the data. This is proved in two steps:
  1. Assuming that gradient descent manages to find a set of network parameters that separate the data, thereafter gradient flow/descent monotonically increases the normalized margin (rather an approximation of it).
  2. The limit points of optimization are KKT points of the margin maximization optimization problem.
While the main body of the paper presents a restricted set of results, the appendix generalizes this much further applying it to various kinds of loss functions (logistic/cross-entropy, exponential), to multi-class classification and to multi-homogeneous models. There seem to be many subtleties in the proofs and the paper seems to be quite thorough. (I must say that I'm not expert enough to assess the technical novelty of this paper over prior works.)

Recommendation:
I recommend "acceptance". The paper takes a significant step by unifying existing results on margin maximization and going beyond them.

Technical comments:
- It is clear that in order to define margin meaningfully, some form of normalization is necessary. But a priori, $\|\theta\|_2^L$ is not the *only* choice; $\|\theta\|^L$ could also work for any norm $\|\cdot\|$. But perhaps the choice of $\|\cdot\|_2$ is special (as Thm 4.4 suggests). It will be nice to have some insights/comments on why this choice of $\|\cdot\|_2$ based normalization is the right one.
- The paper argues that having a larger margin helps in obtaining better robustness to adversarial perturbations (within $\|\cdot\|$ balls for some choice of $\|\cdot\|$). However note that the notion of "margin" is not just a function of the decision boundary, but instead depends on the specific function computed by the neural network --- this is unlike margin maximization in linear models, where "margin" in determined entirely by the decision boundary. As the paper argues, if we have an upper bound on the Lipschitz constant w.r.t. $\|\cdot\|$ norm, then we get a lower bound on required adversarial perturbations for any training point. However, this does not mean that training longer is necessarily better because by doing so, we might end up with a larger Lipschitz constant (even after normalizing). So even if the "margin" is larger, the actual adversarial perturbations (in $\|\cdot\|$ norm) allowed might get smaller. So I'm not sure how relevant this result is for adversarial robustness.


**Experience Assessment:**

I have read many papers in this area.

**Review Assessment: Checking Correctness Of Derivations And Theory:**

I assessed the sensibility of the derivations and theory.

**Review Assessment: Checking Correctness Of Experiments:**

I assessed the sensibility of the experiments.

**Review Assessment: Thoroughness In Paper Reading:**

I read the paper at least twice and used my best judgement in assessing the paper.

---

> ### Author Response · Authors · 2019-11-13
> **Response to Reviewer #3**
>
> Thanks for your comments!
>
> The L2-normalization is due to the use of gradient descent (GD is the steepest descent algorithm w.r.t. L2). If we change the optimization algorithm, the normalized margin being optimized should be also changed. Note that this has been studied in the linear case (Gunasekar et al., 2018a): if we run steepest descent with respect to a generic norm $\|\cdot\|$, then the $\|\cdot\|$-normalized margin is maximized. When $\|\cdot\|$ is the L2 norm, it is just the case of gradient descent; when $\|\cdot\|$ is the L1 norm, the corresponding optimization problem is coordinate descent, and it maximizes the L1-normalized margin. Right now, our results only hold for gradient descent and L2 norm. Extending it to more general norm and optimization problems is an interesting future direction.
>
> For robustness, we want to emphasize that the Lipschitz constant is evaluated after normalizing the weight norm. As the weight norm is always $1$ during training, we can expect that the Lipschitz constant does not get extremely large. In our experiments, we admit that the normalized margin and robustness do not grow in the same speed, so the Lipschitz constant may change; however, the normalized margin and robustness do have quite positive correlations, and we think improving robustness by maximizing normalized margin is relevant (it may be able to provide certified robustness). So we will discuss this phenomenon in the next version of our paper to encourage further discussions.

---

### Official Review · AnonReviewer1 · 2019-10-23
**Official Blind Review #1**

**Rating:** 8

**Review:**

This is a strong deep learning theory paper, and I recommend to accept.

This paper studies the trajectory induced by applying gradient descent/gradient flow for optimizing a homogeneous model with exponential tail loss functions, including logistic and cross-entropy loss in particular. This is an important direction in recent theoretical studies on deep learning as we need to understand which global minimizer the training algorithm picks to analyze the generalization behavior.

This paper makes a significant contribution to this direction. This paper rigorously proves gradient descent / gradient flow can maximize the L2 margin of homogeneous models. Existing works mostly focus on linear models or deep linear networks, and comparing with Nascon et al., 2019a, the assumptions in this paper are significantly weaker. Furthermore, this paper provides convergence rates, which seem to be the first work of this kind for non-linear models.

I really like Lemma 5.1. This is not only a technical lemma for proving the main theorem. Lemma 5.1 itself has a nice geometric interpretation. It naturally decomposes the dynamics of the smoothed version into a radial component and a tangential velocity component. I believe this lemma can be useful in other settings as well.


Comments:
The bibliography should be fixed. Some papers are already published, so they should not be cited as the arXiv version, and author lists in some papers have "et al."

-----------------------------------------------------
I have read the rebuttal and I maintain my score.

**Experience Assessment:**

I have published in this field for several years.

**Review Assessment: Checking Correctness Of Derivations And Theory:**

I assessed the sensibility of the derivations and theory.

**Review Assessment: Checking Correctness Of Experiments:**

N/A

**Review Assessment: Thoroughness In Paper Reading:**

I read the paper thoroughly.

---

> ### Author Response · Authors · 2019-11-13
> **Response to Reviewer #1**
>
> Thanks for your appreciation! We will fix the errors in the bibliography.

---

### Decision · Program_Chairs · 2019-12-19

**Decision:**

Accept (Talk)

**Comment:**

This paper studies the implicit regularization of the gradient descent in homogeneous and shows that when the training loss falls below a threshold, then the smoothed. This study generalizes some of the earlier related works by relying on weaker assumptions. Experiments on MNIST and CIFAR-10 are provided to backup the theoretical findings of the paper.
R2 had some concern about one of the assumptions in this work (A4). While authors admitted that (A4) may not hold for all neural networks and all datasets, they stressed that this assumptions is reasonable when the network is overparameterized and can perfectly fit the training data. Overall, all reviewers are very positive about this submission and find a valuable step toward understanding implicit regularization.